# Modeling the European Neolithic expansion suggests predominant within-group mating and limited cultural transmission

Troy M. LaPolice [ID], Matthew P. Williams [ID] & Christian D. Huber [ID] ✉

The Neolithic Revolution initiated a pivotal change in human society, marking the shift from foraging to farming. Historically, the underlying mechanisms of agricultural expansion have been a topic of debate, centered around two primary models: cultural diffusion, involving the transfer of knowledge and practices, and demic diffusion, characterized by the migration and replacement of populations. More recently, ancient DNA analyses have revealed significant ancestry changes during Europe's Neolithic transition, suggesting a primarily demic expansion. Nonetheless, the presence of 10-15% hunter-gatherer ancestry in modern Europeans indicates cultural transmission and between-group mating were additional contributing factors. Here, we integrate mathematical models, agent-based simulations, and ancient DNA analysis to dissect and quantify the roles of cultural diffusion and between-group mating in farming's expansion. Our findings indicate limited cultural transmission and predominantly within-group mating. Additionally, we challenge the assumption that demic expansion always leads to ancestry turnover. These results offer insights into early agricultural society through the integration of ancient DNA with archaeological models.

The transition to agriculture was transformative for human history, with pronounced effects on economics[1-3], population size[4,5], ancestry[6,7], language[8,9], and societal structure[5,10]. Deemed the "Neolithic Revolution"[11-13], this change in behavior and culture set the basis for modern society, facilitating the construction of permanent settlements and structures within complex social networks centered around a more stable food and resource supply[10,14]. The shift to farming has also been associated with reductions in population health due to nutritional deficiencies and increased disease transmission[14-17]. As a result of the vast impacts of this change on societal structure and its health-related consequences, researchers have long been interested in reconstructing how and why the transition to agriculture occurred.

The spread of agriculture in prehistoric Europe has been a longstanding subject of research, with early archeological studies focusing on the spatial and temporal distribution of material culture associated with farming communities[11,18,19]. Pioneering work by scholars such as V. Gordon Childe introduced the concept of the Neolithic Revolution[11-13], emphasizing the transformative impact of agriculture on human societies. Many archeologists suggested migration as a potential driver of the spread of agriculture, though later some proposed cultural diffusion, where farming practices spread without significant population movement[18,20,21]. Building on these foundational insights, Ammerman and Cavalli-Sforza[22] first introduced a quantitative framework for understanding the Neolithic expansion that integrated archeological data with genetic expectations[22,23].

It is understood that Agriculture emerged independently at several centers[24] in the Fertile Crescent, each with genetically distinct origins[6]. From there, it spread into Europe via central Anatolia. Notably, Anatolia was more than just a corridor for the movement of early farmers; it was a region where local hunter-gatherers transitioned to an agricultural lifestyle, a way of life that then expanded broadly outward[6,25,26,27,28]. Using radiocarbon-dated sites to calculate the speed of the farming expansion across Europe (the so-called "front speed"), Ammerman and Cavalli-Sforza determined an average expansion rate

Pennsylvania State University, Department of Biology, University Park, PA, USA. ✉e-mail: cdh5313@psu.edu

of approximately 1 km per year[22,29] which has been more recently corroborated by Pinhasi et al.[30].

In addition to estimating the front speed, studies have also sought to understand the specific mechanism that drove this expansion. One mechanism is demic diffusion, first coined in 1971 by Ammerman and Cavalli-Sforza[22]. Demic diffusion describes an expansion of agriculture driven by farmers migrating into previously un-farmed territories, thereby introducing their cultural practices as well as genetic ancestry into these new areas[22,29,31–33]. Ammerman and Cavalli-Sforza argued that the rate of farming expansion in Europe is principally compatible with a primarily demic model of population growth and displacement, and suggested that this would result in a genetic cline of Neolithic ancestry in the resulting populations[22,29].

Alternatively, under a cultural diffusion model, first discussed and applied to the Neolithic Expansion by Edmonson in 1961[20], hunter-gatherer (HG) groups learn agriculture, and acquire the means necessary to practice this culture, by living near farmers[20,22,29,31–33]. Within the cultural diffusion process, both vertical and horizontal modes of transmission have been considered[33]. Horizontal transmission refers to individuals learning behaviors from peers in other populations (either by choice or by force), whereas vertical transmission is the passing down of cultural practices from parents to offspring[33]. Notably, in a cultural diffusion model, farming ancestry does not expand to the degree it does in a demic model. This is because farming expands predominantly by learning or imitation, instead of farmers (and their genomes) moving into—and overtaking—new territories.

Aside from the perhaps unrealistic extremes of a fully demic or fully cultural model, the two models can be combined into a demic-cultural mode of transmission, i.e., where farmers migrate into new land but also interact with HGs as they expand, either actively or passively passing knowledge onto those they meet[31–33]. Under this model, both mechanisms are thought to contribute, in varying degrees, to the expansion of farming.

Several previous studies have aimed to quantify the relative contribution of cultural vs. demic transmission under a combined demic-cultural model[31–33]. One measurement used to quantify the cultural contribution is "cultural effect". Cultural effect is defined as the additional contribution of cultural transmission to the front speed on top of an otherwise demic mode of expansion, i.e., the relative increase in front speed due to cultural transmission[31–33]. Previous estimations have suggested that cultural effect could account for anywhere between 0% and ~40% of the expansion speed of farming[34], with others showing a maximum cultural effect of 21% to be consistent with the observed front speed[33]. These differences are mainly due to assumptions on the dispersal behavior of early farmers and the mode of cultural transmission. Thus, these estimates indicate primarily demic expansion, but also display a large uncertainty regarding a precise quantification of cultural effect. Importantly, these estimates are based only on the speed of the expansion and do not consider patterns of genetic ancestry resulting from the respective models. This is important because a purely demic model (zero learning and full within-group mating) predicts a complete genetic turnover, whereas cultural transmission allows for the persistence of indigenous HG ancestry after the expansion.

In addition to mathematical modeling of the Neolithic expansion[31–33,35–39], other studies have gathered novel insights from investigating settlements[40–44], geographical factors[45], language[8], ancient climate conditions[46], and paleoenvironmental data[47]. In particular, recent analyses of ancient DNA (aDNA) have fundamentally transformed our understanding of population movements and associated genetic changes that took place during this period. With the advent of aDNA, researchers were able to scan for genetic adaptations in Neolithic populations[7], determine distinct ancestry patterns and population movements[48], or describe the contributions of ancient groups to modern genomes[49]. Early work based on mitochondrial data

lent support for a significant local hunter-gatherer contribution to modern European populations[50] (i.e., a predominantly cultural expansion), whereas more recent model-based analyses of mitochondrial haplotype frequencies indicated only a 0.7–2.3% range for the cultural effect[34,51]. Ever-growing sample sizes, and especially whole-genome sequences, have revealed in greater detail the significant temporal and geographical ancestry shifts co-occurring with the spread of the Neolithic cultural package[6,7,47,52,53]. But despite aDNA technology providing unprecedented resolution of genetic ancestry and population movements over time, a limitation of DNA is that it does not directly describe or indicate behavior, culture, or specific mechanisms of expansion. As such, understanding the Neolithic farming expansion requires an interdisciplinary approach that integrates behavioral mechanisms with genetic data.

Here, we build on the evolving field of archaeogenetics by incorporating behavioral simulations grounded in both archeological and genetic evidence. Our aim was to create a model that identifies cultural transmission parameters consistent with both archeological and genetic data, enabling a more detailed understanding of the mechanisms responsible for this pivotal societal change. We provide an approach which utilizes a demographic mathematical model and a spatial, agent-based population genetic simulation. We model mortality under density-dependent competition based on carrying capacity estimates from literature[29,54], and osteological age at death data from Neolithic samples[42,55]. Throughout, in describing the groups associated with the Neolithic shift, we use the term Early Farmer (EF) as representative of the genetic ancestry associated with this spread of early farming, and Western Hunter-Gatherer (WHG) representing the local European Mesolithic ancestry. We fit parameters in our model to archaeological front speed estimates[22,29,30] and ancestry estimates derived from 618 European Neolithic individuals that were plausibly modeled as mixtures of WHG and EF ancestry. This allows us to elucidate which cultural transmission characteristics are compatible with archeological and genetic data. We conclude that there must have been near-complete within-group mating in farming and hunter-gathering groups (i.e., very few farmer-HG matings). In addition, we find few farmers participated in cultural transmission, suggesting that less than 0.1% of farmers converted a hunter-gatherer to farming annually. More generally, our modeling cautions that primarily demic expansion does not necessarily lead to a major ancestry turnover, suggesting that a naive interpretation of ancient ancestry patterns does not always reflect underlying behavioral mechanisms.

## Results
### Basic model suggests low levels of cultural transmission
We started by examining the influence of cultural transmission on the dispersal of EF ancestry throughout a Europe inhabited by hunter-gatherers (HGs). We employed a reaction-diffusion model in a one-dimensional (1D) continuum, as a simplified representation of this process. Reaction-diffusion models are commonly used in ecology to describe how populations expand and interact over space and time. These models combine two processes: diffusion, representing random movement or dispersal of individuals, and reaction, which captures local dynamics such as population growth or between-species interaction. In the context of the Neolithic expansion, reaction-diffusion models provide a framework for simulating how farming populations spread geographically and interact with local HG populations. Our model consists of a system of partial differential equations that mirror those utilized in prior models addressing the spread of agriculture (e.g., Fort et al., 2012)[31] and of the farmer-specific mitochondrial haplogroup K[51]. Our basic model, similarly to previous ones[34,51,56], incorporates the genetic ancestry of individuals at a single genetic locus or marker, enabling us to explore how cultural transmission and demographic processes jointly shaped the ancestry landscape of prehistoric Europe.

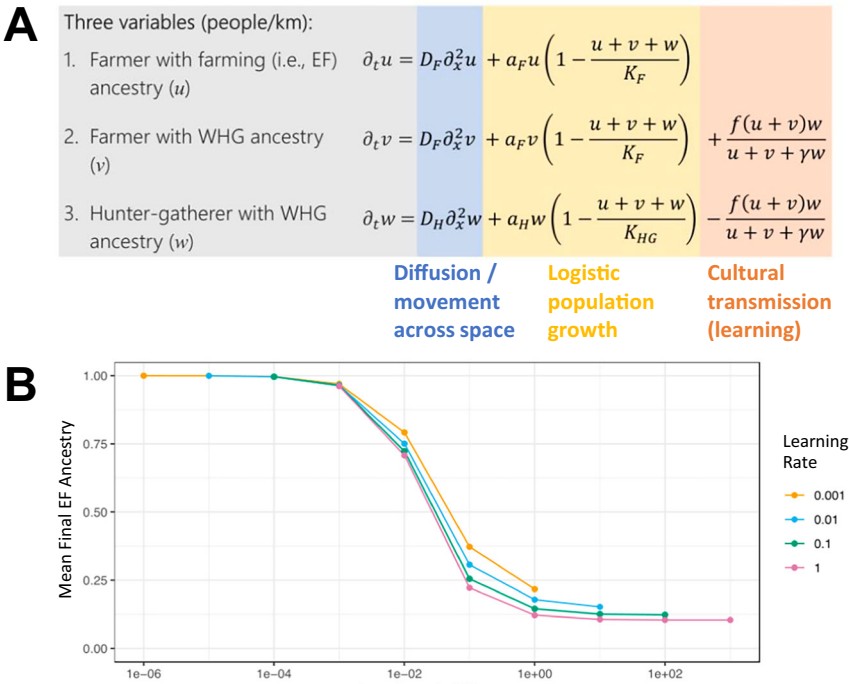

**Fig. 1 | One-dimensional model. A** Our mathematical model utilizes a reaction-diffusion framework to simulate the spread of EF ancestry across a one-dimensional European landscape initially populated by hunter-gatherers. It comprises a set of partial differential equations that describe the movement (diffusion) of individuals, their population growth (logistic), and the impact of cultural transmission on genetic ancestry. For a comprehensive description of the symbols and components of the equations, see Supplementary Note 1. The model tracks three distinct groups: farmers with EF ancestry, farmers with WHG ancestry, and hunter-gatherers with WHG ancestry, each affected by diffusion constants, growth rates, and carrying capacities. EF ancestry is calculated as $u/(u + v + w)$, when farming is fixed and $w = 0$. The cultural transmission aspect is modeled through parameters that dictate the learning rate of new practices ($f$) and a bias toward adopting farming methods from either farmers or hunter-gatherers ($\gamma$). **B** Relationship between the ratio of the learning rate to the bias parameter ($C = f/\gamma$) and the proportion of EF ancestry in the model, averaged across the landscape. For this one-dimensional model, we calculated the EF ancestry proportion at generation 100, shortly after all HGs have been acculturated. This plot illustrates that the final proportion of EF ancestry after the Neolithic expansion is predominantly determined by the ratio $C/\gamma$, rather than the individual values of $f$ or $\gamma$. Colored data points indicate model outcomes across a range of learning rates ($f$) and bias parameters ($\gamma$).

We defined three population groups within the model: cultural farmers with EF ancestry, cultural farmers with WHG ancestry, and cultural HGs of WHG ancestry. As an initial condition, we assumed that the farmers of EF descent are constrained to 200 km at the left end of the range, and the HG group is constrained to the remaining 2800 km of the range, with twenty times lower population density[29]. The spatio-temporal distribution of these groups is then influenced by three principal dynamics: random migration, which is determined by a diffusion constant; logistic population growth, modulated by growth rates and carrying capacities; and cultural transmission through learning. Our cultural transmission mechanism is informed by established theory[31], considering a learning rate ($f$; the product of the number of teachers times the probability of learning per contact), and a bias parameter ($\gamma$), which regulates the HGs' propensity to adopt farming practices from nearby farmers (see Fig. 1A; for a comprehensive description of the symbols and components of the equations, see Supplementary Note 1).

We solved the model equations numerically to investigate the resulting change in farming ancestry, shortly after all HGs have been acculturated, over a time span of 100 generations and across the 3000 km transect. Published aDNA evidence suggests that Neolithic Europe exhibited significant EF ancestry, exceeding 75% across all of Europe[47]. To evaluate which of our numerical solutions are consistent with this pattern, we assessed the average proportion of EF ancestry after the transition to agricultural practices is complete−i.e., when the population of hunter-gathering individuals is nil. We acknowledge the existence of farming individuals in certain regions and time periods who may also practice elements of hunter-gatherer subsistence,

however, for the simplicity of the model, we consider individuals who predominantly farm as farmers.

We based the diffusion constant, growth rate, and carrying capacities for both HGs and farmers on previous estimates from literature[29,54]. By altering the values of $f$ and $\gamma$, we observed that the final spatially-averaged EF ancestry after the Neolithic expansion correlates most strongly with the ratio $C = f/\gamma$, rather than the individual parameters $f$ or $\gamma$, even when $f$ varies over four orders of magnitude (see Fig. 1B). Moreover, an increase in the $f/\gamma$ ratio strongly correlates with a decrease in EF ancestry, falling below 50% when the ratio reaches 0.1 per generation− an outcome that is highly inconsistent with the predominant EF ancestry proportion estimated from aDNA for Neolithic Europe[47].

Given that the ratio of the learning rate to the bias parameter is the critical determinant in this model, we fixed $\gamma$ to 1 (signifying unbiased learning) in subsequent analyses and focused solely on variations in the learning rate $f$. Thus, under unbiased learning, we find an upper limit to the learning rate of approximately 0.1 per generation (~ 0.004 per year, assuming a 25-year generation time) for the proportion of EF ancestry in our model to be consistent with the high levels observed in the aDNA data following the agricultural expansion.

Nonetheless, we acknowledge the model's simplifications: an assumption of infinite population size, a single genetic locus without recombination, no age stratification or age-linked mortality, a uni-dimensional habitat, and exclusive mating within cultural groups. Subsequent sections will present simulations under more realistic conditions to verify the consistency of our findings and to refine our

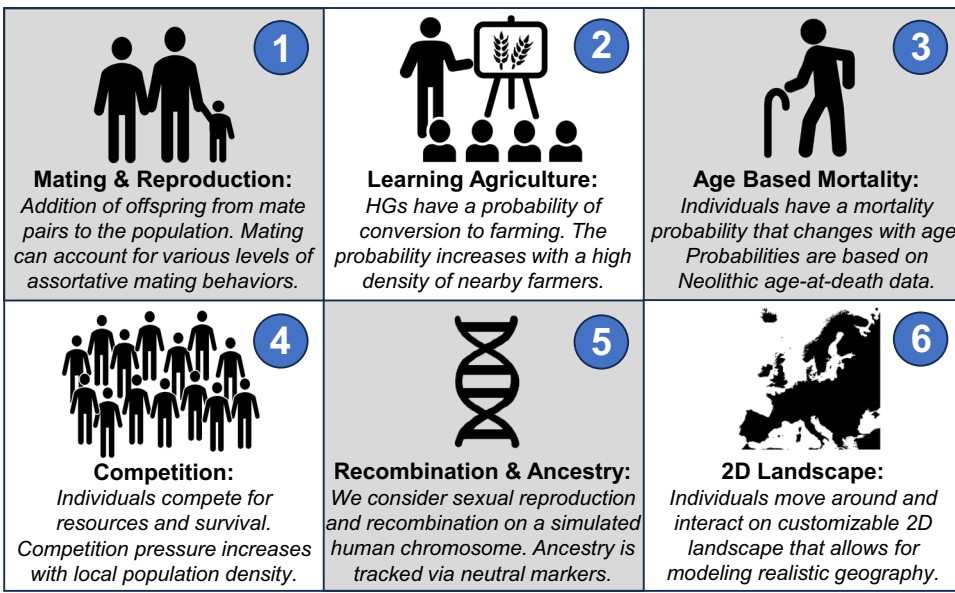

**Fig. 2 | Overview of the components in the agent-based simulation approach.**
**1** Our model includes mating between individuals in the population under frequencies governed by within-group mating preferences and local abundance of cultural groups. Through reproduction, we simulate vertical (parent-offspring) cultural transmission. **2** We also consider hunter-gatherers learning agricultural behaviors via horizontal (peer-to-peer) cultural transmission. Learning occurs with a probability that is determined by a learning rate $f$ and the local proportion of farmers surrounding a given hunter-gatherer. **3** Age-based mortality is implemented using probabilistic mortality curves derived from Neolithic age-at-death studies[43,56]. **4** In addition, mortality is also governed by local competition for resources and considers different carrying capacities for farmers and hunter-gatherers. **5** Each individual has a simulated chromosome that is generated via sexual recombination, allowing us to track ancestry proportions at an individual and population level. **6** Finally, the simulation is two-dimensional, with each individual having a location in space. Individuals exist and interact in a customizable virtual landscape. (Map depicted in Fig. 2 adapted from the European Environmental Agency's (EEA) Elevation map of Europe[59].

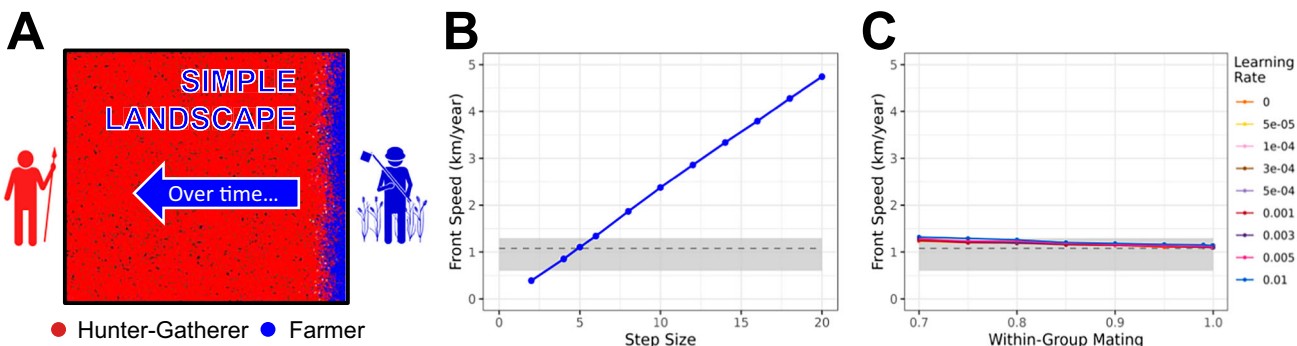

**Fig. 3 | Simple square landscape and the effect of step size, learning rate, and within-group mating on front speed. A** Overview illustrating the progression of farming expansion on a simple square landscape. All front-speed calculations were done using the square landscape model depicted in panel **A**. Over time, the red hunter-gatherer individuals will be displaced (or converted) by blue farmers.
**B** Effect of different σ values (referred to as step sizes) on front speed. For comparisons, the gray shaded box illustrates the front speed range of 0.6–1.3 km/yr estimated by Pinhasi et al.[31], and the gray dashed line represents a front speed of 1.08 km/yr estimated by Ammerman and Cavalli-Sforza[22]. **C** Effect of within-group mating and different learning rates on front speed, given a fixed step size of 5 km. Colored lines represent different learning rates. The gray shaded box again represents the front speed range of Pinhasi et al.[31] and the gray dashed line the front speed estimate by Ammerman and Cavalli-Sforza[22].

estimates of the extent of cultural transmission compatible with the archeological record.

## Front speed is not informative of cultural transmission rate
To assess the validity of the conclusions under more realistic scenarios, we turned to an agent-based model capable of simulating complex individual behavior (Fig. 2). This model, designed in SLiM (v4.0)[57], considers unique individuals with sexually recombining genomes, age-based mortality, and local resource competition. Further, each individual moves and interacts on a customizable two-dimensional landscape that we utilize to simulate both simple and complex geography, adapted from a topological map of Europe[58]. Finally, the agent-based

model has a cultural component in which HGs can learn and convert to agricultural lifestyles. This happens at a rate determined by the previously introduced learning rate parameter, $f$, and local farmer density (Fig. 2).

As the simulation progresses, individuals move across the landscape and interact with each other (Fig. 3A). Each year, for every individual, a distance in the $X$-direction (East-West) and a distance in the $Y$-direction (North-South) is drawn from a normal distribution and added to the individual's previous $X$, $Y$ position. The standard deviations of the distributions (σ) allow us to control the distance individuals can travel within a year. We will refer to σ as "step size".

**A**

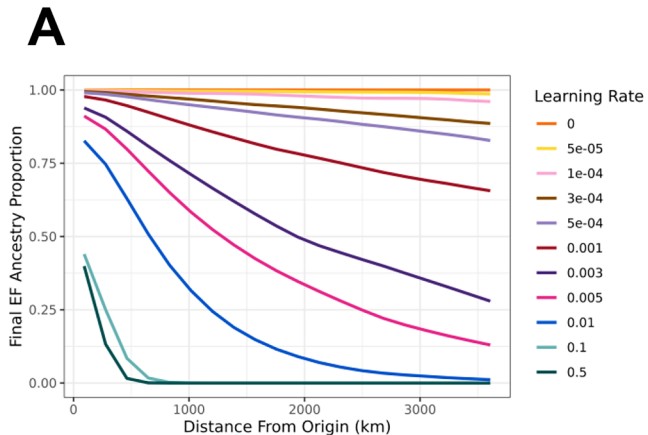
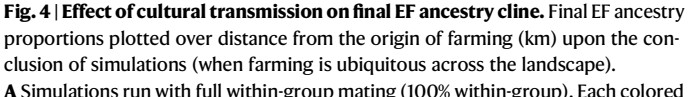

**B**

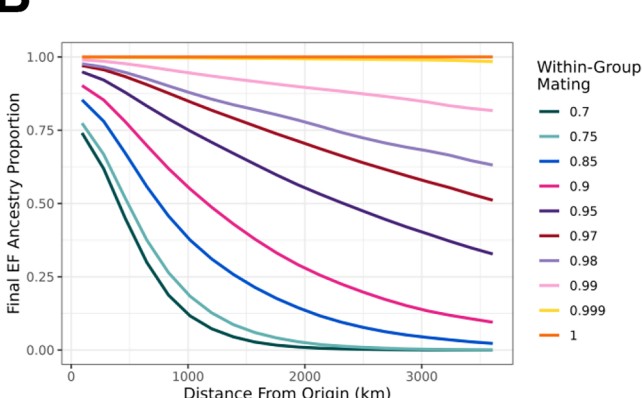

**Fig. 4 | Effect of cultural transmission on final EF ancestry cline.** Final EF ancestry proportions plotted over distance from the origin of farming (km) upon the conclusion of simulations (when farming is ubiquitous across the landscape). **A** Simulations run with full within-group mating (100% within-group). Each colored line represents a different learning rate simulated. **B** Simulations run with zero learning but various within-group mating probabilities. Each colored line represents a different within-group mating probability.

Similar to the diffusion constant in the mathematical model, the step size impacts the speed at which farmers diffuse across the landscape, and as a consequence, the speed of the cultural expansion (Fig. 3B). Based on estimates from the literature, we expect a front speed of approximately 1 km per year[22,29,30] (gray box and dashed line in Fig. 3B, C). These studies estimate the front speed of farming expansion by using C14 radiocarbon dates from Neolithic sites, plotting the earliest evidence of farming against distances from a proposed origin to calculate the average rate of spread[22,29,30].

To identify plausible $\sigma$ values that are consistent with this front speed, we tested 11 values of $\sigma$ between 2–20 km, in increments of 2 km, and calculated the front speed of each simulation under a fully demic model (see "Methods"). We found a $\sigma$ of 5 km to be the most plausible of the tested values (Fig. 3B) and used this step size as our standard throughout the rest of the simulations.

Next, we explored the effect of within-group mating and learning on front speed (Fig. 3C). Using 5 km as a fixed $\sigma$, we see that learning rates and within-group mating parameter values (within realistic ranges) have a negligible effect on the front speed. To affect front speed, very high rates of cultural transmission are required (Supplementary Fig. 1), however, these high rates almost completely inhibit the spread of EF ancestry into Europe (Fig. 1) and thus are incompatible with empirical data[47]. We conclude that, although the front speed is informative about the step size parameter of the model, it does not provide fine-scale insight into the degree of cultural transmission. This conclusion agrees with the wide ranges of the cultural effect calculated from the front speed by using intergenerational dispersal distances and probabilities measured for ethnographic preindustrial farming populations[34]. This is notable because it highlights the importance of co-analyzing ancestry and front speed in a combined model. In subsequent analyses, we explore the relationship between cultural transmission and genetic ancestry patterns and evaluate if ancestry patterns allow the estimation of cultural transmission parameters.

### Effect of cultural transmission on ancestry cline

We used our agent-based model to investigate the effect of within-group mating and learning on EF ancestry proportions. Previous literature observed a cline of Anatolian ancestry with distance from the farming origin in European Neolithic populations[47]. In our simulations, we again assume that initially, all cultural farmers are of pure EF ancestry, and we explore how this ancestry spreads with the cultural expansion of farming across a simple square landscape (Fig. 3A). We varied our learning rate parameter and restricted mating to be 100%

within-group by inhibiting matings between cultural farmers and HGs. We found all learning parameters result in a cline of EF ancestry–estimated once all individuals are practicing farming (i.e., when there are no longer any hunter-gatherer individuals)–that decreases with increasing distance from the farming origin. As the learning rate increases, the slope of the ancestry cline also increases (Fig. 4A), such that learning rates larger than 0.003 per year result in less than 50% final EF ancestry across most of the simulated landscape (Fig. 4A). This is consistent with our one-dimensional reaction-diffusion model that saw an equally strong reduction in final ancestry proportions for equivalent learning rates, further demonstrating that even low levels of learning can impede the spread of EF ancestry.

To determine the sole effect of vertical cultural transmission (from between-group mating) on ancestry proportions, we fixed the learning rate to 0, i.e., restricting cultural transmission to cases where the offspring of a farmer and hunter-gatherer adopts farming. We then varied the within-group mating parameter, selecting 10 values between 0.7 and 1 (Supplementary Fig. 2). We found that low levels of between-group mating (i.e., HG with farmer) lead to similar ancestry clines (Fig. 4B) as the simulations with variable learning rates (Fig. 4A). This shows that both vertical and horizontal cultural transmission are analogous in their genetic effects. We further saw a rapid loss of EF ancestry when we decreased the within-group mating rate (i.e., increased the between-group rate), such that with within-group mating rates less than 90%, EF ancestry quickly declined to zero toward the far end of the range (Fig. 4B). This has been previously observed using mtDNA haplogroup frequencies[51], and is analogous to what we observed with learning rates greater than 0.01 per year (Fig. 4A).

In sum, both horizontal and vertical cultural transmission can generate spatial clines in EF ancestry after the farming expansion, with higher rates of cultural transmission leading to steeper clines and lower final proportions. Given that the proportion of EF ancestry estimated from aDNA in post-Neolithic Europe is considerably high, we conclude that cultural transmission must have only played a limited role in the farming expansion. This conclusion is also supported qualitatively by the major genetic[59] and genomic[60] turnover in Europe at the time of the Neolithic transition, as well as quantitatively by the low intensities of cultural transmission implied by the shapes of mitochondrial and Y-chromosome clines in Neolithic Europe[34,51,56].

### Estimating cultural transmission parameters from ancient DNA

Based on our aDNA estimations (see below) and published literature[47], we know there to be a cline in EF ancestry and not a

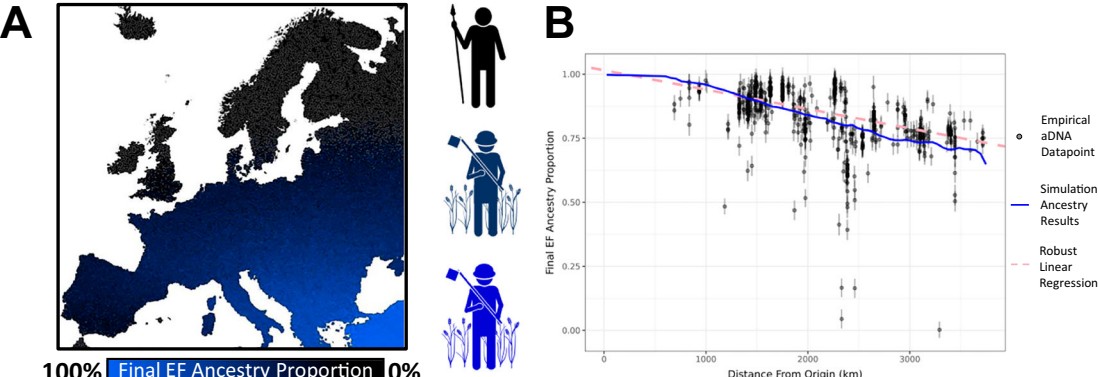

**Fig. 5 | Effect of cultural transmission on final EF ancestry proportion on a European landscape.** Point estimates of the final EF ancestry proportion derived from the software qpAdm[63] over distance from farming origin (km) plotted upon the conclusion of simulations when farming is ubiquitous across the landscape. **A** Figure showing a visual depiction of a purely hypothetical ancestry cline, shown for illustration. Individuals on the map are colored by ancestry proportion. Blue represents high levels of EF ancestry, and black represents high levels of hunter-gatherer ancestry. This is depicted on an adapted map file from the European Environment Agency[59]. **B** Empirical aDNA ancestry estimates and our best-fitting simulation run from the standard complex landscape (Supplementary Fig. 6) simulation model. The blue line shows the simulated ancestry results for the best-fitting simulation run (learning rate 0.001). Black points show empirical aDNA data points and their estimated ancestry proportions, including the 95% CI (see Supplementary Figs. 3, 16, 17, and Supplementary Data Files 1 and 2 for more detail on the aDNA samples used for fitting). The pink dashed line is a robust linear regression of the empirical aDNA data showing the significant negative cline of the EF ancestry with increasing distance from the farming origin (slope = −7.555e-05, $R^2 = 0.366$, *p*-value < 0.001). The regression was performed using the 'lmrob' function from the R package 'robustbase'[108].

complete replacement of HG ancestry after the farming expansion. This means some level of either vertical or horizontal cultural transmission must have occurred, as a fully demic model, as defined by Ammerman and Cavalli-Sforza[29], would result in complete ancestry replacement. Because different learning rates lead to different clines of EF ancestry (Fig. 4A), we can derive a learning rate that is most consistent with the observed European cline of EF ancestry following the Neolithic expansion[47]. To this end, we estimated EF ancestry in aDNA samples from the Allen Ancient DNA Resource (AADR) v62.0[61]. We selected ancient samples dating between 5000 and 8500 years before present to represent individuals concurrent with, and post the farming expansion, but before the subsequent Steppe expansion[47,62]. A complete metadata file for each of the 1675 analyzed individuals (target, source, and outgroups) can be found in Supplementary Data File 1. EF, Steppe, and WHG ancestry components were estimated for each site using the software qpAdm[62] following Patterson et al.[63] (see "Methods"). Ancestry estimates for all 1531 analyzed individuals can be found in Supplementary Data File 2. For transparency, in Supplementary Data File 2, we have also included columns showing the nested qpAdm[62] model for each individual, i.e., assuming only EF and WHG ancestry sources. The correlation between ancestry estimates from the full and the nested model is high ($r^2 = 0.996$, $p < 0.001$), suggesting that the addition of the Steppe component did not affect the EF ancestry estimation. To increase the robustness and accuracy of our individual-based ancestry modeling for estimating the WHG and EF components, we restricted our analysis to individuals with qpAdm[62] standard errors less than 0.02 and filtered out individuals with greater than 5% Steppe ancestry, leaving a dataset of 618 ancient individuals. Consistent with previous aDNA studies of the Neolithic[47], when plotting the estimated ancestry against direct distance from the farming origin (black points with error bars representing 95% CI, Fig. 5B), a clear trend of decreasing ancestry with increasing distance can be observed (dashed pink line, Fig. 5B; solid black line, Supplementary Fig. 3; slope = −7.554537e-05, $r^2 = 0.366$, *p*-value < 0.001). However, we note four outlier individuals, one individual from Spain and three from Germany, with very low EF ancestry (< 25%), that do not follow this trend. Each of these four outliers was found with paleoanthropological characteristics that place them within either

mixed or hunter-gatherer contexts[64–68]. See Supplementary Note 2 for a detailed description of the archeological context for these outliers.

Next, we used our "simple" square landscape model (Fig. 3A) to estimate a learning parameter that is consistent with the ancestry cline observed from empirical data. We tested 16 learning rates in increments of 0.0003 between 0 and 0.005 per year (Supplementary Fig. 4) and computed the likelihood of each learning rate by taking into account site-specific point estimates and standard error rates of EF ancestry extracted from the qpAdm[62] analysis (see "Methods"). Likelihood values across samples were multiplied, assuming that estimation errors are independent across sites. A quadratic polynomial was then fit to the likelihood values to derive the maximum likelihood estimate and standard error of the learning parameter (Supplementary Fig. 5; details in Supplementary Note 3). This approach leads to an estimated learning rate and 95% CI of 0.000798 [0.000787, 0.000808] per year for 100% within-group mating (Fig. 6; Supplementary Fig. 5A). Using the same approach and the simulated clines in Fig. 4B, we also derived an estimated within-group mating parameter of 0.9835 [0.9833, 0.9837] per year with a learning rate of zero (Supplementary Fig. 5B).

To enhance the realism of the simulated landscape, we subsequently generated data from a more detailed landscape map, incorporating the outline of the European landscape and ensuring that individuals' movement was confined within this outline (Supplementary Fig. 6). We based our simulations on the assumption of farming spread originating in Anatolia and tracked the ancestry cline as a distance from the farming origin following complete expansion of farming (see "Methods"). Finally, we repeated the same inference procedure as before and estimated a learning rate of 0.00101 [0.000998, 0.001023] per year (Fig. 6; Supplementary Fig. 5C). Using the same parameters and model but under an alternative mortality curve[55] (Supplementary Fig. 7; for standard mortality curve, see Supplementary Data File 3; alternate table in Supplementary Data File 4) we saw similar results, with an estimated learning rate of 0.000711 [0.000701, 0.000721] per year (Fig. 6; Supplementary Fig. 5D). Thus, the complex geography and an alternative mortality curve only minimally impact our parameter estimation.

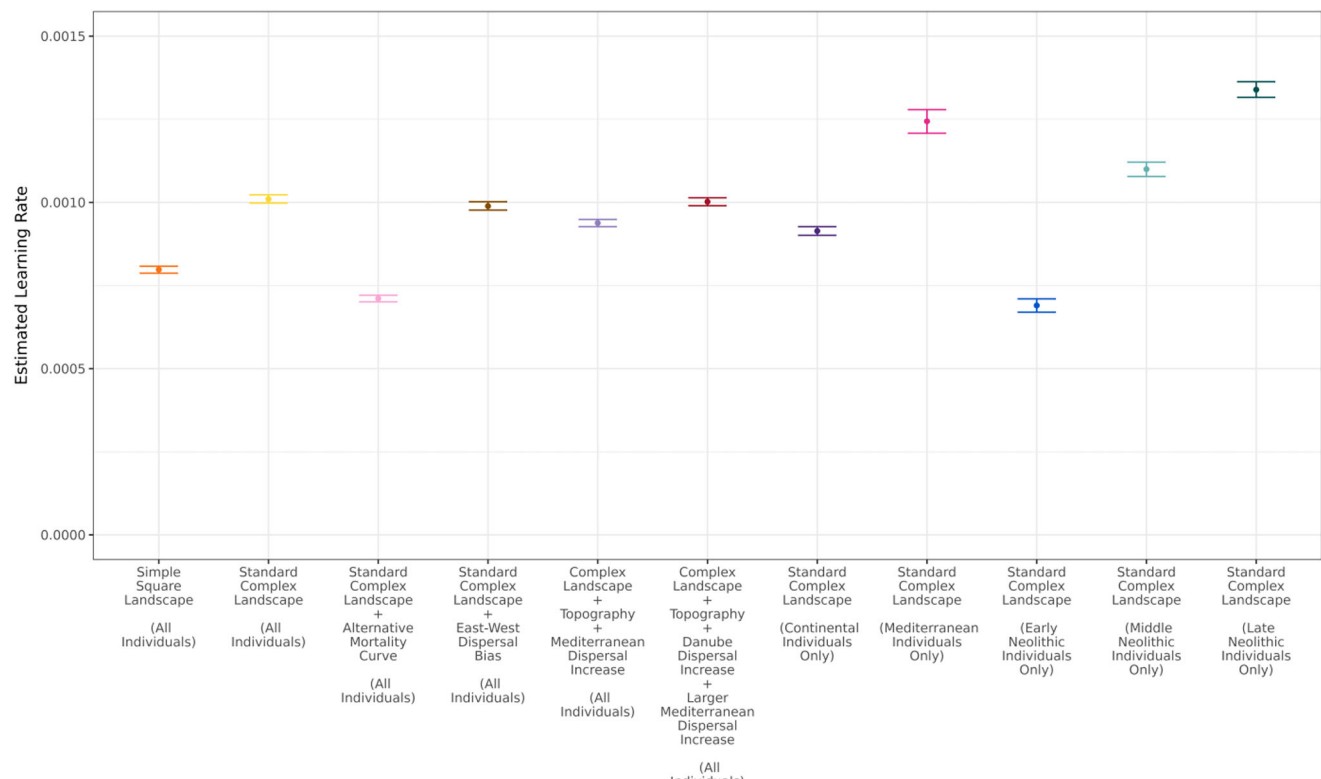

**Fig. 6 | Estimated learning rates for each model and aDNA subset.** Maximum likelihood estimate of the learning parameter and 95% confidence intervals for the six different models tested and the five alternative aDNA subsets. Models and aDNA subsets shown left to right: Orange point = simple landscape map, yellow point = standard complex landscape map (Supplementary Fig. 6), light pink point = standard complex landscape map with alternative mortality curve, brown point = standard complex landscape map with farmer dispersal biased in the East-West Direction, light purple point = complex landscape map with topography and small Mediterranean dispersal increase (Supplementary Fig. 8), dark red point = complex landscape map with topography, Danube/Rhine dispersal increase and larger Mediterranean dispersal increase (Supplementary Fig. 9), dark purple = complex landscape map fitting Continental Route simulated individuals (latitude > 45 degrees) to Continental Route ancient individuals, dark pink point = complex landscape map fitting Mediterranean Route simulated individuals (latitude ≤ 45 degrees) to Mediterranean Route ancient individuals, blue point = complex landscape map fitting to Early Neolithic ancient individuals (sample age > 6500 ybp, $n = 288$ individuals) (Supplementary Fig 10), light teal point = complex landscape map fitting to Middle Neolithic ancient individuals (sample age ≥ 5500 ybp and sample age ≤ 6500 ybp, $n = 206$ individuals) (Supplementary Fig 10), and finally, dark teal point = complex landscape map fitting to Late Neolithic ancient individuals (sample age < 5500 ybp, $n = 288$ individuals) (Supplementary Fig 10). All aDNA datasets had a sample size of $n = 618$ individuals unless otherwise specified.

However, prior studies have indicated that the agricultural expansion was not uniform and might have spread faster along a coastal Mediterranean route compared to a northerly route through the Balkans and into Central Europe[45,56,69,70]. To assess how non-uniform expansion might have influenced our analysis, we conducted simulations under three different non-uniform models. The first model used the same parameters and maps as the uniform complex map expansion, but adjusted the step size parameter to reflect directional differences in dispersal. Specifically, we reduced the step size in the North-South direction to 4 km per year and increased it in the East-West direction to 6 km per year, instead of the uniform step size of 5 km per year. Using the same maximum likelihood approach as before, we found a similar estimated learning rate of 0.001124 [0.00111, 0.001137] per year (Fig. 6; Supplementary Fig. 5E).

To further investigate the effects of heterogeneous dispersal, we applied a model that increased the step size for individuals near the Mediterranean coast, based on speed estimates from Fort and Pérez-Losada[56]. In addition, mountainous regions were defined as barriers to dispersal. This model yielded results that were again in line with previous analyses, with an estimated learning rate of 0.000938 [0.000927, 0.000949] per year (Fig. 6; Supplementary Fig. 5F). To introduce further heterogeneity, we incorporated parameters from Davison et al.[45] which suggest that farming expansions were approximately 5x faster along the Danube-Rhine Corridor and 10x faster along the Mediterranean Coast. The estimated learning rate again remained

consistent with earlier findings, yielding a value of 0.000991 [0.000978, 0.001003] per year (Fig. 6; Supplementary Fig. 5G). All simulation maps with expansion corridors can be found in Supplementary Figs. 8 and 9. Finally, we allowed for regional variation in learning rate by dividing individuals into a northern and a southern group based on a latitude of 45°. Separate learning rates were estimated for each group, revealing a slightly higher rate in the South compared to the North (0.001244 [0.001208, 0.001279] per year, and 0.000914 [0.000901, 0.000927] per year, respectively) (Fig. 6; Supplementary Fig. 5H, I). Nonetheless, our overall conclusion of a cultural learning rate on the order of 0.1% per year remains robust and independent of the expansion route.

After confirming that our learning rate estimate is robust to a variety of landscape and expansion models, we next evaluated how the time period of the ancient individuals used in the fitting process may affect our conclusions. We separated the 618 ancient individuals into bins corresponding to the Early (greater than 6500 ybp), Middle (6500–5500 ybp) and Late (less than 5500 ybp) Neolithic. We repeated our learning rate estimation with each subset of ancient individuals and found consistently low estimates regardless of time period: 0.00069 [0.00067, 0.00071] per year for Early, 0.0011 [0.001078, 0.001121] per year for Middle, and 0.001339 [0.001316, 0.001363] per year for Late Neolithic (See Fig. 6; Supplementary Fig. 5J–L). A linear regression on the ancient individuals from each time period also suggests there is no significant difference in the slope of the ancestry

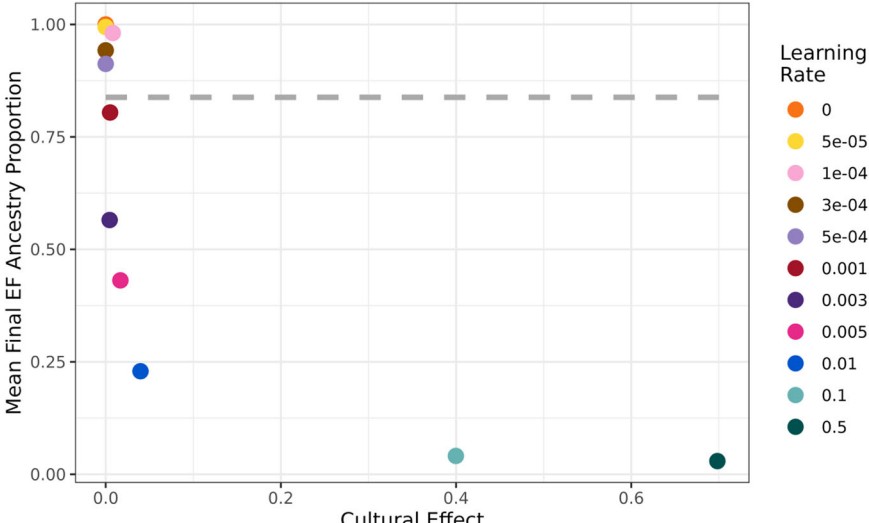

**Fig. 7 | Mean final EF ancestry proportion as a function of the cultural effect.**
Cultural effect is defined as the percent contribution of cultural transmission to the front speed, in relation to the baseline speed of a fully demic model. The figure shows the cultural effect under various learning rate parameters and the final mean EF ancestry proportion upon the conclusion of the simulation when farming has become ubiquitous (colored points). Simulations were run with full within-group mating and various learning rates (right). For comparison to empirical data, the dashed gray line represents the mean proportion of EF ancestry across the whole landscape from our aDNA estimates.

cline between the Early and Middle Neolithic ($p = 0.0636$) and no significant difference when comparing Early to Late ($p = 0.0786$). However, there is a significant drop in the intercept from Early to Late ($p < 0.001$) (Supplementary Fig. 10). Despite this, we do not see a considerable change in our estimated learning rate across time periods, with all returning estimates of approximately 0.001 per year.

Our learning rate estimate assumes complete within-group mating such that only horizontal, but no vertical cultural transmission, takes place. Similarly, our within-group mating estimate assumes no learning and thus no horizontal transmission. Due to the equivalence of horizontal and vertical transmission in their effect on the ancestry cline (Fig. 4), this implies that our estimated learning rate can be considered an upper limit to the true learning rate in a combined model of both horizontal and vertical transmission, and vice versa for the estimated within-group mating rate (see Supplementary Fig. 11).

In sum, despite very different landscape and expansion models, we consistently estimate a yearly learning rate of ~ 0.1% per year (Fig. 6). Given this low rate, it suggests that at the tip of the wave-front, only 1 in 1000 farmers converted 1 HG each year (see "Methods"). This estimate is robust to modeling assumptions and substantially smaller than previously estimated upper limits of the learning rate based on the speed of the farming expansion alone[31,33]. Further, our learning rate of ~ 2.5% per 25-year generation, which we calculate using genomic data, is within the learning (and/or interbreeding) range of 1–8% per generation previously estimated by comparing clines of mitochondrial and Y-chromosome haplogroups to spatial simulations with a generation time of 32 years[51,56].

**Demic expansion without ancestry turnover?**
We have shown that learning rates can vary substantially without affecting front speed (Fig. 3C), but that the observed expansion of EF ancestry is only consistent with very low learning rates and almost complete within-group mating (Figs. 4 and 5B). Here, we more generally evaluate the contribution of cultural transmission to front speed in our simulations, and how this contribution relates to the turnover in ancestry. To this end, we evaluate the cultural effect[31–33]. Cultural effect differs from our learning rate parameter in that cultural effect is a population-level measurement that we use to quantify the overall contribution of learning (or cultural transmission) to the spread of

farming relative to a purely demic model[31–33]. The cultural effect for each set of parameters tested is calculated using the following equation:

$$\text{Cultural effect}(\%) = (\text{front speed} - \text{demic speed})/\text{front speed} \times 100$$

(1)

We assessed the cultural effect for a wide range of learning rates under our simple landscape model (Fig. 7). We assumed complete within-group mating in these simulations but also explored a model of varying within-group mating rates (Supplementary Fig. 12). For our empirically estimated learning rate of approximately 0.001 per year, the cultural effect is close to zero (0.5%). This is similar to the range estimated using mtDNA haplogroup data[34] (0.7–2.3%), both results suggesting that this level of learning does not substantially increase the front speed relative to a pure demic model. Thus, cultural transmission does not appear to have increased the expansion speed during the European Neolithic spread.

However, when the learning rate is further increased from this value, we see that for a large range of learning rates (i.e., $f$ in [0.005, 0.1 per year]) EF ancestry does not spread across Europe despite the cultural effect being low. I.e., despite the model essentially being demic, it does not lead to a turnover in ancestry (Fig. 7). Only when $f$ is larger than 0.1 per year does cultural transmission significantly increase the front speed, indicated by a cultural effect larger than 50%. This is notable because a lack of ancestry turnover during a cultural expansion has been used previously as evidence for a cultural transmission model, e.g., regarding the spread of farming within the Near East[6,50], or the expansion of the Bell Beaker culture between Iberia and Central Europe[71]. However, our results indicate that there is a broad range of cultural transmission rates (i.e., learning rates $f$) that prevent significant genetic ancestry turnover—i.e., geographic ancestry patterns remain largely unchanged during the expansion—while still resulting in a process that is predominantly demic in nature (cultural effect < 50%). The reason for this is that for low values of $f$, the expansion rate varies slowly with $f$, so the spread is overwhelmingly demic[31,34] but, in contrast, the genetic ancestry cline varies rapidly with $f$[51,56], so there is not a genetic ancestry turnover (except near the origin of the spread) e.g., for $f \approx 0.1$ per generation[51]. We note, however, a lack of genetic

turnover for a mainly demic process (cultural effect < 50%) does not happen for the spread of the Neolithic in Europe (in this case, Fig. 5B implies a genetic turnover with $f \approx 0.001$ per year) but may be applicable to other cultural expansions classified as not being demic in nature.

## Discussion

While archeological research has been instrumental in demonstrating the profound changes associated with the Neolithic transition, including the scale and timing of this transformation, it left unresolved questions about the specific mechanisms driving these changes. For example, archeological evidence alone has not decisively revealed the relative contributions of migration versus the diffusion of ideas to cultural shifts[21]. Genetic data complements this by highlighting processes such as mass migration[47,52], but similarly cannot fully quantify the interplay between cultural and demographic factors. Our interdisciplinary approach bridges this gap by using mathematical and simulation-based modeling to integrate insights from both archeology and genetics. This enables us to explore cultural and behavioral mechanisms that are not directly observable in the material record, offering a more nuanced understanding of the Neolithic expansion as a transformative process in human history.

Our study focuses on the extent to which cultural transmission facilitated the widespread adoption of agricultural practices during the Neolithic Revolution in Europe. Using reaction-diffusion modeling and agent-based simulations, we show that the expansion of agriculture across Europe was predominantly driven by a continuous process of farmer migration and demographic replacement, with only a minimal contribution of cultural transmission of agricultural practices. This rate of about 0.1% of farmers converting a hunter-gatherer per year is significantly more precise than inferences from earlier studies based solely on the speed of the expansion, which estimated that up to 10% of farmers each year were converting HGs, assuming a 25-year generation time[31]. Our result of ~ 0.1% per year, or ~ 2.5% per generation, is consistent with the range 1–8% previously estimated from Neolithic mitochondrial haplogroup clines[51,56].

Another metric for quantifying the relative contribution of different mechanisms in the spread of farming is the "cultural effect". Cultural effect is the extent to which cultural transmission impacts the front speed within a demic-cultural framework[31–33]. We estimate that the cultural effect is negligibly small (0.5%), whereas prior studies have reported estimates up to 40%[31,33]. However, these prior estimates fail to consider the distinct ancestry patterns that would result from such high levels of cultural transmission. As such, those estimates were later refined using haplogroup ancestry patterns, leading to the range 0.7%-2.3%[34,51]. This narrowed range aligns more closely with our results, which make use of whole-genome ancestries.

We caution that our model and previous models are based on some assumptions. For example, there is no data on individual dispersal distances in prehistory, but such data are necessary in any spatial model of Neolithic spread. Previous authors[29,31,72] used parent-child birthplace and spousal birthplace distances measured for present preindustrial populations. Those distance distributions do not resemble a normal distribution, as assumed by us, but comparing both approaches is of interest and justified by the lack of prehistoric dispersal distances. Another source of uncertainty in the earliest models[31] was the learning or interbreeding rate, but recently the use of genetics[51,56], and genomics (present paper) has made it possible to estimate this parameter directly from prehistoric data. Similarly, it has been proposed that prehistoric dispersal distances could be measured in the future by identifying parent-child pairs buried in different places[32,34].

Our model also does not consider the possibility of individuals reverting from farming back to hunting and gathering, maintaining that once an individual adopts farming, their descendants are invariably farmers. This assumption, while grounded in ethnographic evidence[73,74] and consistent with previous models[33], may oversimplify the transmission of cultural practices. To address this, we tested our model under alternative conditions, i.e., where offspring could adopt either lifestyle of their two parents, and found similar ancestry clines, indicating a limited impact of this assumption on our results (Supplementary Fig. 13). Overall, our modeling implies that mating between cultural groups must have been remarkably rare (< 3% between-group mating; Supplementary Fig. 5B), in agreement with previous work[51,56].

Notably, aDNA studies from Atlantic France and Scandinavia show no evidence of farmer-associated ancestry in hunter-gatherers who overlapped in time with neighboring Neolithic farmers[75,76], corroborating our finding of low between-group mating. In addition, in some regions, evidence suggests that foragers and farmers coexisted for centuries with minimal gene flow. In northeastern France and Germany, some late Neolithic farmers display unexpectedly high levels of hunter-gatherer ancestry (e.g., Mont-Aimé: 50–63%, Tangermünde: ~ 63%, Blätterhöhle: up to 85%), reflecting prolonged coexistence and limited genetic integration[64,77–79]. Similarly, in the Iron Gates region, hunter-gatherers and early farmers lived side by side for centuries, exhibiting only minimal admixture despite extended contact[80].

These patterns suggest persistent but low levels of asymmetric admixture[64,81] and reveal long-term demographic dynamics, such as the survival of regional hunter-gatherer groups and potential cultural reversions by farmers to hunter-gatherer subsistence strategies. While our model focuses on the initial farming expansion, integrating these long-term dynamics into our model could offer valuable insights into the diversity and variability of local interactions and population structures following the initial farming expansion.

Previous studies suggested a deceleration in the spread of farming in northern latitudes, attributed to several factors[36–38,70]. Climate has been posited as a significant influence, shaping agricultural practices, interactions between distinct groups, and crop performance[70]. In addition, the presence of denser HG populations in these regions has been proposed as a competitive barrier to the northward agricultural spread[36–38]. This deceleration of the farming expansion was linked to higher retention of HG ancestry in northern latitudes, suggesting that the slower rate of expansion allowed for greater integration or coexistence between incoming farmers and indigenous HG communities[70]. In our model's assessment of non-uniform expansion, conducting additional simulations with a decreased step size of farmers indeed demonstrated that a slower expansion speed leads to greater HG ancestry retention in the population following the expansion (Supplementary Fig. 14). This may explain why in northern regions where the farming expansion was presumably slower, e.g., due to the southwest Asian crop package underperforming in the colder and wetter climate, higher levels of HG ancestry are found[70].

Importantly, estimated learning rates remained consistently low (Fig. 6), irrespective of whether we assumed a simple model with uniform dispersal, a complex geographic model with uniform dispersal, a model with faster latitudinal expansion, or models with independent Mediterranean and inland expansion routes. This illustrates the robustness of our findings to spatial dispersal heterogeneity. Interestingly, when estimating learning rates separately for Southern and Northern European individuals, we found that the estimated rate for the South was slightly higher than for the North. This finding contrasts with a recent study based on mitochondrial haplotype clines, which reported uniform learning rates across both Mediterranean and inland routes[56]. Our analysis leveraged a larger sample size and the finer resolution provided by genome-wide ancestry estimates, which likely enabled us to detect subtle differences in learning rates that may be overlooked with mitochondrial data. However, additional analyses are needed to assess the robustness of our findings to other model parameters. In particular, a sensitivity analysis over the ranges of all parameters would be needed to see if our southern and northern estimations for the learning rate overlap or not.

Future studies should aim to integrate autosomal and mitochondrial data into a unified framework to better understand the influence of sex-specific factors on cultural transmission and migration dynamics. They could also delve into the potential for regional variation of cultural transmission rates and how it could be linked to observed local patterns derived from archeological studies. For example, analyzing material evidence at archeological sites along the wavefront could connect our estimated learning rates with indicators of interaction, such as shared tool traditions or shifts in subsistence strategy. However, to assess how learning rates change regionally, two key components are needed. First, local estimates of front speed are necessary, which requires archeological data documenting the regional pace of farming expansion. Second, more extensive sampling of Early Neolithic ancient DNA in specific regions is essential. This would enable the construction of regional ancestry clines that could be used to estimate localized learning rates from simulations under our model. While ancient genomic data are available for many parts of Neolithic Europe, current sample sizes may be insufficient—once filtered by region and time—to reliably establish local ancestry clines. Expanding aDNA datasets in key geographic regions and generating localized front speed estimates will ultimately allow us to bridge the gap between our large-scale model and finer-scale regional inferences.

We also acknowledge that in addition to geographic variability, temporal changes in ancestry—such as the resurgence of WHG ancestry during the Middle and Late Neolithic[62]—may influence estimates of the learning rate. Our primary modeling goal is to capture dynamics during the initial farming expansion; thus, our estimation procedure assumes the analyzed individuals are representative for a time period immediately after the conclusion of the farming expansion. To assess the robustness of this assumption, we divided the ancient DNA dataset into Early (greater than 6500 ybp), Middle (6500–5500 ybp), and Late (less than 5500 ybp) Neolithic subsets (Supplementary Fig. 10), and re-estimated the learning rate for each group independently using our complex landscape model. As expected, the learning rate estimates increase slightly over time (0.0007, 0.0011, and 0.0013 for the Early, Middle, and Late Neolithic, respectively), consistent with the gradual reintegration of WHG ancestry after the expansion phase. While a significant drop in the ancestry intercept is observed in the Late Neolithic ($p < 0.001$), we interpret this as reflecting post-expansion dynamics—such as admixture with remnant WHG groups—that are not central to our model's focus. For this reason, we consider the Early Neolithic estimate (0.0007) to be more representative of the initial expansion process. Importantly, even the higher estimates from later periods (0.0006–0.0014 per year or 1.5%–3.5% per generation of 25 years, Fig. 6) remain two orders of magnitude lower than previous upper estimates based solely on front speed[31], and within the range 1%–8% per generation implied by mitochondrial haplogroup clines[56], supporting our main conclusion that cultural transmission played a minimal role in the spread of farming during the Neolithic.

A potential limitation of our study is that the simulated and empirical ancestry estimates are derived using different approaches: the simulations use ancestry-specific markers to directly compute ancestry, while the empirical data relies on qpAdm[62], which infers ancestry from genome-wide allele frequency differences. While these methods are conceptually distinct, we show that they yield consistent results. Specifically, we identified 53 ancestry-informative markers that are fixed between EF and WHG source individuals and used them to directly estimate Neolithic ancestry across our dataset. These marker-based estimates are strongly correlated with qpAdm[62]-based ancestry estimates ($r = 0.56$, $p < 0.001$; Supplementary Fig. 15), and the resulting cline of farming ancestry yields a nearly identical learning rate estimate (0.00103 per year) to that obtained using qpAdm[62] (0.00798 per year). This convergence across methods suggests that our main results are robust to how ancestry is defined and quantified. We provide the

ancestry marker set and derived ancestry values for all individuals in the Supplementary Data File 5.

We want to emphasize that the learning rate is an abstract parameter in our model that is ignorant about the underlying mechanism of "learning" itself. Cultural transmission may have been a purely neutral, voluntary exchange of knowledge between groups. However, the spread of farming during the Neolithic may have involved a mix of voluntary adoption and coercion. Competition for resources, social hierarchies, and economic dependencies could have pressured hunter-gatherers to adopt agriculture. Evidence of Neolithic conflict[82,83] further suggests that some groups may have been compelled to abandon traditional ways, indicating that the transition to farming may not have always been by choice. While our model estimates a rate of conversion from HGs to farming, it does not capture the specific cultural mechanisms at play, particularly whether this shift occurred willingly or under coercion.

Our study primarily examines the agricultural expansion in Europe. In the global context, Neolithic expansions exhibited considerable variability in their replacement of native hunter-gatherer ancestry, challenging the notion that agricultural spread always was accompanied by significant genetic turnover[84]. In fact, the agricultural expansion within southwest Asia, before its spread into Europe, is not associated with ancestry turnover, which has led to the conclusion that this initial expansion was propelled more by the dissemination of ideas and farming technology than by the movement of people[6]. However, our simulations suggest complexities beyond conventional classifications into cultural vs. demic diffusion based on ancestry turnover, revealing scenarios where agricultural expansion—despite being predominantly demic—does not lead to the anticipated widespread dissemination of farmer ancestry. In a primarily demic model, large-scale ancestry turnover is expected as populations with a novel ancestry component migrate into new areas and displace local populations[29] (e.g., EF ancestry replacing local WHG ancestry during the Neolithic expansion in Europe). In contrast, a primarily cultural diffusion model predicts that original ancestry patterns remain intact as culture spreads through learning or imitation rather than population displacement. In ancient DNA research, expansions are often classified as demic when substantial ancestry turnover occurs, or as cultural when ancestry patterns are preserved. Our findings challenge this binary framework, introducing a third, intermediate scenario: a predominantly demic expansion that occurs without significant ancestry turnover.

To this end, we use the cultural effect metric[31–33], which quantifies the role of cultural transmission in the expansion process. This metric measures the unique contribution of cultural transmission to the overall expansion speed of the farming cultural package relative to a purely demic model driven by population movement alone. We find that, under a wide range of low but sufficient learning rates, the cultural effect is minimal, suggesting a predominantly demic expansion speed. However, under these conditions, the ancestry patterns resemble those of a cultural diffusion model, where original ancestry patterns are preserved. This can result in an expansion being misclassified as cultural due to the absence of ancestry turnover, despite its predominantly demic nature. This third scenario is also seen in previous simulations of haplogroup clines, where some learning rates (e.g., about 10% per generation) lead to waves of advance that are mainly cultural genetically (no major genetic replacement except near the origin of the spread)[51], but mainly demic archeologically (no significant increase in the spread rate relative to the purely demic case)[31].

Thus, our results demonstrate that initial ancestry patterns can be preserved even in a predominantly demic expansion model, underscoring the limitations of traditional binary classifications based on ancestry turnover. Consequently, our findings prompt a reevaluation of other significant cultural shifts, such as the spread of the Bell Beaker

phenomenon around 4800–4000 years ago. Similar to the farming expansion in southwest Asia, the Bell Beaker cultural phenomenon initially spread across parts of Europe with little impact on genetic ancestry−individuals from Iberia and Central Europe show limited genetic affinity to one another. However, the later expansion of the Beaker complex to Britain led to a massive replacement of nearly all of Britain's gene pool[71]. Our modeling suggests that a wide range of cultural transmission rates can prevent a turnover of genetic ancestry− but are small enough to lead to a mainly demic diffusion of culture (i.e., with a cultural effect < 50%; Fig. 7). Thus, even the initial expansion of the Beaker complex is, in principle, consistent with a predominantly demic mechanism. In sum, our insights call for a reconsideration of how we understand cultural spreads and their impact on genetic ancestry patterns, potentially revising narratives around major prehistoric and historic cultural expansions.

In conclusion, our research offers an estimate of the contribution of cultural transmission to the Neolithic agricultural expansion in Europe, suggesting that the front speed was not significantly increased by cultural transmission. Through the application of diverse modeling approaches−including reaction-diffusion and agent-based models, alongside simulations of complex geography and varied European expansion axes−we estimate a cultural effect of about 0.5%, which improves previous estimates that did not consider the whole genome but haplogroup frequencies and led to the range 0.7–2.3% for the cultural effect[34,51]. Our analysis, incorporating both front speed and empirical ancestry data from ancient DNA, suggests that only a minute fraction of farmers might have been actively involved in cultural interactions with HGs. More generally, our modeling stresses the possibility of demic expansion without complete ancestry turnover, offering a refined perspective on historical expansions like the Bell Beaker phenomenon in Europe. Our interdisciplinary approach allows us to more accurately dissect the dynamics of the Neolithic expansion, emphasizing the need for integrated methodologies and modeling approaches in historical analysis.

## Methods

### One-dimensional equation model

We set up a reaction-diffusion model, based on partial differential equations, which incorporates spatial diffusion of individuals, cultural transmission of farming practices, and genetic ancestry at a single locus. Under this model, we assumed a 3000 km long one-dimensional (1D) habitat along which the density of three types of individuals is modeled: cultural farmers with EF ancestry, cultural farmers with WHG ancestry, and cultural hunter-gatherers (HGs) with WHG ancestry. As an initial condition, we assume that the farmers with EF ancestry are constrained to ~ 200 km at one end of the range, and the HG group is constrained to the remaining 2800 km of the range, with about a twenty-fold difference in population density[29]. Our mathematical model assumes a diffusion component, a logistic population growth component, and a cultural transmission component (see formulas in Fig. 1A; for a comprehensive description of the symbols and components of the equations in Fig. 1A, see Supplementary Note 1). The cultural transmission model was originally developed by Fort (2012)[31], and accounting for haplogroup frequency at a single genetic marker by Isern et al.[51]. The cultural transmission component of our model decreases the HG density and to the same extent increases the "farmers with WHG ancestry" density as a function of two parameters, the learning rate $f$ and the bias parameter γ. The learning rate $f$ is a product of the number of teachers an HG contacts during their lifetime, and the probability that an HG converts to farming after contact with a farmer. The bias parameter γ modifies the preference of HGs to learn from a farmer rather than from another HG[31]. The diffusion constant $D$ was chosen such that the expansion speed of farming under purely demic parameters is the empirically observed 1 km per year, given a growth rate α of 3% per year[29,85]. We assumed that both farmers

and HGs have the same diffusion constant and growth rate. In accordance with the initial condition, we assumed that the carrying capacity of farmers was twenty times larger than that of HGs[29]. We set up the partial differential equations model in Mathematica 13.3[86] and numerically solved it over a time span of 100 generations (i.e., ~ 2500 years when assuming a generation time of 25 years), which was shortly after all HGs had been acculturated, using the Mathematica function *NDSolve*. The boundary conditions of the numerical solution were set up such that at positions 0 km and 3000 km (the left and right boundary of the 1D landscape), the first derivative of the three population densities with respect to the space coordinate was set to zero across all time points.

### Two-dimensional agent-based simulation model

Using the insights from the basic 1D reaction-diffusion model, we then developed a more realistic two-dimensional (2D) agent-based model, written in the Eidos language[87] designed for the SLiM simulation framework[57] (v4.0). In these simulations, each individual exists on a simulated 2D landscape, has a diploid genome, and a cultural identity (farmer or HG). We adopt annual timesteps and model overlapping generations, meaning that individuals persist across multiple yearly intervals until they pass away. Upon initialization of the simulation (year one), the ancestry assigned to the individuals' chromosomes is based on their initial cultural identity (i.e., farmers have pure EF ancestry and HGs have pure WHG ancestry). In each subsequent year, the ancestry proportion of new offspring is determined via sexual recombination of the individual's parental genomes. The cultural identity of each individual is independent of their genetic ancestry and can change from HG to farmer through means of cultural exchange as the simulation progresses.

### Genome simulation

To determine the ancestry proportion of each simulated individual, one neutral marker mutation is initialized every megabase along the 247 Mb simulated chromosome (human chromosome 1) of all farmers. Therefore, since we consider diploid individuals, each individual has a simulated genome of 247 * 2 = 494 markers. All initial farmers have all 494 markers, which are set to equal "1". These markers "tag" EF ancestry along the chromosome of each individual. All initial hunter-gatherers have all markers equal to "0". Accordingly, the proportion of EF ancestry is calculated in all individuals subsequent to year zero by counting the number of markers equal to one and dividing by the total number of marker locations (i.e., by 494). The genomes of new individuals are generated via simulated sexual recombination of the two parents as part of the built-in SLiM reproduction function[57], assuming a recombination rate of 1 cM/Mb. This function takes the two parental genomes and simulates crossing over between the two gametes at the specified recombination rate[57] and is explained in more detail in Supplementary Note 4.

### Mortality and competition

Mortality in our agent-based model is governed by two main factors: age-dependent mortality and density-dependent competition. Our age-dependent mortality curve is based on osteological age-at-death data[42,55]. We generated an age-specific 'equilibrium' mortality rate that is applied to both farmers and hunter-gatherers, which corresponds to the age-at-death distribution observed in the osteological data[42,55] (Supplementary Data File 3, 4; Supplementary Note 5; Supplementary Fig. 7), i.e., in equilibrium, this mortality rate leads to the age-at-death distribution observed in the osteological data[42,55]. Using these 'equilibrium' mortality rates for each age, we computed an equilibrium fertility rate of 0.1 for each mature individual over the age of 11. I.e., this fertility rate compensates for yearly deaths, which allows the population size to stay constant over time in regions where it has reached the carrying capacity. The ages of individuals at the initiation of the

simulation were sampled from the equilibrium age distribution. Our primary age-specific mortality rates (Supplementary Data File 3; Supplementary Fig. 7) correspond to the age-at-death distribution observed by Papathanasiou[42], but to assess if different mortality curves impact our results, we sought out an alternative mortality curve based on Eshed et al.[55] (Supplementary Fig. 7; Supplementary Data File 4). We ran simulations with this alternative curve to evaluate the robustness of our inference to the assumed mortality rates.

To model density-dependent competition—based on the idea that increased density leads to higher health-related mortality costs[14,88,89]—we implemented a density-dependent scaling of the 'equilibrium' mortality curve. In our agent-based model, density-dependent competition is determined by the total local population density, regardless of cultural identity, and this is compared to the carrying capacity specific to each group. This approach is consistent with our reaction diffusion model, in which both farmers and hunter-gatherers contribute equally to population pressure. To this end, all age-dependent mortality rates were scaled down in regions with population density below $K$, and scaled up in regions with population density above $K$. The local population pressure is calculated as the ratio of the number of neighbors within a radius of 30 km surrounding each individual, relative to the expected local $K$ (or the expected number of individuals in this area calculated using the carrying capacity). We derive each individual's 'experienced' mortality rate by multiplying its 'equilibrium' mortality rate by the local population pressure, which allows the population to grow until it hits the carrying capacity $K$. Each cultural group has a unique carrying capacity— $K_F$ for farmers, and $K_{HG}$ for HGs, which allows farming to support higher population densities.

We set the $K_{HG}$ to 0.064 individuals per square kilometer. This is following the modeling approach by Currat and Excoffier[54], who selected this value based on two previous papers exploring palaeo-demographics[90,91]. For the farmer carrying capacity ($K_F$), we used a value of 1.28 individuals per square km. We calculated this number based on estimates by a model by Ammerman and Cavalli-Sforza (1984). In this model, the $K_F$ is expected to be 20 times larger than $K_{HG}$[29].

We assume a constant fertility rate of 0.1 per mature individual per year (see *Reproduction and within-group mating* section below). The age-specific mortality curve is scaled by population density such that, at carrying capacity, the number of deaths balances the number of births, maintaining a stable population size. At lower densities, mortality decreases, allowing the population to grow toward the carrying capacity. In our model, density-dependent competition takes into account all individuals, independent of cultural identity. As a result, HG populations gradually decline over time as the population pressure from close by farming populations becomes too large for their smaller $K_{HG}$. This is based on the idea that farmers compete with HGs for land and resources, which was also incorporated in previous mathematical models of the farming expansion[36-38]. To reduce the significant run-time and memory requirements of the agent-based simulation, we downscaled the number of individuals by a factor of five. Thus, we divided all carrying capacities mentioned above, as well as the starting population density, by 5 for all agent-based simulation runs. This results in an average runtime of 4-5 days on a computing cluster using 20 Gb or less of RAM per individual simulation.

### Reproduction and within-group mating
Each year, each mature individual randomly chooses a mate within its cultural group (within-group mating) or independent of cultural group (between-group mating), governed by a within-group mating probability parameter set before runtime. In addition, mate selection only occurs between mature individuals (age > 11) within a radius of 10 km. When a suitable mate is found, then the number of offspring is sampled from a Poisson distribution with rate parameter $\lambda = 0.1$ using SLiM's[57] built-in pseudorandom number generator. This produces a

non-negative integer number of offspring per mating event, drawn according to the Poisson probability mass function. This rate is calculated based on the age-specific mortality curve, such that population size is maintained when the population is at carrying capacity (i.e., deaths are compensated by births; see above). If no suitable mate is found (e.g., if an HG is set to mate within-group, but is not surrounded by any other HG within a radius of 10 km), then no offspring is produced by that individual in that year, except if it is selected as a mate by another individual. The location of offspring is chosen as one of the parents' locations, simulating maternal care.

If a HG and a farmer reproduce, their offspring is assigned to be a farmer. This assumption is supported by cultural observations of the assimilation of individuals into farming groups[73,74], and the lack of farming ancestry in the genomes of HGs in ancient DNA studies[75,76]. However, we have validated the robustness to this assumption with a model where offspring of farmer-HG matings randomly choose their subsistence strategy (Supplementary Fig. 13).

### Learning and cultural transmission
Each year, every HG individual has the opportunity to learn agriculture and transition to being a farmer. The individual probability of transitioning to farming takes into account the local proportion of farmers surrounding the HG individual in a radius of 10 km (i.e., the more farmers there are in the vicinity, the higher the probability of learning; when no farmers are nearby, learning will not take place). Thus, for each HG in the simulation at a given year, the transitioning to farming is a random event with probability calculated as the product of the local proportion of farmers and a simulation-wide learning rate $f$ (assuming $\gamma = 1$, as justified in the first section of *Results*). This model is consistent with the cultural transmission model of Fort (2012)[31]. The product of the fixed number of teachers encountered each year and the probability of learning farming from a single farmer contact, in their model, equals our learning rate $f$. It can be interpreted as the 'number of HGs converted per farmer' when the local proportion of farmers is close to zero (i.e., at the tip of the wave-front of the farming expansion), or alternatively as the proportion of HGs that gets converted to farming when farming is the dominant local subsistence strategy. Further note that our rates are yearly rates, whereas Fort (2012) models cultural transmission on a per-generation basis[31].

### Simulation landscape
The landscape map in our model is represented by a black-and-white image file that is utilized by SLiM[57]. The black pixels represent inhabitable land, and the white represents uninhabitable regions (i.e., oceans). Within our 2D spatial model, each individual has a positive $X$ and $Y$ coordinate that represents their location on the inhabitable landscape. The 2D position of the individuals is compared to the location on the landscape map, allowing us to achieve geographic boundaries for expansion.

We used this feature to run simulations on both a simple all-black pixel plane where all regions of a 3700 × 3700 km landscape are inhabitable, as well as a map outlining the geography of Europe. The latter map was adapted from the European Environmental Agency's (EEA) Elevation map of Europe[58], available at the following URL: (https://www.eea.europa.eu/ds_resolveuid/558D91E1-3DB0-4639-9F70-2012CC4453A5). Using Adobe Photoshop (CC 2022)[92], we made the landscape image file dichromatic by selecting the land areas and filling them with black pixels, whereas we changed the remaining pixels (i.e., oceans) to white. To facilitate water travel via ocean, we added water crossings to the map using the directional maps from Fort (2015) as reference[32] (Supplementary Fig 6). The same process was used for the simulations that modeled multiple distinct routes. However, in these versions of the landscape map, additional maps were overlaid on top of the base map that corresponded to the coastal route and the Danube/Rhine routes. These overlays (Supplementary

Figs. 8 and 9) allowed us to check if individuals were located on a route that facilitated a greater dispersal potential (see below for more detail on the movement of individuals).

## Movement

Each year, for every individual, a distance in the $X$-direction (East-West) and a distance in the $Y$-direction (North-South) is drawn from normal distributions with standard deviations $\sigma_X$ and $\sigma_Y$, which are then added to the individual's previous $X$, $Y$ position. These standard deviations allow us to control the distance individuals can travel within a year. Whenever the coordinates of the newly sampled position are out of bounds (i.e., off the map or on a white pixel of the landscape), this position is rejected, and a new position is sampled until a legitimate position is found.

For most simulations, we assume isotropy, i.e., $\sigma_X$ and $\sigma_Y$ are the same. However, we explore a model with $\sigma_X > \sigma_Y$ to investigate the impact of a favored East-West movement direction, e.g., reflecting the faster expansion along the Mediterranean route[45,56,69]. In addition, we explored another model of biased dispersal, where we added a coastal route to the landscape (Supplementary Fig. 8). In this model, when individuals were located along ocean waterways or along the Mediterranean coast, they had a slightly increased dispersal distance based on coastal expansion speed calculations from Fort & Pérez-Losada[56]. Individuals only had this increased dispersal associated with the route if they continued traveling along the route. The likelihood of them continuing along the route or leaving was randomly sampled at equal probability. If an individual left and returned to the greater continental area, they would revert to the standard continental $\sigma_X$ and $\sigma_Y$. In these more complex landscape simulations, we also incorporated mountainous regions as barriers to dispersal[45] (Supplementary Fig. 8). White pixels were added to the map based on regional topography[58]. The combination of the physical barriers and the waterways enforced distinct routes of expansion across the landscape.

Finally, we developed another biased dispersal model, again with mountainous barriers[58] as well as larger dispersal distances along both the Mediterranean Coast and the Danube-Rhine Corridor, relative to a smaller dispersal distance in the greater continental area (Supplementary Fig. 9). For these simulations, we kept both $\sigma_X$ and $\sigma_Y$ the same (3 km), but we increased $\sigma_X$ and $\sigma_Y$ for individuals along the Mediterranean Coast (30 km) and the Danube-Rhine Corridor (15 km). These values are based on work by Davison et al., (2006), who found 10x faster dispersal along the Mediterranean Coast and 5x faster dispersal along the Danube-Rhine Corridor[45]. To accommodate for the increased dispersal along the waterways while still maintaining an overall front speed of ~1 km per year, we needed to reduce the base $\sigma_X$ and $\sigma_Y$ from the 5 km we used in other models to 3 km.

## Front speed calculation

Our simple square landscape was used to investigate how front speed, i.e., the speed with which farming expands across the $X$-axis of the square, depends on step size and learning rate parameters. Upon initialization, we filled up the landscape with individuals in accordance with the HG carrying capacity ($K_{HG}$). Individuals with $X$-coordinates between 0 and 74 km were designated as original farmers, while the rest of the individuals on the map were initialized as HGs. We then tracked the expansion of farming each year by calculating the proportion of farmers within twenty 185 km wide bins on the $X$-axis. The extent of farming expansion was defined as the $X$ coordinate of the bin at which the farmer proportion had just reached 50%. The speed of farming expansion was then calculated by the slope of a linear regression of the extent of farming against time.

## Empirical ancestry estimates

We used qpAdm[62] from the ADMIXTOOLS2[93] package with parameters 'allsnps = TRUE' and 'afprod = TRUE' to obtain empirical estimations of

EF, Steppe, and WHG ancestry contributions to Neolithic aDNA samples from Europe (Allen Ancient DNA Resource (AADR) v62.0)[61]. Including Steppe as a source population allowed us to filter individuals from our dataset that contained Steppe ancestry or possible modern contamination. i.e., by filtering out these individuals, we were able to capture the population dynamics unique to the farming expansion. Following Patterson et al.[63], we formed the WHG source population by grouping 18 ancient hunter-gatherer individuals[53,64–66,68,79,94–96], 19 individuals for the Steppe population[7,53,62,63,97,98], and formed the EF population from 21 individuals from the Balkans and Aegean[7,53,62,99]. We supplied the following four groups as fixed right-group populations in qpAdm[62]: Russia_Afanasievo[97], WHGB[53,100,101], Turkey_N[7,102], and OldAfrica[103,104]. We restricted our analysis to individuals dating between 5000 and 8500 ybp and filtered any related individuals or those assigned as contaminated in the AADR metadata, resulting in 1675 individuals. We classified qpAdm[62] models as plausible with p-value ≥ 0.01 and admixture weights between 0-1. To increase the confidence in our EF and WHG ancestry estimates, we restricted our qpAdm[62] analysis to individuals with admixture weight standard errors < 0.022, following Patterson et al.[63], and individuals with less than 5% Steppe ancestry, resulting in a final set of 618 individuals. Ancestry estimates of the 618 ancient individuals, plotted geographically through time, can be found in Supplementary Fig. 16). In our qpAdm[62] analyses, we used Balkan_N as the proxy source population for Early Farmer (EF) ancestry, reflecting previous recommendations[63]. While not the earliest point of Neolithic expansion, Balkan_N serves as a geographically and genetically intermediate farming population that provides broad utility as a proxy source across European Neolithic contexts. Notably, Balkan_N individuals were selected to possess minimal hunter-gatherer admixture[63], which improves qpAdm[62] model fit typically yielding higher p-values when modeling admixed Neolithic populations in Europe. To further stabilize ancestry estimates, we included Turkey_N and a set of deep outgroups in the right populations (see full list in Supplementary Data File 1). As a consequence of using Balkan_N, which has minimal WHG ancestry, some ancient samples—especially those from southeastern Europe or regions east of the Balkans—can yield slightly negative WHG ancestry estimates, while samples from northern and northeastern Europe (particularly those younger than 7000 ybp) may show negative EF ancestry values. In sum, of the samples that passed the plausibility criteria of p-value ≥ 0.01, Steppe ancestry [0, 0.05] and EF ancestry S.E. ≤ 0.022, only ~3.8% and ~3.3% were filtered due to negative EF and WHG ancestry estimates, respectively. Under these conditions, we followed standard qpAdm[62] practices and excluded individuals with negative ancestry proportions from downstream analyses. This filtering step ensures that only biologically interpretable ancestry estimates are included in our regressions and simulation-based inference approach and improves model robustness across diverse European Neolithic samples.

Finally, we acknowledge that reliance on a single aDNA database (in this case, AADR v62.0)[61] has the potential to introduce bias. However, the AADR[61] remains one of the most comprehensive, well-curated, and widely used sources of publicly available ancient DNA data. Moreover, recent work by Schmid et al.[105] comparing the AADR[61] and the Poseidon database—another large, community-maintained ancient DNA resource—demonstrates substantial overlap between the two, particularly for European samples. This concordance across independently maintained datasets suggests that our sample set reflects the broader landscape of available ancient genomes relevant to our study.

We visualized broad genetic ancestry patterns of the analyzed individuals through principal component analysis (PCA) (Supplementary Fig. 17) using the smartpca v.8.0 software[106] with parameters "numoutevec: 10", "numoutlieriter: 0", "hiprecision: YES", "shrinkmode: YES" and "lsqproject: YES", projecting the ancient individuals onto a set of 69 modern Eurasian individuals[49]. A complete metadata table for each of the ancient individuals can be found in

Supplementary Data File 1 and ancestry estimates for all ancient individuals are located in Supplementary Data File 2.

## Fitting simulations to empirical ancestry data

To fit parameters of our simulations to the qpAdm[62] results, we compared the empirical EF ancestry estimates to the simulated ancestry clines (i.e., the EF ancestry proportion as a function of the distance to the farming origin) under different learning and within-group mating rates. Thus, for each aDNA sample in our study, we derived an observed EF ancestry proportion using the empirical estimates from qpAdm[62], and we derived an expected EF ancestry proportion, assuming the same distance to the farming origin, from our agent-based simulations. To this end, we set the location of the farming origin on our simulation landscape just East of Ankara, Turkey, and took the straight-line distance of each ancient sample from this point. We then used our simulation data to interpolate an expected EF ancestry proportion with the same straight-line distance as each aDNA data point. We took into account uncertainty in the estimation of the empirical ancestry proportion by assuming a normally distributed measurement error with standard deviation set to the sample-specific standard error output of qpAdm[62]. For each simulated learning rate, the observed and expected EF ancestry proportions were used to calculate a log-likelihood of each ancient sample, which was then summed up across all samples. The simulation-wide log-likelihood was then plotted against each respective learning rate. We fit a quadratic function to the log-likelihood values to derive a maximum likelihood estimate of the learning rate and a standard error as the square root of the negative inverse of the second derivative (Supplementary Fig. 5; see Supplementary Note 3 for full derivation of the statistical approach).

It has been shown that the Neolithic expansion was not uniform and featured Northern (Continental), and Southern (Mediterranean) routes[45,56,69,70]. To check for heterogeneity in the estimated learning rate, we fit simulated individuals along the Southern and Northern parts of the map to ancient individuals from the same regions. We used a latitude of 45 degrees as an approximate cutoff, converting our X, Y coordinate-based simulation plane to approximate latitude, longitude values to allow for comparison with aDNA data.

We also assessed if temporal ancestry variation affected our fitting results. To this end, we binned the ancient individuals by age, corresponding to three periods of the Neolithic: Early (sample age greater than 6500 ybp), Middle (sample age greater than or equal to 5500 ybp and sample age less than or equal to 6500 ybp), and Late (sample age less than than 5500 ybp). This resulted in 288 Early individuals, 206 Middle individuals, and 124 Late individuals.

To assess the robustness of our ancestry estimates to the choice of source population, we repeated the qpAdm[62] analysis using Anatolian early farmers ($n = 26$) instead of Balkan early farmers ($n = 21$) as the proxy for Neolithic ancestry. We found that the resulting ancestry proportions produced similar spatial clines and learning rate estimates (see Supplementary Fig. 18). This suggests that our inferences are not sensitive to the specific choice of early farmer reference group, and supports the robustness of the qpAdm[62]-based ancestry estimates. Finally, note that to derive the expected EF ancestry proportion from our simulations as a function of distance, we had to bin simulated individuals into different distance bins to compute their average ancestry proportion as a function of distance. For the simple square landscape, we binned all individuals into 20 equal-sized bins along the X-axis, since farming progresses along the X-axis from left to right in this model. On the simulated European map, where farming starts in Anatolia and then expands into Europe, we created bins based on the straight-line distance of each individual from the origin in the southeast corner of the map, representing the farming origin. We then sampled 10,000 individuals randomly throughout the map and binned them according to this distance into 100 bins with 52.4 km width, and for each bin we computed the average EF ancestry.

## Reporting summary

Further information on research design is available in the Nature Portfolio Reporting Summary linked to this article.

## Data availability

All genetic data used for model fitting were obtained from the publicly available Allen Ancient DNA Resource (AADR), version 62.0[61,107]. No new empirical data were generated in this study. The full simulation code and instructions for reproducing all results and figures in this paper are provided at the following GitHub repository: https://doi.org/10.5281/zenodo.15924221. As all simulated data can be regenerated from the provided code, the full set of simulation outputs is not included.

## Code availability

The Edios code for the SLiM agent-based model used in this study, a file explaining the code for the SLiM agent-based model— as well as landscape files for the model, the R data analysis scripts, and the Mathematica code for the 1D model, can all be found in the following repository: https://github.com/TroyLaPolice/European-Neolithic-Expansion GitHub Code Permalink https://doi.org/10.5281/zenodo.15924221 Code Under MIT License (see GitHub for Details): *Copyright (c) 2024 Troy M. LaPolice, Matthew P. Williams & Christian D. Huber.*

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

## Acknowledgements

Research reported in this publication was supported by the National Institutes of Health Grant R35GM146886 (C.D.H.), by the Pennsylvania State University (C.D.H., T.M.L., M.P.W.), and by the National Institutes of Health T32 GM102057 (T.M.L.). The content is solely the responsibility of the authors and does not necessarily represent the official views of the National Institutes of Health.

## Author contributions

T.M.L. and C.D.H. conceptualized and designed the study, T.M.L. developed and analyzed the 2D agent-based simulations, C.D.H. developed the 1D mathematical model, M.P.W. conducted aDNA ancestry estimations, T.M.L. and C.D.H. prepared the manuscript, T.M.L., C.D.H., and M.P.W. revised and edited the manuscript, C.D.H. supervised the study. All authors read and approved the final manuscript.

## Competing interests

The authors declare no competing interests.
