## [Transparent Peer Review file · Nature Communications]

Modeling the European Neolithic expansion suggests predominant within-group mating and limited cultural transmission

Corresponding Author: Professor Christian Huber

Version 0:

Reviewer comments:

Reviewer #1

(Remarks to the Author)

The authors assess the mechanisms of agricultural expansion in Neolithic Western Eurasia. The authors suggest that presence of hunter-gatherer ancestry in modern populations indicates a certain degree of cultural transmission and non-assortative mating. The authors use modeling and ancient DNA data to quantify the contribution of assortative mating and learning in the Neolithic expansion. They find limited cultural transmission and mainly assortative within-group mating.

I find the manuscript very interesting and think the modeling that was performed is of high quality. However, I think that the manuscript does not significantly deepen our understanding of the Neolithic transition. The authors find a mechanism of primarily demic expansion with limited learning, and primarily within-group mating, which have been shown several times already using different lines of evidence.

An important aspect of the Neolithic that is not being discussed here is the change through time in the 3000-year time transect that the authors are modeling. Hunter-gatherer ancestry proportion in Europe goes up from the Early to Middle Neolithic, with approximately 5% HG ancestry in early vs ~20-30% in the middle Neolithic.

Learning is being discussed throughout the manuscript but the term is vague and assumes a process that is neutral (L 58). There is a possibility that that the process of learning was forced.

The authors suggest that “primarily demic expansion does not necessarily lead to a major ancestry turnover, suggesting that a naive interpretation of ancient ancestry patterns does not always reflect underlying behavioral mechanisms.” (L 104-106). However, it is unclear to me how exactly their modeling challenges this assumption.

Please see specific comments and questions below:

- The abstract states that the primary mechanism of Neolithic expansion is debated. However, it has been shown many times using genetic data that the primary mechanism was demic diffusion.
- L. 37 in the paragraph the authors say that the first people to describe the spread of agriculture were Ammerman and Cavalli-Sforza. There is a rich history of this research in archaeology that is not being discussed.
- L. 97: the authors say here that they derived ancestry estimates from 911 early Neolithic farmers. However, L288: it is said that they include all Neolithic populations until the Steppe ancestry arrival, which would also include the middle and late Neolithic.
- L. 103 “... less than 0.1% of farmers converting an HG to farming annually”: phrase is confusing. Please rephrase the sentence.

- L 129 -131. The Neolithic ancestry proportion varies through time and space. Both have to be taken into account in the modeling.
- Figure 1B: The different learning rates seem very similar to each other, i.e. there is a very small difference in Anatolian Ancestry between learning rate 1 and 0.001. Could you explain why there is so little variation?
- Could you please provide more details on what the “learning rate” means? Does learning rate of 0.001 indicate demic diffusion vs learning rate of 1 cultural diffusion?
- Figure 2. Please remove the icon of the tie, since it is not accurate for the time period and has inherent assumptions about the populations under study.
- Why only learning rates between 0 and 0.005 were tested (L 245)? L162: upper limit of learning rate of 0.1 was found under unbiased learning.
- The authors should include the IDs of all individuals included in their study. I would also advise against binning samples from the same geographic location (L 291) due to potential ancestry changes through time, and to also to include a temporal component into the modeling.
- The authors should include the qpAdm results (L 293) as a supplementary data file.
- L. 351: the learning rate interpretation states: “only 1 in 1000 farmers converted a hunter-gatherer each year”. Naively, I would have thought that the learning rate reflects how many HGs are converted, since more than one farmer can convert one HG.
- L. 361: What is the difference between “cultural effect” and “learning”, since the two are related and somewhat try to capture the same process of learning the cultural practice of farming?
- L 390-393: “However, our results indicate that there is a wide range of cultural transmission rates that prevent a turnover of genetic ancestry, but are small enough to lead to a predominantly demic diffusion of culture (cultural effect < 50%), i.e. geographic ancestry patterns do not necessarily change during a demic expansion of culture.”
 - o Could you please explain what you mean here, since your modeling shows that with cultural effect of more than 50% there is almost no Anatolian ancestry, and with the cultural effect of 0.05-0.1 the mean Anatolian ancestry looks like 0.25, which indicates an ancestry change.
- General comment. Could you please change the color of the points in figures? Some colors are too similar to easily differentiate visually.

(Remarks on code availability)

Reviewer #2

(Remarks to the Author)

In this study, LaPolice and colleagues present a mathematical model to quantify the roles of cultural diffusion and mating during Neolithic expansion. The originality of their work is to combine agent-based simulations and ancient DNA analyses. The first part of their work examines the influence of cultural transmission on the dispersal of agricultural ancestors with a unidimensional model. From this simplified model, they suggest a low level of cultural transmission, i.e. 1 farmer per 1000 farmers converts one HG per year. Then, they offer different simulations to test more complex scenarios. They estimate the effect of a parameter such as front speed on the rate of cultural transmission and found no effect in their simulations. Finally, they test the effect of assortative mating and learning on the proportion of Anatolian ancestry using simulations.

Statistical modeling and simulation are not my area of expertise so I will not be able to comment on the choice of method but will rather focus on the use of ancient DNA in this study or the choice of certain parameters. Nevertheless, I found the article well written and the explanation of the mathematical models and simulation rather clear and precise on the text. But being a non-specialist, I am surprised that the different symbols used in the mathematical equation in Fig1 are not explained, neither in the text, nor in the methods or supplement, this makes it really difficult to assess the validity of the model. ...

Validity:

Combining mathematical models, simulations and ancient DNA to test the process behind the Neolithic expansion is a very interesting subject and with the recent increase in genomic data it has become possible to do so at a finer scale. A wide range of recent studies have shown regional variability in the neolithization process, particularly with regard to interbreeding between farmers and HG. This variability was highlighted within each Neolithization route and between the two routes, the continental route and the Mediterranean route (i.e. Valdiosera 2018, Brace 2019, Rivollat 2020, Arzelier 2022, Lipson 2017...). It is therefore unfortunate that the authors did not attempt to apply their model on a finer scale, first by separating the two pathways of neolithization and then by looking at the process on a regional scale and finally by comparing them with the archaeological data.

This makes the conclusions of this article very difficult to transcribe in a context as varied as this neolithization process may have been.

Significance:

If modeling necessarily involves simplification of data to fit a mathematical model with a limited number of parameters, the meaning of the conclusion will depend on how well it fits reality. In the case of this study, working on the neolithization process as a whole and erasing regional diversity and the existence of the two waves are good reasons to question the main conclusion. The authors themselves discuss the specificity of Northern Europe and its impact on their conclusion (lines 435-451). It seems to me that it is these kinds of questions that interest research on the neolithization process rather than defining general rules which would ignore the variability of this phenomenon. I would strongly recommend at least testing the model with a separation between the two waves of Neolithization.

Data and methodology:

My main concern with the data used for this study is the choice of ancient DNA samples used to estimate the proportion of

Anatolian ancestry. There is not enough information in the main text or method to know which samples were used, where they came from, their dating, or their genomic coverage. The version of the AADR used should at least be mentioned as it is regularly updated and may have already changed since the article was submitted. The selection of samples "that represent populations competing with and after agricultural expansion but before subsequent steppe expansion", meaning several thousand years, is surprising enough to quantify a process that occurs throughout along the progression of neolithization. It would have been more appropriate to select only the samples corresponding to the early neolithic period along the diffusion route.

I have other concerns regarding the choice of some model parameters. For example, using a generation time of 30 years, instead of 25 years which is commonly used for prehistoric populations or the difference in population density between HG and farmers (can be quite different depending on the environment and landscape)... I am also wondering if the parameters should be the same between HG and farmers, for example the parameters governing the spatial distribution of each of these two groups will clearly be different and dependent on different factors related to the needs of the agriculture or hunting. Thus, the three main dynamics listed in line 122 for the different groups seem appropriate for predominantly sedentary farmers but perhaps not for seasonally mobile HGs...

The discussion around the implication of its results for the interpretation of archaeological contexts is almost absent from this work even though it would allow this approach to be recontextualized.

To summarize, the idea of combining genomic data and modeling is very interesting but it is not understandable that the authors chose to simplify their model so much and ignore the existence of the two neolithization currents. Under these conditions, it is justified to question the relevance of the conclusions and how they should be interpreted. I therefore recommend a major revision of this work with the consideration of regional differences or the application of the model to only one of the two diffusion trends.

(Remarks on code availability)

Reviewer #3

(Remarks to the Author)

LaPolice et al. present a novel work by combining, for the first time, a model that co-analyzes genetic ancestry and the speed of advance in the context of the Neolithization of the European continent. This represents a significant advance compared to previous studies on this phenomenon.

My background as a paleogeneticist leads me to focus my reviews on the use of concepts and empirical data drawn from archaeology and genetics, leaving aside modeling aspects. In this regard, I believe there are elements that need further explanation or discussion and that should be improved in the manuscript, as well as the inclusion of a section that addresses the limitations of the model.

IMPORTANT POINTS

1) Neolithization Routes

Although I understand the complexity of the topic, I missed a distinction between Mediterranean and Central European dispersion for a more complete Neolithization model. At least discussing this aspect would be helpful, and if this distinction was not made explaining why.

This point is important, especially when the discussion mentions in lines 444-449: "In our model's assessment of non-uniform expansion, conducting additional simulations with a decreased step size of farmers indeed demonstrated that a slower expansion speed leads to greater HG ancestry retention in the population following the expansion (Supplementary Fig. 9). This may explain why in northern regions where the farming expansion was presumably slower, e.g. due to the southwest Asian crop package underperforming in the colder and wetter climate, higher levels of HG ancestry are found 51."

This explanation fits well with the archaeological and genetic context in northern Europe. However, it would not be applicable to regions like the Iberian Peninsula, where the amount of hunter-gatherer ancestry is higher than in other areas of Europe, and the Neolithic arrived earlier due to maritime dispersion, which is faster.

2) Assortative Mating

I find it a term that implies too many social aspects that are difficult to prove in this case. Mating should not be understood as assortative if these individuals (hunter-gatherers and farmers) did not coincide in space/time. The survival of the last genetic hunter-gatherers has been demonstrated in the extremes of Europe, probably avoiding contact (Posth et al. 2023, Nature) or in forested areas or not adapted to agriculture (Rivollat et al. 2020, Science Advances; Tangermünde individual or Blätterhöhle individual). The choice emphasizes a social behavior/taboo when choosing a partner, but we do not know if these interactions between the two groups occurred simply because they did not occupy the same spaces. I do not consider it appropriate to use the term "assortative" for these reasons. Assortative or non-assortative is a preference but not finding people around is not a matter of choice.

3) Cultural Transmission and Learning Rate

It is said that "cultural transmission allows for the persistence of indigenous HG ancestry after the expansion." I believe it is necessary to point out that if individuals occupy different environments, this transmission does not occur, although hunter-gatherer genetic ancestry may persist. The examples I could cite are the same as I provided for assortative mating. I ask the authors to mention these cases in the discussion, as the archaeological context is important to assess the model's viability.

4) Ancestry Modeling

The explanation of data filtering is adequate, but it would be more illustrative if accompanied by a table with the data used and individuals grouped by geographical proximity. It is said that it filters between 5000 and 8000 BP. However, in these chronologies of 5000 BP in Central Europe, steppe ancestry is already found. I am not saying that the analysis is wrong, but some individual could have been included. To avoid these doubts, the manuscript would benefit greatly from a supplementary table with the samples used and a PCA of Western Eurasia where the individuals are seen in the genetic diversity pool representing the ancestry of the first farmers.

qpADM: The set of outgroups used in the article would be considered obsolete, although not incorrect. Currently, we avoid modern outgroups because we already have enough ancient data to represent all ancestries required in a distal model. Although I do not believe this significantly changes their estimates, I recommend following Patterson et al. 2023, which presents a set of outgroups very close to the sources which helps to narrow down the standard errors from admixture proportions. If in the future the authors wish to investigate bell beaker movements as well (as the manuscript seems to point out), this qpAdm model would be compatible. Another alternative would be to work with supervised ADMIXTURE.

Line 97: Who are these 911 individuals and what is the SNP cutoff used to estimate ancestry proportions? With lower SNP coverage, there is less chance of detecting small components. Also, I doubt there are 911 early Neolithic individuals. What other Neolithics are being considered? Would it be interesting to analyze only the early Neolithics from each geographic area? In the late Neolithic of many places, we see an increase in hunter-gatherer ancestry, interpreted as the final incorporation of the few remaining hunter-gatherers. If their model detects this increase in areas where only middle or late Neolithics have been sampled, but not early Neolithics, could their inferences vary? It is necessary to report specific Neolithics individuals used, their coverage, and their C14-group cultural dates, and discuss whether using one or the other can influence certain values (e.g., advance speed).

Line 292: Binning strategy: verify with PCA if they can be considered a cluster.

Line 294: "we only selected sites with more than 2 samples per site." The number of samples does not matter, but their coverage does. If the coverage is low, a component can be lost in the qpAdm model.

Line 296: It would be useful to mention the sites or contexts of these outliers with very low Anatolian ancestry. Are they matching with the examples I provided above?

Line 320: If considering qpAdm errors, I strongly recommend using (Patterson 2021; Nature).

5) Mortality Rates

Reference number 34. I am sure there is a better reference for Neolithic mortality than a report from a final Neolithic burial cave in Greece. There are many references from particular studies. If it is not explained why this particular one is chosen, it would be better to look for a more generic one. They can rely on Samanta-Cox et al. 2023, which states "Ages were determined on the basis of the average of the age range reported for each individual in their original publications," using the same AADR dataset. Through these age-at-death calculations, they can obtain a more suitable result for a Neolithic average in Europe, although they must consider that there is always a sampling, excavation, and analysis bias. For example, the absence of infant individuals may be due to a different ritual or much lower preservation.

Supplementary Table 1: it shows that the highest survival probability is at the age of 0 years. I do not believe is correct. Until the first years of life, survival probabilities are low in all non-industrialized population groups.

MINOR POINTS

Line 37: "It is believed that early European agriculture originated in southwest Asia 17–19 and first expanded into Europe from northwestern Anatolia 17."

Agriculture was developed earlier/or simultaneously in other areas, and genetically they show independent origins. I recommend mentioning this; a good summary paper can be Feldman et al. 2019.

Line 79: You mention the word "flora," but the referenced article refers to the "vegetational landscape." I would change "flora" to "paleoenvironmental data" or "vegetational landscape."

Line 85: I would say HG in general. Paleolithic HG are only Pleistocene HG, but the Holocene HG (Epipaleolithic and Mesolithic) also contributed.

Line 88: Western Anatolia is more accurate.

Lines 90-92: "limitation of DNA is that it does not directly describe or indicate behavior, culture, or specific mechanisms of expansion. As such, an interdisciplinary approach linking behavioral mechanisms with genetic data is required to fully understand the Neolithic farming expansion."

Clarify that archaeological contexts do this. Interdisciplinarity already exists in disciplines such as archaeogenetics.

Line 120: It is necessary to introduce the term WHG and not just talk about HG, since in Anatolia there would be other HG that are not the HG that you can infer with your qpAdm model. However, this does not mean they did not interact.

Reference 21: There is no genetic reference that reports more updated data on lower population density? Maybe based on heterozygosity or background relatedness with ROH, for example.

Line 130: Saying "post-Neolithic" is incorrect. It should be "Neolithic." I would avoid the term "post-Neolithic" because it is unclear whether it refers to another chronological period or simply to modern data.

Line 140: The same. What does "post-Neolithic" refer to? Be careful, there are other migrations that reduce farming and HG ancestry in post-Neolithic Europe. This is not mentioned, and I am not sure if it has been considered.

Line 133: Defining an HG culturally only by hunting would mean it never reaches zero, as hunting is never abandoned. There are sites called hunting grounds that individuals with Anatolian ancestry may use for hunting and not for farming. This occurs in most mountainous areas even in much later periods than the Neolithic.

Figure 1B: I might not understand this correctly, but why does Anatolian ancestry reach 1? In Europe, it usually does not reach 1 unless samples of such low coverage are included that a component is lost.

Line 188: "peer to peer transmission": Does this estimate align with what we know today? Only in Iron Gates do we find HGs in farming contexts and vice versa. However, HG individuals move to the extremes of Europe, and HG contexts, once farming arrives, can still be sites with a specific hunting functionality. I believe all this should be incorporated into an elaborate discussion explaining that the archaeological context is necessary for the discussion.

Figure 3A: The gradient should point from right to left, like eastern to western farming expansion?

Line 221: Please briefly explain how the models of references 20 and 21 calculate the "front speed." Do they take into account C14 dating?

Line 242: "post-neolithic" when it would be "neolithic expansion" or "neolithization" if you prefer.

Line 285: "This means some cultural transmission must have occurred, as a fully demic model would result in complete replacement."

I disagree with this phrase. Does "fully demic" not assume admixture can occur? Can it not be fully demic with local admixture? I do not think one thing excludes the other. If it is an assumption of the model, please explain it that way.

Line 396: "While archaeological and genetic research has described the impacts of the Neolithic transition from their respective disciplines, our interdisciplinary approach enables us to understand cultural and behavioral mechanisms invisible to the material evidence. Our mathematical and simulation-based modeling of the Neolithic expansion, for the first time, simultaneously predicts archaeological and genetic patterns linked to one of the most significant transformations in human history."

Do you believe that archaeology does not allow understanding "cultural and behavioral mechanisms"? We would not have studied this phenomenon from a genetic point of view if the archaeological context had not shown such a drastic change.

Lines 432-434: "Notably, this is corroborated by aDNA studies which do not find any farmer-associated ancestry in hunter-gatherers that overlapped in time with neighboring Neolithic farmers 56,57."

Could the examples I mention at the beginning be such? Tangermünde and Blätterhöhle. I also suggest a review of the genetic studies conducted in Iron Gates, where HG (Koros individual and others) were also found in farmer/nearby farmer contexts. They are exceptions, but I think they need to be mentioned in the discussion.

Lines 469-470: "Again, this was interpreted as a cultural expansion that was initially mediated by cultural diffusion 52."

This needs clarification, as the phrase would be true only if the Bell Beaker originated in Iberia (which is only one of the theories). If it originated in Central Europe, there is mostly genetic change associated with cultural dispersion. I do not think reference 52 leans towards either option, it just presents both as possible.

I hope my comments can help improve the manuscript and I urge the authors to follow the comments of the reviewers who are experts in modelling as this is outside my expertise.

(Remarks on code availability)

Version 1:

Reviewer comments:

Reviewer #1

(Remarks to the Author)

I have reviewed the authors' revisions and the responses to the reviewer comments. I have no further comments or suggestions. The authors have adequately addressed the comments from the reviewers.

(Remarks on code availability)

Reviewer #2

(Remarks to the Author)

First, I would like to thank the authors for providing a well-structured and detailed response to my main concerns. Their thorough reply includes meaningful methodological refinements while maintaining the study's original scope. They acknowledge the importance of regional variability in the Neolithic expansion and address this by modifying their model to distinguish between the Mediterranean and continental routes. The incorporation of multiple-route models, adjustments for movement speed, and geographic barriers demonstrate a genuine effort to refine their analysis. Additionally, the authors clarify that while regional variation is significant, their primary objective is to establish a broad baseline for cultural interactions and ancestry estimates. I also appreciate their discussion of the study's limitations and suggestions for future research, including integrating autosomal and mitochondrial data to investigate potential sex-specific migration and interaction patterns.

As a non-specialist in statistical modeling, I appreciate the effort to make the manuscript more accessible to researchers from different disciplines. The mathematical framework is now much clearer, and the added references in the figure caption, results, and methods sections help guide readers to the explanations more effectively. However, I suggest ensuring that the explanations in Supplementary Note 1 are as concise and intuitive as possible, particularly for readers without a strong background in mathematical modeling. A brief introductory sentence explaining the general role of reaction-diffusion models in studying population dynamics could further enhance clarity for a broader audience.

The inclusion of a supplementary data file detailing all analyzed individuals, along with metadata, filtering procedures, and qpAdm results, significantly improves transparency in data selection, ensuring clarity and reproducibility. I also appreciate the mention of the AADR database version in both the methods and results sections. However, it is worth noting that reliance on this database introduces a bias, as not all available samples are included.

While I acknowledge the effort to incorporate regional variability and differentiate between Neolithic expansion routes, I encourage a deeper discussion on how these findings align—or contrast—with existing archaeological evidence from specific regions. Some of the observed variations in learning rates could be explored in greater detail through references to archaeological material culture, settlement patterns, or subsistence strategies. Would the authors consider integrating more direct comparisons with archaeological datasets to further support their interpretations?

The additional analyses are valuable, but I still have some concerns about the inclusion of Middle and Late Neolithic samples. While learning rate estimates appear stable, the significant drop in the intercept during the Late Neolithic ($p < 0.001$) suggests that temporal variation does have an effect. Could the authors elaborate further on why they believe this does not impact their conclusions? Additionally, discussing the implications of using a broader time range in the main text would improve transparency.

Finally, while the modifications add regional nuance, the model remains relatively high-level. I understand that more detailed regional modeling would require extensive local archaeological data. However, given the increasing resolution of ancient DNA data, a finer-scale approach may be necessary to fully capture the complexity of regional variability. Would the authors consider discussing how future studies might bridge this gap?

(Remarks on code availability)

Reviewer #3

(Remarks to the Author)

I thank the authors for carefully addressing all my comments and for their clear point-by-point response.

I simply suggest that the authors explain in the methods section why Steppe was included as a source. Although I suggested using this model, it is true that this component is generally assumed to be unnecessary for explaining the ancestry of Neolithic individuals. In many of your models, this component is negligible, and when this happens, it is good practice to show the nested model (i.e., the same qpAdm model without the source that can be discarded, e.g., because its percentage is lower than its error margin). I do not consider it necessary to perform all possible nested models since the percentage changes would be minimal, but I believe it would be beneficial to explain why Steppe was retained in the model. In my view, the two main reasons for this would be:

1. As a validation method to confirm the absence of Steppe ancestry in the Late Neolithic individuals analyzed (serving the same purpose as a PCA).
 2. To quantify potential modern contamination, as any post-Neolithic human contaminant source would introduce some degree of Steppe ancestry.
- Excluding samples with more than 5% Steppe, as the authors have done, seems like very good practice.

Another point I would like to mention, just in case the authors are not aware, is that it might be useful to discuss in the methods section the occurrence of negative EF or WHG components. By using Balkan_N as a source, you standardize the type of farming ancestry used, as it has been observed that this ancestry may vary slightly by region (Lazaridis, 2024). Since it is a generalist source, it tends to yield higher p-values (which is the goal in qpAdm). However, by using Balkan_N, we create a farming ancestry pool that already contains some HG ancestry (similar to what happens with Anatolia, though perhaps to a lesser extent). This is what could lead to negative values for some samples, either for HG or farming ancestry, suggesting that the population source contains more of that component than the target.

To summarize, these negative values are informative and can be left as they are. It might be worth mentioning that the geographic origin of farming or regions east of the Balkans is where these negative components appear, as some individuals slightly deviate from the values of Balkan_N chosen as the EF source. Perhaps this could be briefly noted when performing linear regression and observing EF ancestry with increasing distance from the farming origin.

In conclusion, your source is not the exact origin, but it is an adequate proxy that improves p-values in Europe. Furthermore, having Anatolia in the outgroups and the Balkans in the sources helps reduce errors. For all these reasons—and because this approach is directly applicable to the future study of the Bell Beaker phenomenon—I strongly believe it is the best strategy to follow.

Line 863: The text states "85000," but the correct date is "8500."

Line 871: It would be better to say that the PCA reveals broad genetic ancestry rather than genetic relatedness.

These are minor comments, and I am satisfied with the thorough revision process carried out by the authors. I congratulate them on their work.

(Remarks on code availability)

Version 2:

Reviewer comments:

Reviewer #2

(Remarks to the Author)

The authors have thoroughly addressed all of my previous comments and suggestions. I am satisfied with the revisions and have no further comments.

(Remarks on code availability)

Reviewer #3

(Remarks to the Author)

I have no further changes to suggest. Thanks to the authors for all the work done, and congratulations on it.

(Remarks on code availability)

Reviewer #4

(Remarks to the Author)

Please see attached pdf file

(Remarks on code availability)

--

Version 3:

Reviewer comments:

Reviewer #4

(Remarks to the Author)
See attachment

(Remarks on code availability)
I reviewed the mathematica code but not the agent-based code.

LaPolice T.M., Williams M.P. & Huber C.D. Point by Point Response to Reviewers

We thank the reviewers for their time and their detailed insight. Below, we have responded to the comments from each reviewer, and made changes to our manuscript based on their valuable suggestions.

Reviewer #1

(Remarks to the Authors, Responses from the Authors, and Relevant Excerpts from the Revised Manuscript):

The authors assess the mechanisms of agricultural expansion in Neolithic Western Eurasia. The authors suggest that presence of hunter-gatherer ancestry in modern populations indicates a certain degree of cultural transmission and non-assortative mating. The authors use modeling and ancient DNA data to quantify the contribution of assortative mating and learning in the Neolithic expansion. They find limited cultural transmission and mainly assortative within-group mating.

I find the manuscript very interesting and think the modeling that was performed is of high quality. However, I think that the manuscript does not significantly deepen our understanding of the Neolithic transition. The authors find a mechanism of primarily demic expansion with limited learning, and primarily within-group mating, which have been shown several times already using different lines of evidence.

We appreciate the reviewer's time and effort in providing comments to improve our manuscript. We have made changes to the manuscript and our analysis to strengthen our conclusions based on the suggestions put forth by the reviewer.

Additionally, we respect the reviewer's concerns about novelty but want to emphasize that our model is novel in that it specifically has the ability to quantify the cultural contributions to an otherwise demic model. While previous aDNA studies have been transformative in revealing ancestry shifts between the European HGs and farmers, it remained unclear to what degree expanding farmers still interacted with indigenous HGs. Our approach, which combines aDNA with behavioral modeling, allows us to specifically quantify how frequently these interactive learning and mating behaviors occurred.

Further, in broadly assessing the relationship between ancestry changes and strength of population interaction, we have identified novel insights. We describe specific parameters under which ancestry and culture can become decoupled. Thus, our modeling is unique in that it is the first to describe a situation in which primarily demic expansion can occur

without ancestry turnover. This phenomenon has not been described to our knowledge in previous studies.

An important aspect of the Neolithic that is not being discussed here is the change through time in the 3000-year time transect that the authors are modeling. Hunter-gatherer ancestry proportion in Europe goes up from the Early to Middle Neolithic, with approximately 5% HG ancestry in early vs ~20-30% in the middle Neolithic.

WHG ancestry proportions indeed increased from around 5% in the Early Neolithic to roughly 20–30% in some Middle Neolithic communities. However, this so-called resurgence of WHG ancestry was not uniformly widespread across Europe but was particularly noticeable in Central and Western Europe. In these regions, small groups of WHG populations likely persisted in isolated areas even as farming spread. These populations may have interbred with Neolithic farming groups during the Middle Neolithic, reintroducing WHG ancestry into the gene pool.

This resurgence is a fascinating phenomenon; however, it lies beyond the scope of our modeling approach and does not alter our main conclusion regarding the strength of cultural interaction during the initial farming expansion. Our modeling explains the broader-scale patterns of ancestry turnover during the Neolithic Expansion. It could also be used to identify local deviations from this overall expectation. However, while future studies may explore specific local trends through detailed archaeological research or targeted cultural simulations, such analyses are outside the scope of our current study.

Nonetheless, to ensure that our analysis is robust to samples that might have been affected by a later WHG resurgence, we repeated our inference on samples exclusively from the Early Neolithic. In support of the robustness of our inference, we find similarly low learning rate estimates for samples from the Early, Middle, and Late Neolithic. A new Supplementary Figure has been added (Supplementary Figure 10) which shows the variability in the ancestry cline between time periods, and a new figure in the Main Text (Fig. 6) shows its influence on our inference.

We have added this caveat and new analysis now to the Results section on lines 427-437 which read:

“After confirming that our learning rate estimate is robust to a variety of landscape and expansion models, we next evaluated how the time period of the ancient individuals used in the fitting process may affect our conclusions. We separated the 618 ancient individuals into bins corresponding to the Early, Middle and Late Neolithic. We repeated our learning rate estimation with each subset of ancient individuals and found consistently low estimates regardless of time period: 0.00069 [0.00067, 0.00071] for Early, 0.0011 [0.001078, 0.001121] for Middle, and 0.001339 [0.001316, 0.001363] for Late Neolithic (See Fig. 6; Supplementary Fig. 5j-l). A linear regression on the ancient individuals from each time period also suggests there is no significant difference in the slope of the

ancestry cline between the Early and Middle Neolithic ($p=0.0636$) and no significant difference when comparing Early to Late ($p=0.0786$). However, there is a significant drop in the intercept from Early to Late ($p<0.001$) (Supplementary Fig. 10)."

(and lines 582-592 in Discussion):

"We also acknowledge that besides geographic variability, temporal variability of the ancestry proportions in the ancient individuals (for example, the WHG resurgence observed during the Middle and Late Neolithic⁵⁹) may also impact our estimated learning rate. In our estimation procedure, we assume that analyzed individuals are representative for a time period immediately after the conclusion of the farming expansion. To evaluate the robustness to this assumption, we split the dataset of ancient individuals into categories by age, corresponding to Early, Middle, and Late Neolithic (Supplementary Fig. 10). We then refit our complex landscape map simulation data to the ancient data from each time period. We again find little variability in the estimated learning rate with estimates all corresponding to approximately 1 in 1000 farmers participating in cultural transmission. This indicates that the WHG resurgence observed during the Middle and Late Neolithic⁵⁹ does not significantly impact our general conclusions."

Learning is being discussed throughout the manuscript but the term is vague and assumes a process that is neutral (L 58). There is a possibility that the process of learning was forced.

We have now noted that horizontal cultural transmission can occur either by choice or by force. The line referenced by the reviewer now has been clarified (with changes in red) to read (lines 69-71):

"Horizontal transmission refers to individuals learning behaviors from peers in other populations (either by choice or by force), whereas vertical transmission is the passing down of cultural practices from parents to offspring³³."

We also clarify in the discussion that our estimated learning rate may reflect various underlying processes (see lines 593-602):

"We want to emphasize that learning rate is an abstract parameter in our model that is ignorant about the underlying mechanism of "learning" itself. Cultural transmission may have been a purely neutral, voluntary exchange of knowledge between groups. However, the spread of farming during the Neolithic may have involved a mix of voluntary adoption and coercion. Competition for resources, social hierarchies, and economic dependencies could have pressured hunter-gatherers to adopt agriculture. Evidence of Neolithic conflict^{82,83} further suggests that some groups may have been compelled to abandon traditional ways, indicating that the transition to farming may not have always been by choice. While our model estimates a rate of conversion from HGs to farming, it does not

capture the specific cultural mechanisms at play, particularly whether this shift occurred willingly or under coercion.”

The authors suggest that “primarily demic expansion does not necessarily lead to a major ancestry turnover, suggesting that a naive interpretation of ancient ancestry patterns does not always reflect underlying behavioral mechanisms.” (L 104-106). However, it is unclear to me how exactly their modeling challenges this assumption.

We have altered the results and discussion in the main text to better explain this idea.

In the Results, lines (lines 454-462):

“Here, we more generally evaluate the contribution of cultural transmission to front speed in our simulations, and how this contribution relates to the turnover in ancestry. To this end, we evaluate the *cultural effect*^{31–33}. Cultural effect differs from our learning rate parameter in that cultural effect is a population-level measurement that we use to quantify the overall contribution of learning (or cultural transmission) to the spread of farming relative to a purely demic model^{31–33}. The cultural effect for each set of parameters tested is calculated using the following equation:

Cultural effect (%) = (front speed - demic speed)/front speed × 100”

And lines 482-493:

“...we see that for a large range of learning rates (i.e., f in [0.005, 0.1]) EF ancestry does not spread across Europe despite the cultural effect being low. I.e., despite the model essentially being demic, it does not lead to a turnover in ancestry (Fig. 7). Only when f is larger than 0.1 does cultural transmission predominantly determine the front speed, indicated by a *cultural effect* larger than 50%. This is notable because a lack of ancestry turnover during a cultural expansion has been used previously as evidence for a cultural transmission model, e.g. regarding the spread of farming within the Near East^{6,49}, or the expansion of the Bell Beaker culture between Iberia and Central Europe⁷⁰. However, our results indicate that there is a broad range of cultural transmission rates (i.e., learning rates f) that prevent significant genetic ancestry turnover—i.e., geographic ancestry patterns remain largely unchanged during the expansion—while still resulting in a process that is predominantly demic in nature (cultural effect < 50%).”

As well as (lines 613-633) in the discussion:

“In a primarily demic model, large-scale ancestry turnover is expected as populations with a novel ancestry component migrate into new areas and displace local populations (e.g., EF ancestry replacing local WHG ancestry during the Neolithic expansion in Europe). In contrast, a primarily cultural diffusion model predicts that original ancestry patterns remain intact as culture spreads through learning or imitation rather than population displacement.

In ancient DNA research, expansions are often classified as demic when substantial ancestry turnover occurs, or as cultural when ancestry patterns are preserved. Our findings challenge this binary framework, introducing a third, intermediate scenario: a predominantly demic expansion that occurs without significant ancestry turnover.

To this end, we use the cultural effect metric^{31–33}, which quantifies the role of cultural transmission in the expansion process. This metric measures the unique contribution of cultural transmission to the overall expansion speed of the farming cultural package relative to a purely demic model driven by population movement alone. We find that, under a wide range of low but sufficient learning rates, the cultural effect is minimal, suggesting a predominantly demic expansion speed. However, under these conditions, the ancestry patterns resemble those of a cultural diffusion model, where original ancestry patterns are preserved. This can result in an expansion being misclassified as cultural due to the absence of ancestry turnover, despite its predominantly demic nature.

Thus, our results demonstrate that initial ancestry patterns can be preserved even in a predominantly demic expansion model, underscoring the limitations of traditional binary classifications based on ancestry turnover.”

Specific comments and questions below:

- The abstract states that the primary mechanism of Neolithic expansion is debated. However, it has been shown many times using genetic data that the primary mechanism was demic diffusion.

We have altered this portion of the abstract to reflect this was primarily a historical debate rather than an ongoing one. We also altered slightly to reflect that our model is unique in that it specifically *quantifies the role of cultural transmission within a primarily demic model*. This point was not clear in our first abstract. The portion of the abstract the reviewer commented on now reads as follows (changes noted in red):

“Historically, the underlying mechanisms of agricultural expansion have been a topic of debate, centered around two primary models: cultural diffusion, involving the transfer of knowledge and practices, and demic diffusion, characterized by the migration and replacement of populations. More recently, ancient DNA analyses have revealed significant ancestry changes during Europe’s Neolithic transition, suggesting a primarily demic expansion. Nonetheless, the presence of 10-15% hunter-gatherer ancestry in modern Europeans indicates that low levels of cultural transmission or between-group mating were additional contributing factors.”

- L. 37 in the paragraph the authors say that the first people to describe the spread of agriculture were Ammerman and Cavalli-Sforza. There is a rich history of this research in archaeology that is not being discussed.

We thank the reviewer for this helpful note and acknowledge there is also a large body of archaeological research pre-dating Ammerman and Cavalli-Sforza's work. We have now altered this section and noted this history.

This section of the introduction has been revised and we have added lines 39-48 to acknowledge archaeological research on the spread of farming predating Ammerman and Cavalli-Sforza:

“The spread of agriculture in prehistoric Europe has been a longstanding subject of research, with early archaeological studies focusing on the spatial and temporal distribution of material culture associated with farming communities^{11,18,19}. Pioneering work by scholars such as V. Gordon Childe introduced the concept of the Neolithic Revolution¹¹⁻¹³, emphasizing the transformative impact of agriculture on human societies. Many archaeologists suggested migration as a potential driver of the spread of agriculture, though later some proposed cultural diffusion, where farming practices spread without significant population movement^{18,20,21}. Building on these foundational insights, Ammerman and Cavalli-Sforza (1971)²² first introduced a quantitative framework for understanding the Neolithic expansion, that integrated archaeological data with genetic expectations^{22,23}.”

- L. 97: the authors say here that they derived ancestry estimates from 911 early Neolithic farmers. However, L288: it is said that they include all Neolithic populations until the Steppe ancestry arrival, which would also include the middle and late Neolithic.

We thank the reviewer for their comment. Yes, we had included individuals from the Middle and Late Neolithic in our qpAdm analysis, as indicated in former line 288. Moreover, following Reviewer 3's recommendations, we have now substantially refined our qpAdm analysis by implementing the framework established by Patterson et al. (2022). This more accurate approach emphasizes individual-level ancestry estimation and employs strategically selected source and reference populations. Our analysis quantifies three distinct ancestral components: Steppe-related, Western European hunter-gatherer (WHG)-related, and Early European farming (EF)-related ancestry. The EF ancestral component is represented by 22 Mediterranean and Balkan individuals exhibiting minimal WHG ancestry (Patterson et al., 2022). In accordance with qpAdm ancient DNA standards (Harney et al., 2021), our updated qpAdm reference group is exclusively from high-coverage ancient DNA samples. With this new approach we restricted our analysis to individuals dating between 5000 and 8500 BP and filtered any related individuals or those assigned as contaminated in the AADR metadata, resulting in 1531 analyzed individuals. We then further restricted our analysis to individuals with a plausible qpAdm model (p-value ≥ 0.01 and admixture weights [0,1]), admixture weight standard error < 0.022 , and removed any individuals with $> 5\%$ Steppe ancestry, resulting in a final empirical analysis set of 618 individuals.

To avoid misunderstanding, we changed former line 97 (now lines 122-124) to:

“We fit parameters in our model to archeological front speed estimates^{21,28,29} and ancestry estimates derived from 618 European Neolithic individuals that were plausibly modeled as mixtures of WHG and EF ancestry.”

- L. 103 “... less than 0.1% of farmers converting an HG to farming annually”: phrase is confusing. Please rephrase the sentence.

We have clarified this line and section. Lines 125-128 now read:

“We conclude that there must have been near complete cultural within-group mating in farming and hunter-gathering groups. Additionally, we find few farmers participated in cultural transmission, suggesting that less than 0.1% of farmers converted a hunter-gatherer to farming annually.”

- L 129 -131. The Neolithic ancestry proportion varies through time and space. Both have to be taken into account in the modeling.

We thank the reviewer for this comment. The reviewer is correct that Neolithic ancestry proportions vary across both time and space, and we account for this in our modeling. We solve the reaction-diffusion equation over both dimensions. However, we observe that, following the initial farming expansion, the spatial gradient of farming ancestry remains largely stable over hundreds of years, with migration exerting only a minor effect on this gradient. In response to the reviewer’s suggestion, we have clarified this in the manuscript (lines 155-156), now stating, “We solved the model equations numerically to investigate the resulting change in farming ancestry over a time span of 150 generations and across the 3000 km transect.”

- Figure 1B: The different learning rates seem very similar to each other, i.e. there is a very small difference in Anatolian Ancestry between learning rate 1 and 0.001. Could you explain why there is so little variation?

We thank the reviewer for their observation. In Figure 1B, the x-axis of the plot varies not only in learning rate but also in the gamma parameter, which represents the preference of hunter-gatherers (HGs) to adopt behaviors from farmers rather than from other HGs. A gamma value of one indicates no preference, meaning copying is random. More specifically, the x-axis of Figure 1B represents the *ratio* of learning rate and gamma. A previous study demonstrated that front speed is determined by this ratio, rather than by each parameter individually (Fort, 2012). The reason is that this ratio equals the number of hunter-gatherers converted to farming per farmer per year, and thus is the main determinant of the front speed. We observe a similar effect of learning rate and gamma on ancestry expansion (i.e., the amount of farming ancestry remaining after the expansion): it primarily depends on the ratio of these two parameters (as shown on the x-axis in Fig. 1B), rather than on each parameter alone.

For this reason, we assume a gamma parameter of one in subsequent analyses, which does not limit the generality of our findings. However, it's important to note that varying the learning rate alone does indeed have a significant impact on the amount of farming ancestry that persists.

- Could you please provide more details on what the “learning rate” means? Does learning rate of 0.001 indicate demic diffusion vs learning rate of 1 cultural diffusion?

In our simulation, the transition of a hunter-gatherer individual to farming is modeled as a probabilistic event. This probability is determined by the product of the local proportion of farmers surrounding the hunter-gatherer and a global learning rate f (again, assuming a gamma value of one). The learning rate f can be understood as the "conversion rate"—specifically, it represents either the number of HGs converted per farmer when the local proportion of farmers is very low (at the leading edge of the farming expansion) or the proportion of hunter-gatherers adopting farming when it becomes the dominant local subsistence strategy. This approach to learning is consistent with the cultural transmission model described by Fort (2012).

A learning rate of zero represents purely demic diffusion, as no hunter-gatherers adopt farming based on cultural transmission. Conversely, a learning rate close to one (i.e., close to 100%) would indicate a primarily cultural diffusion process, where HGs almost always adopt farming if they are near a farmer, implying that “ideas spread faster than people”.

Rather than categorizing cultural spread strictly as either demic or cultural, we introduce a continuous measure called the *cultural effect*. The cultural effect quantifies the contribution of cultural transmission (learning) to the expansion speed, relative to a purely demic model. This approach allows us to measure the balance between demic and cultural diffusion when both processes are at play.

A detailed explanation of the learning rate and its implementation in our agent-based simulations is provided in the Methods, including its interpretation.

- Figure 2. Please remove the icon of the tie, since it is not accurate for the time period and has inherent assumptions about the populations under study.

The tie icon has been removed from the teacher in Figure 2 to remove any inherent bias in the depiction of a teacher.

- Why only learning rates between 0 and 0.005 were tested (L 245)? L162: upper limit of learning rate of 0.1 was found under unbiased learning.

The upper limit of the learning rate, identified as 0.1 in the mathematical reaction-diffusion model, represents a per-generation rate. However, our agent-based simulation operates

on an annual time step, estimating rates on a per-year basis. Converting the per-generation learning rate of 0.1 to a yearly rate yields approximately 0.004, assuming a generation time of 25 years. As anticipated from the mathematical model, we find that in our agent based model a learning rate of 0.005 or higher results in high retention of WHG ancestry after the farming expansion (see Fig. 4A). Based on this, when running simulations for fitting, we set an upper boundary of 0.005 for the learning rate in our annual agent-based simulations.

- The authors should include the IDs of all individuals included in their study. I would also advise against binning samples from the same geographic location (L 291) due to potential ancestry changes through time, and to also to include a temporal component into the modeling.

In response to this and another reviewer's comment, we have re-analyzed our ancient DNA data using the latest version of the AADR database (v62.0) and conducted individual-level qpAdm ancestry estimations following the approach in Patterson et al. (2022). Thus, rather than binning samples by geographic location, we now capture inter-individual variability in farming ancestry across Europe during the Neolithic.

Further, our modeling approach includes a temporal component by accounting for the speed of the farming expansion, with movement parameters aligned to the empirically observed rate of 1 km/year. However, it primarily fits the ancestry data to the spatial cline in farming ancestry that remains post-expansion, as this cline is highly informative for estimating the learning parameter, whereas the expansion speed is not (e.g., see Figs. 3B and 3C).

Additionally, we now examined changes in the empirical cline over time (from Early to Middle to Late Neolithic) to assess any impact on our learning rate estimates (Supplementary Figure 10). While there is a known increase in WHG ancestry during the Middle Neolithic due to WHG resurgence in certain regions, we find that the overall cline in ancestry remains largely stable. Importantly, our inference of a low learning rate is robust to the time period within the Neolithic from which samples are drawn.

This new analysis is now included in the results section on lines 479-489 and in lines 651-661 of the discussion. Please see also our response to the previous comment on the increase in Hunter-Gatherer ancestry from the Early to the Middle Neolithic.

See lines 427-437 in the Results:

“After confirming that our learning rate estimate is robust to a variety of landscape and expansion models, we next evaluated how the time period of the ancient individuals used in the fitting process may affect our conclusions. We separated the 618 ancient individuals into bins corresponding to the Early, Middle and Late Neolithic. We repeated our learning rate estimation with each subset of ancient individuals and found consistently low

estimates regardless of time period: 0.00069 [0.00067, 0.00071] for Early, 0.0011 [0.001078, 0.001121] for Middle, and 0.001339 [0.001316, 0.001363] for Late Neolithic (See Fig. 6; Supplementary Fig. 5j-l). A linear regression on the ancient individuals from each time period also suggests there is no significant difference in the slope of the ancestry cline between the Early and Middle Neolithic ($p=0.0636$) and no significant difference when comparing Early to Late ($p=0.0786$). However, there is a significant drop in the intercept from Early to Late ($p<0.001$) (Supplementary Fig. 10).”

(and lines 582-592 in Discussion):

“We also acknowledge that besides geographic variability, temporal variability of the ancestry proportions in the ancient individuals (for example, the WHG resurgence observed during the Middle and Late Neolithic⁵⁹) may also impact our estimated learning rate. In our estimation procedure, we assume that analyzed individuals are representative for a time period immediately after the conclusion of the farming expansion. To evaluate the robustness to this assumption, we split the dataset of ancient individuals into categories by age, corresponding to Early, Middle, and Late Neolithic (Supplementary Fig. 10). We then refit our complex landscape map simulation data to the ancient data from each time period. We again find little variability in the estimated learning rate with estimates all corresponding to approximately 1 in 1000 farmers participating in cultural transmission. This indicates that the WHG resurgence observed during the Middle and Late Neolithic⁵⁹ does not significantly impact our general conclusions.”

- The authors should include the qpAdm results (L 293) as a supplementary data file.

Per the reviewer’s request we have added a supplementary data file containing the qpAdm results for all 1531 analyzed individuals (pre-filtering) (Supplementary Data File 2) and a metadata file for each individual (pre-filtering) (Supplementary Data File 1). The metadata file provides detailed information for all individuals tested (pre-filtering), including the genetic ID, sample ID, qpAdm analysis ID, publication source, 1240k SNP coverage, data type (capture or shotgun), mean BP date, molecular sex, and site latitude and longitude. We have also provided detailed information on our filtering procedure following guidelines by Patterson et al. 2022, and note which individuals eventually were used for our analyses (post-filtering). These files ensure transparency and allow readers to assess the dataset more comprehensively.

- L. 351: the learning rate interpretation states: “only 1 in 1000 farmers converted a hunter-gatherer each year”. Naively, I would have thought that the learning rate reflects how many HGs are converted, since more than one farmer can convert one HG.

Thanks to the reviewer for raising this point—it does warrant clarification. In our model, the probability of an HG adopting farming is calculated as the product of the local proportion of farmers and the simulation-wide learning rate f . This has two possible interpretations, as noted in Fort (2012): (1) when the local proportion of farmers is low

(e.g., at the wave-front of the farming expansion), it represents the "number of HGs converted per farmer"; (2) when farming is the dominant local strategy (i.e., there are far more farmers than HGs), it reflects the proportion of HGs converting to farming. We outline these interpretations in the Methods section. In the text at the referenced line, we are referring specifically to the first interpretation—what occurs at the tip of the wave-front. Importantly, since the learning rate is significantly less than one, instead of indicating the number of HG converted by a farmer, it indicates the probability that a farmer converts an HG.

We have clarified the text as follows (lines 445-447):

"In sum, despite very different landscape and expansion models, we consistently estimate a yearly learning rate of ~0.1% or less (Fig. 6). Given this low rate, it suggests that at the tip of the wave-front, only 1 in 1,000 farmers converted 1 HG each year (see Methods)."

- L. 361: What is the difference between "cultural effect" and "learning", since the two are related and somewhat try to capture the same process of learning the cultural practice of farming?

Thanks to the reviewer for this insightful question. While "learning" and "cultural effect" are indeed related concepts in our model, they capture different aspects of the spread of farming.

The learning rate is a model parameter that represents the probability of a hunter-gatherer adopting farming based on the presence of nearby farmers. It quantifies the likelihood of cultural transmission on an individual level.

The cultural effect, on the other hand, is a metric we use to quantify the impact of cultural transmission on the overall expansion speed of farming. It measures the extent to which cultural transmission (i.e., learning) contributes to the spread of farming, relative to a purely demic expansion model. In other words, while the learning rate operates at the individual level, the cultural effect captures the broader, population-level influence of learning on the speed of the expansion of farming culture.

We have added the following sentences to lines 457-462 to help clarify this distinction:

"Cultural effect differs from our learning rate parameter in that cultural effect is a population-level measurement that we use to quantify the overall contribution of learning (or cultural transmission) to the spread of farming relative to a purely demic model³¹⁻³³. The cultural effect for each set of parameters tested is calculated using the following equation:

Cultural effect (%) = (front speed - demic speed)/front speed × 100 "

- L 390-393: “However, our results indicate that there is a wide range of cultural transmission rates that prevent a turnover of genetic ancestry, but are small enough to lead to a predominantly demic diffusion of culture (cultural effect < 50%), i.e. geographic ancestry patterns do not necessarily change during a demic expansion of culture.” Could you please explain what you mean here, since your modeling shows that with cultural effect of more than 50% there is almost no Anatolian ancestry, and with the cultural effect of 0.05-0.1 the mean Anatolian ancestry looks like 0.25, which indicates an ancestry change.

Thank you for this comment. The expectation for a fully or predominantly demic diffusion model is a substantial turnover in ancestry, while a purely cultural diffusion is expected to preserve existing ancestry patterns (Ammerman & Cavalli-Sforza, 1984). This binary classification has been used in previous studies to distinguish between demic and cultural diffusion. However, our results show that for a wide range of learning rates, the expansion remains primarily demic (cultural effect < 50%) but does not result in substantial ancestry turnover. This finding challenges the traditional binary classification.

In response to the reviewer’s comment, we note that in Figure 7 (previously Figure 6 in original submission), the y-axis represents the mean ancestry across the entire simulated population, averaged over the full geographic range. Notably, individuals with high levels of farming ancestry are clustered near the origin of farming, and this ancestry becomes increasingly diluted with distance. Thus, an overall mean farming ancestry of 25% reflects a concentration of ancestry near the origin, with negligible levels across most of the map (as shown in Figure 4A for a learning rate of 0.01). Thus, although some expansion of farming ancestry occurs, this pattern would likely be interpreted as "cultural diffusion without ancestry replacement" in an empirical ancient DNA study, especially for regions far from the origin of the culture.

We have revised the sentence in question for greater clarity, indicating that ancestry patterns remain largely unchanged (see lines 489-493):

“However, our results indicate that there is a broad range of cultural transmission rates (i.e., learning rates f) that prevent significant genetic ancestry turnover—i.e., geographic ancestry patterns remain largely unchanged during the expansion—while still resulting in a process that is predominantly demic in nature (cultural effect < 50%).”

We have also added clarification to the Fig. 7 caption (changes in red):

“For comparison to empirical data, the dashed gray line represents the mean proportion of EF ancestry ancestry **across the whole landscape** from our aDNA estimates.”

- **General comment.** Could you please change the color of the points in figures? Some colors are too similar to easily differentiate visually.

We appreciate this feedback from the reviewer. To improve clarity, we have adjusted the color scheme by using a custom color palette which provides a better visual contrast between lines/points on plots (see Figures 3c, 4a, 4b, 6, and 7 as well as Supplemental Figures 1, 12, and 14).

Reviewer #2

(Remarks to the Authors, Responses from the Authors, and Relevant Excerpts from the Revised Manuscript):

In this study, LaPolice and colleagues present a mathematical model to quantify the roles of cultural diffusion and mating during Neolithic expansion. The originality of their work is to combine agent-based simulations and ancient DNA analyses. The first part of their work examines the influence of cultural transmission on the dispersal of agricultural ancestors with a unidimensional model. From this simplified model, they suggest a low level of cultural transmission, i.e. 1 farmer per 1000 farmers converts one HG per year. Then, they offer different simulations to test more complex scenarios. They estimate the effect of a parameter such as front speed on the rate of cultural transmission and found no effect in their simulations. Finally, they test the effect of assortative mating and learning on the proportion of Anatolian ancestry using simulations.

Statistical modeling and simulation are not my area of expertise so I will not be able to comment on the choice of method but will rather focus on the use of ancient DNA in this study or the choice of certain parameters. Nevertheless, I found the article well written and the explanation of the mathematical models and simulation rather clear and precise on the text. But being a non-specialist, I am surprised that the different symbols used in the mathematical equation in Fig1 are not explained, neither in the text, nor in the methods or supplement, this makes it really difficult to assess the validity of the model.

We appreciate the reviewer's comments and suggestions regarding the ancient DNA analysis and all other comments to improve the manuscript. Given the interdisciplinary nature of the work, and the potential for readers from multiple subspecialties, we are committed to making the manuscript as accessible as possible. To this end, we thank the reviewer for pointing out this issue with the equations in Fig. 1. We now provide a comprehensive explanation of the symbols and the different components of the equations in Supplementary Note 1.

We now reference this explanation in the figure caption of Fig. 1 (red text):

“Our mathematical model utilizes a reaction-diffusion framework to simulate the spread of EF ancestry across a one-dimensional European landscape initially populated by hunter-gatherers. It comprises a set of partial differential equations that describe the movement

(diffusion) of individuals, their population growth (logistic), and the impact of cultural transmission on genetic ancestry. For a comprehensive description of the symbols and components of the equations see Supplementary Note 1.”

And again in the Results text on lines 150-154:

“Our cultural transmission mechanism is informed by established theory³¹, considering a learning rate (f , the product of the number of teachers times the probability of learning per contact), and a bias parameter (γ), which regulates the HGs’ propensity to adopt farming practices from nearby farmers (see Fig. 1a; for a comprehensive description of the symbols and components of the equations, see Supplementary Note 1).”

In the Methods lines 670-673 we also reference the new supplementary note:

“Our mathematical model assumes a diffusion component, a logistic population growth component, and a cultural transmission component (see formulas in Fig. 1a; for a comprehensive description of the symbols and components of the equations in Fig. 1a, see Supplementary Note 1).”

The referenced newly added Supplementary Note 1 reads as follows:

“Our reaction-diffusion model of the farming expansion is a system of equations that models three population groups, represented by the variables u , v , and w , which correspond to the densities of farmers with farming (EF) ancestry, farmers with western hunter-gatherer (WHG) ancestry, and hunter-gatherers with WHG ancestry, respectively (all measured in people per km). Each equation has three main components that describe different aspects of population dynamics: diffusion or movement across space, logistic population growth, and cultural transmission (learning).

The first term in each equation represents diffusion, or spatial movement, of each population group across the landscape. The parameters D_u , D_v , and D_w are diffusion coefficients for the respective groups (u , v , and w), reflecting the rate at which each group spreads spatially. The second partial derivatives indicate the spatial distribution or diffusion of each population group over time.

The second term in each equation models logistic population growth, constrained by carrying capacities. The parameters a_u , a_v , and a_w represent the intrinsic growth rates of each population type, while K_F and K_{HG} are the carrying capacities for farmers and hunter-gatherers, respectively. Logistic growth terms, such as $1 - \frac{u+v+w}{K_F}$ or $1 - \frac{u+v+w}{K_{HG}}$, limit population growth as densities approach these carrying capacities, thereby ensuring that each population's growth slows as it nears the environment's maximum sustainable density.

The final term in each equation represents cultural transmission, or learning, where hunter-gatherers may adopt farming practices from neighboring farmers. The learning rate f indicates the probability or rate at which this adoption occurs, while γ is a preference factor reflecting the tendency for hunter-gatherers to copy farming behavior from farmers rather than from other hunter-gatherers. The expression $\frac{(u+v)w}{u+v+\gamma w}$ models the likelihood of hunter-gatherers with WHG ancestry (w) adopting farming practices, depending on the density of farmers ($u+v$) and the preference factor γ . Together, these terms describe the interplay of diffusion, growth, and cultural transmission in the spread of farming practices.”

Validity:

Combining mathematical models, simulations and ancient DNA to test the process behind the Neolithic expansion is a very interesting subject and with the recent increase in genomic data it has become possible to do so at a finer scale. A wide range of recent studies have shown regional variability in the neolithization process, particularly with regard to interbreeding between farmers and HG. This variability was highlighted within each Neolithization route and between the two routes, the continental route and the Mediterranean route (i.e. Valdiosera 2018, Brace 2019, Rivollat 2020, Arzelier 2022, Lipson 2017...). It is therefore unfortunate that the authors did not attempt to apply their model on a finer scale, first by separating the two pathways of neolithization and then by looking at the process on a regional scale and finally by comparing them with the archaeological data.

This makes the conclusions of this article very difficult to transcribe in a context as varied as this neolithization process may have been.

We thank the reviewer for their thoughtful feedback and for highlighting the importance of regional variability in the Neolithic expansion. We agree that recent studies have demonstrated variation in the neolithization process, especially in the interbreeding patterns between farmers and hunter-gatherers along the continental and Mediterranean routes. This is indeed an exciting area of research, and we appreciate the suggestion to apply our model on a finer scale to capture these regional dynamics.

While we agree that understanding local cultural interactions is valuable, the aim of our study is broader: to examine the effects of cultural parameters on ancestry estimates across a continent-wide scale rather than focusing on specific regions. Our approach establishes a baseline relationship between cultural interactions and spatiotemporal genetic ancestry patterns on a broad scale, which will serve as a reference for identifying and investigating regional deviations in empirical data. Applying the model to specific regions, however, would require detailed local archaeological data and methodological adjustments to account for factors like variable interbreeding and expansion rates at the local scale.

Nonetheless, to address the reviewer’s main concern, we have modified our model to differentiate between the continental and Mediterranean/Coastal routes of Neolithic expansion. In addition to our original simulations, we developed two new multiple-route

models. We increased movement speed along waterways based on parameters from literature (Davison et al., 2006; Fort & Pérez-Losada, 2024) and treated mountainous regions as uninhabitable barriers, resulting in distinct expansion pathways. This adaptation allows us to model separate corridors, with faster dispersal along the Mediterranean coast and the Rhine/Danube corridor, and slower expansion in northern regions.

Despite the inclusion of this increased spatial heterogeneity in expansion speed, our central conclusion regarding the low level of cultural interaction between farmers and hunter-gatherers remains consistent, as the estimated learning rates between groups are consistently low (0.001124 [0.00111, 0.001137], 0.000938 [0.000927, 0.000949], 0.000991 [0.000978, 0.001003]) under these revised models compared to 0.00101 [0.000998, 0.001023] under our standard complex model.

For a more detailed analysis, we divided the individual ancestry estimates into two groups based on latitude: southern individuals associated with the Mediterranean route (latitude $\leq 45^\circ$) and northern individuals linked to the Rhine/Danube corridor (latitude $> 45^\circ$). When fitting our model to these groups separately, we found that the estimated learning rate along the Mediterranean route is slightly larger (0.001244 [0.001208, 0.001279]) than for the northern route (0.000914 [0.000901, 0.000927]). This finding contrasts with a recent study based on mitochondrial haplotype clines, which suggested that learning rates are consistent across both routes (Fort & Pérez-Losada, 2024). There are two possible explanations for this discrepancy. First, our analysis benefits from a larger sample size and the higher resolution afforded by genome-wide ancestry estimates, allowing us to detect subtler differences in learning rates that may be missed using mitochondrial haplotype data. Second, mitochondrial haplotypes are maternally inherited and reflect only female ancestry patterns. This raises the possibility of sex-specific cultural interaction patterns and movement rates. Although outside of the scope of our study, future work should aim to integrate autosomal and mitochondrial data into a unified framework to investigate potential sex-specific parameters.

In conclusion, while a more detailed analysis of the Neolithic expansion reveals subtle heterogeneity in cultural transmission rates, this variability does not alter our primary conclusion of low learning rates, estimated to be on the order of 1 in 1000.

We have added new sections to the results and discussion to describe this approach and its findings:

Please see lines 400-426 in Results:

“However, prior archaeological studies have indicated that the agricultural expansion was not uniform but might have spread faster along a coastal Mediterranean route compared to a northerly route through the Balkans and into Central Europe^{44,67–69}. To assess how non-uniform expansion might have influenced our analysis, we conducted simulations under three different non-uniform models. The first model used the same parameters and

maps as the uniform complex map expansion but adjusted the step size parameter to reflect directional differences in dispersal. Specifically, we reduced the step size in the North-South direction to 4 km per year and increased it in the East-West direction to 6 km per year, instead of the uniform step size of 5 km per year. Using the same maximum likelihood approach as before, we found a similar estimated learning rate of 0.001124 [0.001111, 0.001137] (Fig. 6; Supplementary Fig. 5e)..

To further investigate the effects of heterogeneous dispersal, we applied a model that increased the step size for individuals near the Mediterranean coast, based on speed estimates from Fort and Pérez-Losada (2024)⁶⁹. In addition, mountainous regions were defined as barriers to dispersal. This model yielded results that were again in line with previous analyses, with an estimated learning rate of 0.000938 [0.000927, 0.000949] (Fig. 6; Supplementary Fig. 5f). To introduce further heterogeneity, we incorporated parameters from Davison et al., (2006)⁴⁴ which suggest that farming expansions were approximately 5x faster along the Danube-Rhine Corridor and 10x faster along the Mediterranean Coast. The estimated learning rate again remained consistent with earlier findings, yielding a value of 0.000991 [0.000978, 0.001003] (Fig. 6; Supplementary Fig. 5g). All simulation maps with expansion corridors can be found in Supplementary Fig. 8 and Supplementary Fig. 9. Finally, we allowed for regional variation in learning rate by dividing individuals into a northern and a southern group based on a latitude of 45°. Separate learning rates were estimated for each group, revealing a slightly higher rate in the South compared to the North (0.001244 [0.001208, 0.001279], and 0.000914 [0.000901, 0.000927] respectively) (Fig. 6; Supplementary Fig. 5h-i). Nonetheless, our overall conclusion of a cultural learning rate on the order of 0.1% remains robust and independent of the expansion route.”

We have added a note in the discussion to acknowledge the limitations of our study regarding regional variation and to outline potential directions for future research.

Please see lines 563-581 in the Discussion:

“Importantly, estimated learning rates remained consistently low (Fig. 6), irrespective of whether we assumed a square model with uniform dispersal, a complex geographic model with uniform dispersal, a model with faster latitudinal expansion or models with independent Mediterranean and inland expansion routes. This illustrates the robustness of our findings to spatial dispersal heterogeneity. Interestingly, when estimating learning rates separately for Southern and Northern European individuals, we found that the estimated rate for the South was slightly higher than for the North. This finding contrasts with a recent study based on mitochondrial haplotype clines, which reported uniform learning rates across both Mediterranean and inland routes⁶⁹. Two factors may account for this discrepancy. First, our analysis utilized a larger sample size and the finer resolution of genome-wide ancestry estimates, which likely allowed us to detect subtle differences in learning rates that mitochondrial data may not capture. Second, because mitochondrial haplotypes are maternally inherited, they reflect only female ancestry patterns, potentially missing sex-specific cultural interactions and movement dynamics. Future studies should aim to integrate autosomal and mitochondrial data into a unified framework to better

understand the influence of sex-specific factors on cultural transmission and migration dynamics. Future study could also delve into the potential for regional variation of cultural transmission rates and how it could be linked to observed local patterns derived from archaeological studies, integrating our quantitative estimates of interaction rates with local genetic and archeological interpretations.”

And additionally, lines 535-549 we discuss the prolonged coexistence and minimal gene flow between farmers and hunter-gatherers in certain regions:

“Notably, aDNA studies from Atlantic France and Scandinavia show no evidence of farmer-associated ancestry in hunter-gatherers who overlapped in time with neighboring Neolithic farmers^{75,76}, corroborating our finding of low between-group mating. Additionally, in some regions, evidence suggests that foragers and farmers coexisted for centuries with minimal gene flow. In northeastern France and Germany, some late Neolithic farmers display unexpectedly high levels of hunter-gatherer ancestry (e.g., Mont-Aimé: 50–63%, Tangermünde: ~63%, Blätterhöhle: up to 85%), reflecting prolonged coexistence and limited genetic integration^{61,77–79}. Similarly, in the Iron Gates region, hunter-gatherers and early farmers lived side by side for centuries, exhibiting only minimal admixture despite extended contact⁸⁰.

These patterns suggest persistent but low levels of asymmetric admixture^{61,81} and reveal long-term demographic dynamics, such as the survival of regional hunter-gatherer groups and potential cultural reversions by farmers to hunter-gatherer subsistence strategies. While our model focuses on the initial farming expansion, integrating these long-term dynamics into our model could offer valuable insights into the diversity and variability of local interactions and population structures following the initial farming expansion.”

Significance:

If modeling necessarily involves simplification of data to fit a mathematical model with a limited number of parameters, the meaning of the conclusion will depend on how well it fits reality. In the case of this study, working on the neolithization process as a whole and erasing regional diversity and the existence of the two waves are good reasons to question the main conclusion. The authors themselves discuss the specificity of Northern Europe and its impact on their conclusion (lines 435-451). It seems to me that it is these kinds of questions that interest research on the neolithization process rather than defining general rules which would ignore the variability of this phenomenon. I would strongly recommend at least testing the model with a separation between the two waves of Neolithization.

We appreciate the suggestion to test the model by separating the two main Neolithic expansion routes: the continental and Mediterranean pathways. As noted above, we have introduced new multiple-route models that incorporate increased movement along waterways and consider mountainous regions as uninhabitable barriers, creating separate corridors for the continental and Mediterranean routes. These models allow us to explore distinct expansion pathways and their impact on genetic ancestry patterns.

As noted in our response to the previous comment, our findings from these refined models indicate that our main conclusion regarding the low level of cultural interaction between farmers and hunter-gatherers remains robust. Nevertheless, we have added new sections to the methods, results, and discussion, to describe this approach and its implications (see our response above).

Thank you again for your constructive feedback, which has helped us enhance the scope and rigor of our study.

Data and methodology:

My main concern with the data used for this study is the choice of ancient DNA samples used to estimate the proportion of Anatolian ancestry. There is not enough information in the main text or method to know which samples were used, where they came from, their dating, or their genomic coverage.

We have re-analyzed our empirical ancient DNA data from the most up to date AADR database (v62.0). Per the reviewer's request we have added a supplementary data file containing the qpAdm results for all 1531 analyzed individuals (pre-filtering) (Supplementary Data File 2) and a metadata file for each individual (pre-filtering) (Supplementary Data File 1). The metadata file provides detailed information for all individuals tested (pre-filtering), including the genetic ID, sample ID, qpAdm analysis ID, publication source, 1240k SNP coverage, data type (capture or shotgun), mean BP date, molecular sex, and site latitude and longitude. We have also provided detailed information on our filtering procedure following guidelines by Patterson et al. 2022. These files ensure transparency and allow readers to assess the dataset more comprehensively.

The version of the AADR used should at least be mentioned as it is regularly updated and may have already changed since the article was submitted.

The AADR database version was initially v52.2 but has been updated to v62.0 for our reanalysis. While we originally included the version number of the AADR in the methods section of the manuscript, we neglected to put it in the results section of the manuscript as well. The version number is now present in both the results section and the methods section of the manuscript. We thank the reviewer for pointing this out.

The selection of samples "that represent populations competing with and after agricultural expansion but before subsequent steppe expansion", meaning several thousand years, is surprising enough to quantify a process that occurs throughout along the progression of neolithization. It would have been more appropriate to select only the samples corresponding to the early neolithic period along the diffusion route.

We appreciate the reviewer's concern about using samples spanning several thousand years to quantify a process that unfolds over the course of Neolithic expansion. To address this, we re-analyzed the empirical data by separately examining samples from the Early,

Middle, and Late Neolithic periods and re-estimating the learning rate for each. Remarkably, we found very similar learning rates across all periods, consistently around 0.1% (see Fig. 6 and Supplementary Fig. 5j-l), which strengthens our conclusions.

Additionally, we observed no significant difference in the slope of the empirically observed EF ancestry cline between the Early and Middle Neolithic periods ($p > 0.05$; Supplementary Fig. 10), although there is a significant drop in the intercept at the Late Neolithic period ($p < 0.001$). Our simulations are consistent with this finding, showing that the simulated ancestry cline remains relatively stable over several hundreds of years after the expansion, despite continuous migration. This suggests that our learning rate estimates are robust to sample age and are not influenced by Middle Neolithic events such as the resurgence of WHG ancestry (see also our response to reviewer 1).

See lines 427-437 in the Results:

“After confirming that our learning rate estimate is robust to a variety of landscape and expansion models, we next evaluated how the time period of the ancient individuals used in the fitting process may affect our conclusions. We separated the 618 ancient individuals into bins corresponding to the Early, Middle and Late Neolithic. We repeated our learning rate estimation with each subset of ancient individuals and found consistently low estimates regardless of time period: 0.00069 [0.00067, 0.00071] for Early, 0.0011 [0.001078, 0.001121] for Middle, and 0.001339 [0.001316, 0.001363] for Late Neolithic (See Fig. 6; Supplementary Fig. 5j-l). A linear regression on the ancient individuals from each time period also suggests there is no significant difference in the slope of the ancestry cline between the Early and Middle Neolithic ($p = 0.0636$) and no significant difference when comparing Early to Late ($p = 0.0786$). However, there is a significant drop in the intercept from Early to Late ($p < 0.001$) (Supplementary Fig. 10).”

(and lines 582-592 in Discussion):

“We also acknowledge that besides geographic variability, temporal variability of the ancestry proportions in the ancient individuals (for example, the WHG resurgence observed during the Middle and Late Neolithic⁵⁹) may also impact our estimated learning rate. In our estimation procedure, we assume that analyzed individuals are representative for a time period immediately after the conclusion of the farming expansion. To evaluate the robustness to this assumption, we split the dataset of ancient individuals into categories by age, corresponding to Early, Middle, and Late Neolithic (Supplementary Fig. 10). We then refit our complex landscape map simulation data to the ancient data from each time period. We again find little variability in the estimated learning rate with estimates all corresponding to approximately 1 in 1000 farmers participating in cultural transmission. This indicates that the WHG resurgence observed during the Middle and Late Neolithic⁵⁹ does not significantly impact our general conclusions.”

I have other concerns regarding the choice of some model parameters. For example, using a generation time of 30 years, instead of 25 years which is commonly used for prehistoric

populations or the difference in population density between HG and farmers (can be quite different depending on the environment and landscape)... I am also wondering if the parameters should be the same between HG and farmers, for example the parameters governing the spatial distribution of each of these two groups will clearly be different and dependent on different factors related to the needs of the agriculture or hunting. Thus, the three main dynamics listed in line 122 for the different groups seem appropriate for predominantly sedentary farmers but perhaps not for seasonally mobile HGs...

Thank you for raising these concerns regarding our model parameters. Regarding generation time, we use this parameter solely for translating estimates from previous studies, which report parameters on a per-generation basis, into our agent-based model, which operates on a yearly time step. Our model does not inherently require a generation time, as it progresses year by year, and we incorporate a realistic age-at-death distribution to naturally account for generational turnover. In response to the reviewer's comment, we have updated the generation time to 25 years when converting between per-generation and per-year rates.

We also want to emphasize that our model accounts for differences in carrying capacity between farmers and hunter-gatherers, reflecting the generally higher population density sustainable by farming communities (Ammerman & Cavalli-Sforza, 1984). This assumption is based on the modeling approach by Currat and Excoffier, who selected this hunter-gatherer carrying capacity value (Currat & Excoffier, 2005) based on two previous papers (Alroy, 2001; Steele et al., 1998), and while it is an estimate, our results remain robust to changes in this parameter. For example, reducing the population density ratio from 20:1 (farmers to HGs) to 2:1 did not alter our main findings.

Additionally, in our mathematical (reaction-diffusion) model, we explored the effect of an increased diffusion constant for HGs, to reflect their potentially more mobile lifestyle. Even a ten times larger diffusion constant did not significantly impact the resulting ancestry patterns, reinforcing the robustness of our conclusions to variations in model parameters.

The discussion around the implication of its results for the interpretation of archaeological contexts is almost absent from this work even though it would allow this approach to be recontextualized.

Thank you for this insightful comment. While we agree that exploring the implications of our results for interpreting archaeological contexts promises to provide additional depth, we must emphasize that a detailed archaeological interpretation lies outside the scope of our study. Our primary aim is to provide robust quantitative estimates of interaction rates, such as cultural learning and inter-group mating, which can serve as foundational data for interdisciplinary research.

To clarify this, we have revised the discussion to acknowledge the potential for future studies to build on our findings by incorporating more detailed archaeological data. For

example, regional variation in cultural transmission rates derived from our model could be linked to observed patterns in settlement density, material culture, or burial practices. By leaving this interpretative work to experts in archaeology, we hope to provide a framework that enables more precise integration of genetic and archaeological evidence in future research.

See lines 578-581:

“Future study could also delve into the potential for regional variation of cultural transmission rates and how it could be linked to observed local patterns derived from archaeological studies, integrating our quantitative estimates of interaction rates with local genetic and archeological interpretations.”

To summarize, the idea of combining genomic data and modeling is very interesting but it is not understandable that the authors chose to simplify their model so much and ignore the existence of the two neolithization currents. Under these conditions, it is justified to question the relevance of the conclusions and how they should be interpreted. I therefore recommend a major revision of this work with the consideration of regional differences or the application of the model to only one of the two diffusion trends.

We thank the reviewer for their thoughtful comments on integrating genomic data and modeling and for highlighting the importance of considering the two neolithization routes. This feedback is invaluable, and we agree that incorporating regional differences can significantly enhance the interpretability and relevance of our results.

In the initial model, we opted for simplification to focus on broader trends while minimizing the risk of overfitting the data to potentially uncertain regional parameters. However, we acknowledge that this approach may have overlooked key aspects of the neolithization process. In response to the reviewer's feedback, we have revised our work to explicitly incorporate the two major routes of neolithization. Details of the new approach are outlined in our response above.

Additionally, we have expanded the Discussion to emphasize how future studies might build on our model by integrating more detailed archaeological data to investigate specific regional dynamics. Nevertheless, we note that such a detailed regional study lies beyond the scope of the current work.

We appreciate the reviewer's suggestions and believe these adjustments have resulted in a more robust and relevant analysis with a clearer scope.

Reviewer #3

(Remarks to the Authors, Responses from the Authors, and Relevant Excerpts from the Revised Manuscript):

LaPolice et al. present a novel work by combining, for the first time, a model that co-analyzes genetic ancestry and the speed of advance in the context of the Neolithization of the European continent. This represents a significant advance compared to previous studies on this phenomenon.

My background as a paleogeneticist leads me to focus my reviews on the use of concepts and empirical data drawn from archaeology and genetics, leaving aside modeling aspects. In this regard, I believe there are elements that need further explanation or discussion and that should be improved in the manuscript, as well as the inclusion of a section that addresses the limitations of the model.

We sincerely thank the reviewer for their thoughtful comments and for recognizing the novelty and significance of our work in combining genetic ancestry and the speed of advance to study the Neolithization of Europe. We greatly appreciate the reviewer's expertise as a paleogeneticist and are encouraged by their acknowledgment of our contributions to advancing this field.

We also greatly value the feedback provided and take the reviewer's comments into consideration to improve our manuscript.

Important Points:

1) Neolithization Routes

Although I understand the complexity of the topic, I missed a distinction between Mediterranean and Central European dispersion for a more complete Neolithization model. At least discussing this aspect would be helpful, and if this distinction was not made explaining why.

We thank the reviewer for highlighting the importance of regional variability in the Neolithic expansion, particularly the distinction between Mediterranean and Central European dispersion. To address this concern, we have modified our model to differentiate between the continental and Mediterranean/Coastal routes of Neolithic expansion. To this end, we developed new multiple-route models. We increased movement speed along waterways based on parameters from literature (Davison et al., 2006; Fort & Pérez-Losada, 2024) and treated mountainous regions as uninhabitable barriers, resulting in distinct expansion pathways. This adaptation allows us to model separate corridors, with faster dispersal along the Mediterranean coast and the Rhine/Danube corridor, and slower expansion in northern regions.

Despite the inclusion of this increased spatial heterogeneity in expansion speed, our central conclusion regarding the low level of cultural interaction between farmers and hunter-gatherers remains consistent, as the estimated learning rates between groups are consistently low (0.001124 [0.00111, 0.001137], 0.000938 [0.000927, 0.000949], 0.000991 [0.000978, 0.001003]) under these revised models compared to 0.00101 [0.000998, 0.001023] under our standard complex model.

For a more detailed analysis, we divided the individual ancestry estimates into two groups based on latitude: southern individuals associated with the Mediterranean route (latitude $\leq 45^\circ$) and northern individuals linked to the Rhine/Danube corridor (latitude $> 45^\circ$). When fitting our model to these groups separately, we found that the estimated learning rate along the Mediterranean route is slightly larger (0.001244 [0.001208, 0.001279]) than for the northern route (0.000914 [0.000901, 0.000927]). This finding contrasts with a recent study based on mitochondrial haplotype clines, which suggested that learning rates are consistent across both routes (Fort & Pérez-Losada, 2024). There are two possible explanations for this discrepancy. First, our analysis benefits from a larger sample size and the higher resolution afforded by genome-wide ancestry estimates, allowing us to detect subtler differences in learning rates that may be missed using mitochondrial haplotype data. Second, mitochondrial haplotypes are maternally inherited and reflect only female ancestry patterns. This raises the possibility of sex-specific cultural interaction patterns and movement rates. Although outside of the scope of our study, future work should aim to integrate autosomal and mitochondrial data into a unified framework to investigate potential sex-specific parameters.

In conclusion, while a more detailed analysis of the Neolithic expansion reveals subtle heterogeneity in cultural transmission rates, this variability does not alter our primary conclusion of low learning rates, estimated to be on the order of 1 in 1000. We have added new sections to the results and discussion to describe this approach and its findings.

Please see lines 400-426 in Results:

“However, prior archaeological studies have indicated that the agricultural expansion was not uniform but might have spread faster along a coastal Mediterranean route compared to a northerly route through the Balkans and into Central Europe^{44,67–69}. To assess how non-uniform expansion might have influenced our analysis, we conducted simulations under three different non-uniform models. The first model used the same parameters and maps as the uniform complex map expansion but adjusted the step size parameter to reflect directional differences in dispersal. Specifically, we reduced the step size in the North-South direction to 4 km per year and increased it in the East-West direction to 6 km per year, instead of the uniform step size of 5 km per year. Using the same maximum likelihood approach as before, we found a similar estimated learning rate of 0.001124 [0.00111, 0.001137] (Fig. 6; Supplementary Fig. 5e)..

To further investigate the effects of heterogeneous dispersal, we applied a model that increased the step size for individuals near the Mediterranean coast, based on speed

estimates from Fort and Pérez-Losada (2024)⁶⁹. In addition, mountainous regions were defined as barriers to dispersal. This model yielded results that were again in line with previous analyses, with an estimated learning rate of 0.000938 [0.000927, 0.000949] (Fig. 6; Supplementary Fig. 5f). To introduce further heterogeneity, we incorporated parameters from Davison et al., (2006)⁴⁴ which suggest that farming expansions were approximately 5x faster along the Danube-Rhine Corridor and 10x faster along the Mediterranean Coast. The estimated learning rate again remained consistent with earlier findings, yielding a value of 0.000991 [0.000978, 0.001003] (Fig. 6; Supplementary Fig. 5g). All simulation maps with expansion corridors can be found in Supplementary Fig. 8 and Supplementary Fig. 9. Finally, we allowed for regional variation in learning rate by dividing individuals into a northern and a southern group based on a latitude of 45°. Separate learning rates were estimated for each group, revealing a slightly higher rate in the South compared to the North (0.001244 [0.001208, 0.001279], and 0.000914 [0.000901, 0.000927] respectively) (Fig. 6; Supplementary Fig. 5h-i). Nonetheless, our overall conclusion of a cultural learning rate on the order of 0.1% remains robust and independent of the expansion route.”

We have added a note in the discussion to acknowledge the limitations of our study regarding regional variation and to outline potential directions for future research.

Please see lines 563-581 in the Discussion:

“Importantly, estimated learning rates remained consistently low (Fig. 6), irrespective of whether we assumed a square model with uniform dispersal, a complex geographic model with uniform dispersal, a model with faster latitudinal expansion or models with independent Mediterranean and inland expansion routes. This illustrates the robustness of our findings to spatial dispersal heterogeneity. Interestingly, when estimating learning rates separately for Southern and Northern European individuals, we found that the estimated rate for the South was slightly higher than for the North. This finding contrasts with a recent study based on mitochondrial haplotype clines, which reported uniform learning rates across both Mediterranean and inland routes⁶⁹. Two factors may account for this discrepancy. First, our analysis utilized a larger sample size and the finer resolution of genome-wide ancestry estimates, which likely allowed us to detect subtle differences in learning rates that mitochondrial data may not capture. Second, because mitochondrial haplotypes are maternally inherited, they reflect only female ancestry patterns, potentially missing sex-specific cultural interactions and movement dynamics. Future studies should aim to integrate autosomal and mitochondrial data into a unified framework to better understand the influence of sex-specific factors on cultural transmission and migration dynamics. Future study could also delve into the potential for regional variation of cultural transmission rates and how it could be linked to observed local patterns derived from archaeological studies, integrating our quantitative estimates of interaction rates with local genetic and archeological interpretations.”

This point is important, especially when the discussion mentions in lines 444-449: "In our model's assessment of non-uniform expansion, conducting additional simulations with a decreased step size of farmers indeed demonstrated that a slower expansion speed leads

to greater HG ancestry retention in the population following the expansion (Supplementary Fig. 9). This may explain why in northern regions where the farming expansion was presumably slower, e.g. due to the southwest Asian crop package underperforming in the colder and wetter climate, higher levels of HG ancestry are found 51."

This explanation fits well with the archaeological and genetic context in northern Europe. However, it would not be applicable to regions like the Iberian Peninsula, where the amount of hunter-gatherer ancestry is higher than in other areas of Europe, and the Neolithic arrived earlier due to maritime dispersion, which is faster.

We thank the reviewer for their insightful comment. We agree that the higher speed of farming expansion along the Mediterranean would predict lower HG ancestry retention, which might initially seem at odds with the relatively high HG levels observed in the Iberian Peninsula. However, the large geographic distance from Anatolia to Iberia provides more opportunities for interactions between indigenous HG populations and incoming farmers, potentially leading to a stronger dilution of farming ancestry and thus higher observed HG ancestry levels.

To investigate the interaction between these two factors, we first examined whether Iberian populations (i.e., Spain and Portugal) are outliers in the Ancestry vs. distance cline (Supplementary Figure 4). Our analysis indicates that the HG ancestry observed in the Iberian Peninsula aligns with the high travel distance between the farming origin and the Iberian Peninsula. Specifically, in relation to a linear relationship, Iberian populations do not appear as outliers but exhibit levels of HG ancestry consistent with expectations.

Next, we divided the dataset into southern and northern European individuals (using latitude 45 as the dividing line) and observed a larger estimated learning rate in the South compared to the North (as discussed in our previous response). This difference in learning rates may explain why the increased speed of migration along the Mediterranean route does not necessarily result in reduced HG ancestry compared to other regions in Europe.

2) Assortative Mating

I find it a term that implies too many social aspects that are difficult to prove in this case. Mating should not be understood as assortative if these individuals (hunter-gatherers and farmers) did not coincide in space/time. The survival of the last genetic hunter-gatherers has been demonstrated in the extremes of Europe, probably avoiding contact (Posth et al. 2023, Nature) or in forested areas or not adapted to agriculture (Rivollat et al. 2020, Science Advances; Tangermünde individual or Blätterhöhle individual). The choice emphasizes a social behavior/taboo when choosing a partner, but we do not know if these interactions between the two groups occurred simply because they did not occupy the same spaces. I do not consider it appropriate to use the term "assortative" for these reasons. Assortative or non-assortative is a preference but not finding people around is not a matter of choice.

We appreciate the reviewer's insightful feedback and fully understand the concern that the term "Assortative Mating" implies social behaviors or preferences that are not supported by the data or context of our study.

In response, we have replaced "Assortative Mating" with "Within-Group Mating" and "Non-assortative Mating" with "Between-Group Mating" throughout the manuscript. We accordingly changed the title of our manuscript to "*Modeling the European Neolithic Expansion: Predominant Within-Group Mating and Limited Cultural Transmission*". This revised terminology is intentionally more neutral and avoids making assumptions about the underlying reasons for the observed mating patterns. We acknowledge that these patterns could arise due to a lack of spatial overlap, as the reviewer suggests, rather than implying any social preferences or behaviors.

We believe this adjustment directly addresses the reviewer's concern and ensures that the terminology remains agnostic about the underlying drivers of the mating choices while still accurately describing the modeled processes.

3) Cultural Transmission and Learning Rate

It is said that "cultural transmission allows for the persistence of indigenous HG ancestry after the expansion." I believe it is necessary to point out that if individuals occupy different environments, this transmission does not occur, although hunter-gatherer genetic ancestry may persist. The examples I could cite are the same as I provided for assortative mating. I ask the authors to mention these cases in the discussion, as the archaeological context is important to assess the model's viability.

We agree that cultural transmission depends on interactions between populations and that such transmission may not occur if individuals occupy distinct environments or geographic regions. In our model, we specifically focus on HG ancestry retained within farming individuals, which inherently requires some form of cultural transmission. However, we fully acknowledge the reviewer's point that such transmission is unlikely in cases where populations remain isolated due to environmental or geographical factors.

Nonetheless, we propose that our model serves as a critical null framework, offering key estimates of cultural transmission rates and establishing broad-scale expectations for ancestry patterns. Developing such a broad-scale model is critical for detecting significant local deviations, which can then guide more detailed archaeological investigations into the conditions under which regional variations in cultural transmission occur.

To address the reviewer's concern, we now incorporate examples that highlight regional survival of hunter-gatherer populations at the extremes of Europe and the persistence of hunter-gatherer genetic contributions in Central Europe during the Neolithic, which may have been due to prolonged coexistence and minimal gene flow between farmers and hunter-gatherers in certain regions. We further clarify that our modeling focuses on the initial farming expansion (lines 535-549):

“Notably, aDNA studies from Atlantic France and Scandinavia show no evidence of farmer-associated ancestry in hunter-gatherers who overlapped in time with neighboring Neolithic farmers^{75,76}, corroborating our finding of low between-group mating. Additionally, in some regions, evidence suggests that foragers and farmers coexisted for centuries with minimal gene flow. In northeastern France and Germany, some late Neolithic farmers display unexpectedly high levels of hunter-gatherer ancestry (e.g., Mont-Aimé: 50–63%, Tangermünde: ~63%, Blätterhöhle: up to 85%), reflecting prolonged coexistence and limited genetic integration^{61,77–79}. Similarly, in the Iron Gates region, hunter-gatherers and early farmers lived side by side for centuries, exhibiting only minimal admixture despite extended contact⁸⁰.

These patterns suggest persistent but low levels of asymmetric admixture^{61,81} and reveal long-term demographic dynamics, such as the survival of regional hunter-gatherer groups and potential cultural reversions by farmers to hunter-gatherer subsistence strategies. While our model focuses on the initial farming expansion, integrating these long-term dynamics into our model could offer valuable insights into the diversity and variability of local interactions and population structures following the initial farming expansion.”

4) Ancestry Modeling

The explanation of data filtering is adequate, but it would be more illustrative if accompanied by a table with the data used and individuals grouped by geographical proximity. It is said that it filters between 5000 and 8000 BP. However, in these chronologies of 5000 BP in Central Europe, steppe ancestry is already found. I am not saying that the analysis is wrong, but some individual could have been included. To avoid these doubts, the manuscript would benefit greatly from a supplementary table with the samples used and a PCA of Western Eurasia where the individuals are seen in the genetic diversity pool representing the ancestry of the first farmers.

qpADM: The set of outgroups used in the article would be considered obsolete, although not incorrect. Currently, we avoid modern outgroups because we already have enough ancient data to represent all ancestries required in a distal model. Although I do not believe this significantly changes their estimates, I recommend following Patterson et al. 2023, which presents a set of outgroups very close to the sources which helps to narrow down the standard errors from admixture proportions. If in the future the authors wish to investigate bell beaker movements as well (as the manuscript seems to point out), this qpAdm model would be compatible. Another alternative would be to work with supervised ADMIXTURE.

We thank the reviewer for their thoughtful comments and suggestions regarding ancestry modeling and data filtering. To address the reviewer’s recommendation, we have added a supplementary data file containing the qpAdm results for all 1531 analyzed individuals (pre-filtering) (Supplementary Data File 2) and a metadata file for each individual (pre-filtering) (Supplementary Data File 1). The metadata file provides detailed information for

all individuals tested (pre-filtering), including the genetic ID, sample ID, qpAdm analysis ID, publication source, 1240k SNP coverage, data type (capture or shotgun), mean BP date, molecular sex, and site latitude and longitude. We have also provided detailed information on our filtering procedure following guidelines by Patterson et al. 2022. These files ensure transparency and allow readers to assess the dataset more comprehensively. We also have included Supplementary Figures 4, 15 and 16 which visually depict more information about the ancient individuals selected.

Regarding the concern about Steppe ancestry and updating our qpAdm modeling, we have now followed the protocol outlined in Patterson et al. (2022) as suggested by the reviewer. Further, we restricted downstream analyses to individuals with less than 5% estimated Steppe ancestry and with qpAdm ancestry estimates showing standard errors below 0.02, resulting in a final sample size of 618 individuals. This ensures that our dataset remains accurate and focused on the ancestry of early farmers without confounding signals from later Steppe-related admixture.

Additionally, we generated a PCA of the analyzed sample, including the qpAdm source populations, which is now included as Supplementary Figure 16. This PCA illustrates that the individuals analyzed predominantly fall along the cline between Western Hunter-Gatherer (WHG) and Early European Farmer (EEF) ancestry, aligning with expectations based on our study's focus.

We appreciate the reviewer's suggestion to explore updated outgroups for qpAdm modeling, as outlined in Patterson et al. (2022). While our results remain consistent with expectations, adopting this updated set of outgroups has helped refine our estimates, providing narrower standard errors. This protocol also ensures compatibility with future studies, such as investigations into Bell Beaker movements, as mentioned by the reviewer.

Line 97: Who are these 911 individuals and what is the SNP cutoff used to estimate ancestry proportions? With lower SNP coverage, there is less chance of detecting small components. Also, I doubt there are 911 early Neolithic individuals. What other Neolithics are being considered? Would it be interesting to analyze only the early Neolithics from each geographic area? In the late Neolithic of many places, we see an increase in hunter-gatherer ancestry, interpreted as the final incorporation of the few remaining hunter-gatherers. If their model detects this increase in areas where only middle or late Neolithics have been sampled, but not early Neolithics, could their inferences vary? It is necessary to report specific Neolithics individuals used, their coverage, and their C14-group cultural dates, and discuss whether using one or the other can influence certain values (e.g., advance speed).

Under stricter filtering criteria, we now include 618 individuals in our analysis. Instead of using a specific SNP cutoff, we have implemented a threshold on the standard error of the estimated Early Farming (EF) ancestry proportion ($SE < 0.022$; following Patterson et al.

2022). This standard error is calculated using genomic bootstrapping, ensuring that we only include samples where even small ancestry components can be reliably detected. Furthermore, while our primary analysis considers all Neolithic individuals, we have now separated the samples into Early, Middle, and Late Neolithic groups. Notably, there is no significant difference in the EF ancestry cline between the Early and Middle Neolithic groups ($p > 0.05$), and the estimated learning rates across all three epochs remain consistent at approximately 0.1% (ranging from 0.00069 [0.00067, 0.00071] to 0.001339 [0.001316, 0.001363]). This indicates that the WHG resurgence observed during the Middle and Late Neolithic does not significantly impact our general conclusions. We have incorporated this new analysis into the results and discussion sections.

See lines 427-437 in the Results:

“After confirming that our learning rate estimate is robust to a variety of landscape and expansion models, we next evaluated how the time period of the ancient individuals used in the fitting process may affect our conclusions. We separated the 618 ancient individuals into bins corresponding to the Early, Middle and Late Neolithic. We repeated our learning rate estimation with each subset of ancient individuals and found consistently low estimates regardless of time period: 0.00069 [0.00067, 0.00071] for Early, 0.0011 [0.001078, 0.001121] for Middle, and 0.001339 [0.001316, 0.001363] for Late Neolithic (See Fig. 6; Supplementary Fig. 5j-l). A linear regression on the ancient individuals from each time period also suggests there is no significant difference in the slope of the ancestry cline between the Early and Middle Neolithic ($p = 0.0636$) and no significant difference when comparing Early to Late ($p = 0.0786$). However, there is a significant drop in the intercept from Early to Late ($p < 0.001$) (Supplementary Fig. 10).”

(and lines 582-592 in Discussion):

“We also acknowledge that besides geographic variability, temporal variability of the ancestry proportions in the ancient individuals (for example, the WHG resurgence observed during the Middle and Late Neolithic⁵⁹) may also impact our estimated learning rate. In our estimation procedure, we assume that analyzed individuals are representative for a time period immediately after the conclusion of the farming expansion. To evaluate the robustness to this assumption, we split the dataset of ancient individuals into categories by age, corresponding to Early, Middle, and Late Neolithic (Supplementary Fig. 10). We then refit our complex landscape map simulation data to the ancient data from each time period. We again find little variability in the estimated learning rate with estimates all corresponding to approximately 1 in 1000 farmers participating in cultural transmission. This indicates that the WHG resurgence observed during the Middle and Late Neolithic⁵⁹ does not significantly impact our general conclusions.”

To address the reviewer’s request for greater transparency regarding the specific individuals included, we have added a supplementary data file containing the qpAdm results for all 1531 analyzed individuals (pre-filtering) (Supplementary Data File 2) and a metadata file for each individual (pre-filtering) (Supplementary Data File 1). The metadata

file provides detailed information for all individuals tested (pre-filtering), including the genetic ID, sample ID, qpAdm analysis ID, publication source, 1240k SNP coverage, data type (capture or shotgun), mean BP date, molecular sex, and site latitude and longitude. We have also provided detailed information on our filtering procedure following guidelines by Patterson et al. 2022. These files ensure transparency and allow readers to assess the dataset more comprehensively.

Additionally, we have included two new supplementary figures: one showing the dates and geographic regions of the individuals (Supplementary Figures 4 and 15) and another presenting a PCA that visualizes the ancestry of the included individuals.

With lower SNP coverage, there is less chance of detecting small components.

Following Patterson, we used a 0.022 standard error cutoff for each individual to only retain ancestry estimates with small standard errors. This guarantees that we can detect even small ancestry components. Moreover, we include the uncertainty in ancestry estimation in our inference approach when computing the log-likelihood of each cultural transmission rate.

Line 292: Binning strategy: verify with PCA if they can be considered a cluster.

We thank the reviewer for this valid comment regarding the binning strategy. In response, we have revised our approach and no longer bin samples with the same geographic coordinates together. Instead, we now use qpAdm to estimate the EF ancestry fraction for each individual sample directly. This adjustment avoids any potential biases introduced by binning.

Nonetheless, we have included a PCA in Supplementary Figure 16 to enhance transparency about the ancient individuals used in fitting. This PCA illustrates that the individuals analyzed predominantly fall along the cline between Western Hunter-Gatherer (WHG) and Early European Farmer (EEF) ancestry.

Line 294: "we only selected sites with more than 2 samples per site." The number of samples does not matter, but their coverage does. If the coverage is low, a component can be lost in the qpAdm model.

We thank the reviewer for highlighting the importance of coverage over the number of samples per site. As noted above, we have now completely revised our approach. Rather than estimating ancestry proportions for sites with more than two samples, we now calculate ancestry proportions for each individual sample. To ensure the reliability of these estimates, we apply a filter based on the bootstrap-estimated standard error of the ancestry component ($SE < 0.022$). This approach ensures that only individuals with well-estimated ancestry fractions are included in our analysis, improving the accuracy and robustness of our results.

Line 296: It would be useful to mention the sites or contexts of these outliers with very low Anatolian ancestry. Are they matching with the examples I provided above?

In our updated qpAdm analysis, we identified four individuals who passed our filtering threshold but retained low levels of EF ancestry. These include one individual from Spain and three from Germany. The individual from Spain (NEO646.SG_8413_BP_Spain) is from the site of El Mazo and represents one of the earliest known instances of Balkan HG ancestry arriving in Iberia (Olalde et al., 2019; Villalba-Mouco et al., 2019).

The three German individuals—two from Ostorf-Tannenwerder (OST003 and OST002) and one from Blätterhöhle (I1565)—provide critical evidence of late hunter-gatherer genetic signatures and lifestyles persisting within Neolithic cultural contexts. At Blätterhöhle, individual I1565 (Bla8) has previously been identified as associated with a hunter-gatherer–fisher lifestyle, as evidenced by stable isotope analysis (Lipson et al., 2017). The site is known for its coexistence of farmers and hunter-gatherers to a relatively late date.

Similarly, the inhabitants of Ostorf-Tannenwerder represent one of the last occurrences of high levels of hunter-gatherer-related ancestry prior to the European Bronze Age (Fernandes et al., 2015; Posth et al., 2023). While these individuals lived in a Funnel Beaker cultural context and adopted certain Neolithic cultural elements, their subsistence strategies remained consistent with a hunter-gatherer diet (Fernandes et al., 2015; Posth et al., 2023).

We are grateful to the reviewer for highlighting these examples of late-coexisting HGs. These not only provide an explanation for the outliers observed in our dataset but also emphasize the importance of studying such outliers to better understand the complex dynamics of the Neolithic transition.

We have added a Supplementary Note (Supplementary Note 2) detailing these outliers and refer to it in the Results section:

Lines 334-338 in Results:

“However, we note four outlier individuals, one individual from Spain and three from Germany, with very low EF ancestry (<25%) that do not follow this trend. Each of these four outliers was found with paleoanthropological characteristics that place them within either mixed or hunter-gatherer contexts^{56–60}. See Supplementary Note 2 for a detailed description of the archaeological context for these outliers.”

Newly added Supplementary Note 2 Reads:

“In our qpAdm¹ analysis we note four individuals—one individual from Spain and three from Germany—passed our filtering threshold but still retained low-levels of EF ancestry (outlier points on Fig. 5b and Supplementary Fig. 4). In Supplementary Fig. 4, points are colored by geographic region. The outlier sample from Spain (NEO646.SG_8413_BP_Spain) is from the site of El Mazo and is one of the earliest instances of Balkan HG ancestry arriving in Iberia^{2,3}. The three German individuals—two from Ostorf-Tannenwerder (OST003 and OST002), and one from Blätterhöhle (I1565)—represent crucial evidence of late hunter-gatherer genetic signatures and lifestyles persisting within Neolithic cultural contexts. Blätterhöhle has been shown to have featured a combination of farmer and hunter-gatherer occupation to a relatively late date with individual I1565 (Bla8) previously identified from stable isotopes to be associated with a hunter-gatherer–fisher lifestyle⁴. Inhabitants of Ostorf-Tannenwerder have been previously identified to be one of the last occurrences of high levels of hunter-gatherer-related ancestries prior to the European Bronze Age⁵. Whilst they are in a Funnel Beaker context and adopted some Neolithic cultural elements, their subsistence strategy was consistent with a hunter-gatherer diet^{5,6}.”

Line 320: If considering qpAdm errors, I strongly recommend using (Patterson 2021; Nature).

We have reanalyzed our data following the recommendation to use the approach outlined in Patterson et al. (2022, Nature). We thank the reviewer for this valuable suggestion, which has enabled us to refine and consider qpAdm error estimates at the individual sample level rather than at the per-site level.

5) Mortality Rates

Reference number 34. I am sure there is a better reference for Neolithic mortality than a report from a final Neolithic burial cave in Greece. There are many references from particular studies. If it is not explained why this particular one is chosen, it would be better to look for a more generic one. They can rely on Samanta-Cox et al. 2023, which states "Ages were determined on the basis of the average of the age range reported for each individual in their original publications," using the same AADR dataset. Through these age-at-death calculations, they can obtain a more suitable result for a Neolithic average in Europe, although they must consider that there is always a sampling, excavation, and analysis bias. For example, the absence of infant individuals may be due to a different ritual or much lower preservation.

We appreciate the reviewer’s suggestion to use the age-at-death distribution from the AADR dataset. However, while this dataset provides valuable information on ancient individuals, it is important to note that the AADR dataset likely reflects considerable sampling bias, as researchers preferentially select individuals most likely to yield high-

quality DNA. This bias could explain an overrepresentation of older individuals with better-preserved skeletal elements rather than an even sampling of age-at-death distributions.

To address the reviewer's concerns, we have evaluated the robustness of our results to the shape of the age-based mortality curve. We conducted additional simulations using a new mortality curve based on (Eshed et al., 2004). This new curve is based on a substantially larger dataset (sample size 161 vs. 262) with lower mortality probability at most ages and more longevity within the population. Despite this significant change in the mortality curve, our results remained consistent 0.00101 [0.000998, 0.001023] vs. 0.000711 [0.000701, 0.000721] under the alternate mortality curve. The negligible effect of this updated mortality curve on our conclusions reinforces our confidence that uncertainty in age-specific mortality rates does not influence the overall findings.

To present this new analysis, we have added the following supplementary materials:

- Supplementary Figure 7: Shows the new mortality curve alongside the previous version for comparison.
- Figure 6: Compares the estimates from each model, including results based on the updated mortality curve.
- Supplementary Table 2: Provides detailed data on the updated alternative mortality curve.

We believe these additions address the reviewer's concern and further support the robustness of our conclusions.

Supplementary Table 1: it shows that the highest survival probability is at the age of 0 years. I do not believe is correct. Until the first years of life, survival probabilities are low in all non-industrialized population groups.

We agree with the reviewer that child mortality is high in pre-industrialized populations. In fact, both the primary and alternative mortality curves used in our simulations reflect this pattern: mortality probability is high during the first years of life, drops to a minimum around age 10, and then increases again at older ages.

Regarding the reviewer's comment, we believe there may be some confusion between the per-age mortality rate and the survival probability presented in the last column of Supplementary Table S1. The survival probability indicates the likelihood of an individual surviving up to a specific age. As expected, this value is higher for younger ages and decreases over time because, mathematically, the probability of surviving to an older age (e.g., 10 years) represents only a subset of individuals who survived to earlier ages (e.g., 1, 2, 3, 4, 5 years... etc). Each successive survival probability reflects the cumulative effect of earlier age-specific mortality rates. However, the age-specific mortality rate—representing the probability at which individuals in a specific age group die—is still high

during early childhood and drops as children grow older, consistent with the pattern expected for non-industrialized populations, as the reviewer suggested.

To clarify this further, we have added a new supplementary figure (Supplementary Figure 7) comparing the two empirical mortality curves. This figure highlights the elevated child mortality rates captured by both curves and provides a clearer visualization of the patterns discussed.

Minor Points:

Line 37: "It is believed that early European agriculture originated in southwest Asia 17–19 and first expanded into Europe from northwestern Anatolia 17."

Agriculture was developed earlier/or simultaneously in other areas, and genetically they show independent origins. I recommend mentioning this; a good summary paper can be Feldman et al. 2019.

We have updated the sentence to acknowledge the independent origins of farming in Southwest Asia:

"Agriculture emerged independently at several centers in the Fertile Crescent, each with genetically distinct origins⁶. From there, it spread into Europe via central Anatolia. Notably, Anatolia was more than just a corridor for the movement of early farmers; it was a region where local hunter-gatherers transitioned to an agricultural way of life²³."

Line 79: You mention the word "flora," but the referenced article refers to the "vegetational landscape." I would change "flora" to "paleoenvironmental data" or "vegetational landscape."

This line (now lines 94-96) has been changed based on the reviewer's suggestion and now reads:

"In addition to mathematical modeling of the Neolithic expansion^{30–37}, other studies have gathered novel insights from investigating settlements^{38–42}, geographical factors⁴³, language⁸, ancient climate conditions⁴⁴, and paleoenvironmental data⁴⁵."

Line 85: I would say HG in general. Paleolithic HG are only Pleistocene HG, but the Holocene HG (Epipaleolithic and Mesolithic) also contributed.

This line has been changed based on the reviewer's suggestion and now reads:

"Early work based only on mitochondrial data lent more support for a significant local HG contribution to modern European populations thereby indicating a primarily cultural transmission."

Line 88: Western Anatolia is more accurate.

We have changed these lines (104-107) and they now read:

“Ever-growing sample sizes, and especially whole-genome sequences, have revealed in greater detail the significant temporal and geographical ancestry shifts co-occurring with the spread of the Neolithic cultural package^{6,7,46,52,53.}”

Lines 90-92: "limitation of DNA is that it does not directly describe or indicate behavior, culture, or specific mechanisms of expansion. As such, an interdisciplinary approach linking behavioral mechanisms with genetic data is required to fully understand the Neolithic farming expansion."

Clarify that archaeological contexts do this. Interdisciplinarity already exists in disciplines such as archaeogenetics.

We agree with the reviewer’s comment that interdisciplinarity already exists in fields like archaeogenetics. However, as a relatively young and evolving field, archaeogenetics continues to benefit from new approaches that enhance its scope. Through our work, we aim to contribute to this development by incorporating simulation-based behavioral modeling into archaeological and genetic contexts. Our goal is to deepen the understanding of the mechanisms driving the Neolithic farming expansion by linking behavior with genetic and archaeological evidence.

To reflect this more clearly, we have revised the relevant sentence in the main text. The updated text now reads (lines 109-116):

“As such, understanding the Neolithic farming expansion requires an interdisciplinary approach that integrates behavioral mechanisms with genetic data.

Here, we build on the evolving field of archaeogenetics by incorporating behavioral simulations grounded in both archaeological and genetic evidence. Our aim is to create a model that identifies cultural transmission parameters consistent with both archaeological and genetic data, enabling a more detailed understanding of the mechanisms responsible for this pivotal societal change.”

We also have added lines 496-506 in the Discussion which discuss the interdisciplinarity of our approach:

“While archaeological research has been instrumental in demonstrating the profound changes associated with the Neolithic transition, including the scale and timing of this transformation, it left unresolved questions about the specific mechanisms driving these changes. For example, archaeological evidence alone has not decisively revealed the relative contributions of migration versus the diffusion of ideas to cultural shifts²¹. Genetic

data complements this by highlighting processes such as mass migration^{52,71} but similarly cannot fully quantify the interplay between cultural and demographic factors. Our interdisciplinary approach bridges this gap by using mathematical and simulation-based modeling to integrate insights from both archaeology and genetics. This enables us to explore cultural and behavioral mechanisms that are not directly observable in the material record, offering a more nuanced understanding of the Neolithic expansion as a transformative process in human history.”

Line 120: It is necessary to introduce the term WHG and not just talk about HG, since in Anatolia there would be other HG that are not the HG that you can infer with your qpAdm model. However, this does not mean they did not interact.

We thank the reviewer for pointing out the need to clarify the term HG in this context. We acknowledge that the hunter-gatherers in Anatolia may include groups with ancestries distinct from those inferred with our qpAdm model. To address this, we have revised the text to specifically refer to HG with WHG ancestry where applicable. Further, we now introduced early in the manuscript the term Western Hunter Gatherer (WHG).

Lines 119-122 in the introduction:

“Throughout, in describing the groups associated with the Neolithic shift we use the term Early Farmer (EF) as representative of the genetic ancestry associated with this spread of early farming and Western Hunter-Gatherer (WHG) representing the local European Mesolithic ancestry.”

Reference 21: There is no genetic reference that reports more updated data on lower population density? Maybe based on heterozygosity or background relatedness with ROH, for example.

To the best of our knowledge, there are no suitable genetic estimates of Neolithic population sizes. Recent studies have attempted to estimate population sizes from runs of homozygosity (ROH) in Yamnaya (Lazaridis et al., 2024) and Iron Age populations (Fournier et al., 2023), but these estimates are not necessarily reflective of Neolithic population sizes. Other studies provide a population size estimate for the ancestral Anatolian population (e.g., Marchi et al., 2022), but this does not reflect the increased density due to farming. Moreover, those estimates provide overall counts and not per-area densities that we need to initialize our simulations.

Most importantly, genetic approaches typically estimate the effective population size (N_e) rather than the actual census size. Effective population size is usually smaller than the census population size due to factors such as variance in reproductive success and the complexities introduced by spatial genetic structure. While N_e is a valuable parameter for evolutionary studies, it is not directly applicable to our purposes, as we require the census

population density—the actual number of individuals in the population per unit of area—for our simulations.

Line 130: Saying "post-Neolithic" is incorrect. It should be "Neolithic." I would avoid the term "post-Neolithic" because it is unclear whether it refers to another chronological period or simply to modern data.

We thank the reviewer for pointing out the ambiguity in using the term "post-Neolithic." At the reviewer's suggestion this line has been changed (now lines 156-158) and now reads:

“Published aDNA evidence suggests that Neolithic Europe exhibited significant EF ancestry, exceeding 75% across all of Europe⁴⁶.”

Line 140: The same. What does "post-Neolithic" refer to? Be careful, there are other migrations that reduce farming and HG ancestry in post-Neolithic Europe. This is not mentioned, and I am not sure if it has been considered.

We have replaced the terminology "post-Neolithic" with "Neolithic" in the revised text. Also, as noted above, we restricted downstream analyses to individuals with less than 5% estimated Steppe ancestry.

Line 133: Defining an HG culturally only by hunting would mean it never reaches zero, as hunting is never abandoned. There are sites called hunting grounds that individuals with Anatolian ancestry may use for hunting and not for farming. This occurs in most mountainous areas even in much later periods than the Neolithic.

We do not necessarily preclude culturally defined farmers from hunting in our model. We acknowledge regional specificity regarding farming individuals who may also hunt (as the reviewer mentions, hunting is never fully abandoned). However, for the simplicity of the model, we classify individuals who predominantly farm or descend from those who farm as farmers.

We have now noted this assumption on lines 158-163 in our current submission (changes noted in red):

“To evaluate which of our numerical solutions are consistent with this pattern, we assessed the average proportion of EF ancestry after the transition to agricultural practices is complete—i.e., when the population of hunter-gathering individuals is nil. We acknowledge the existence of farming individuals in certain regions and time periods who may also practice elements of hunter-gatherer subsistence. However, for the simplicity of the model, we consider individuals who predominantly farm as farmers.”

Figure 1B: I might not understand this correctly, but why does Anatolian ancestry reach 1? In Europe, it usually does not reach 1 unless samples of such low coverage are included that a component is lost.

Figure 1B shows the average Anatolian ancestry after the farming expansion is complete, based on a purely theoretical 1-dimensional mathematical model. In this simplified model, very low rates of cultural transmission (left side of the x-axis) can result in complete ancestry turnover, with Anatolian ancestry reaching 100% as farmers fully replace HGs who do not transition to farming. However, these scenarios are unrealistic and do not fit empirical data. As the reviewer correctly notes, only parameter combinations that retain some degree of HG ancestry after the farming expansion are consistent with observed data and are therefore considered realistic.

Line 188: "peer to peer transmission": Does this estimate align with what we know today? Only in Iron Gates do we find HGs in farming contexts and vice versa. However, HG individuals move to the extremes of Europe, and HG contexts, once farming arrives, can still be sites with a specific hunting functionality. I believe all this should be incorporated into an elaborate discussion explaining that the archaeological context is necessary for the discussion.

We consider both peer-to-peer transmission (e.g., learning or imitating farming practices by a HG individual or group) and vertical transmission (e.g., the assimilation of HG individuals into farming communities, where offspring are raised as farmers). Both mechanisms can contribute to the incorporation of HG ancestry into culturally farming populations.

To address the reviewer's comment, we have clarified the limitations of our approach discriminating between those two scenarios, and highlighted the importance of archaeological data in the following ways:

First, we explicitly state that our results constrain the extent of peer-to-peer transmission that might have occurred. On lines 490-496, however, we note that these results are also consistent with very low levels of assimilation of HG individuals into farming populations. While our model provides a quantitative upper bound on these interactions, only archaeological data can resolve the specific mechanisms behind this transition. We leave such analyses to experts in archaeology.

Lines 438-444:

"Our learning rate estimate assumes complete within-group mating such that only horizontal, but no vertical cultural transmission, takes place. Similarly, our within-group mating estimate assumes no learning and thus no horizontal transmission. Due to the equivalence of horizontal and vertical transmission in their effect on the ancestry cline (Fig. 4), this implies that our estimated learning rate can be considered an upper limit to

the true learning rate in a combined model of both horizontal and vertical transmission, and vice versa for the estimated within-group mating rate (see Supplementary Fig. 11).”

Second, we now use the more neutral term “between-group mating” instead of “non-assortative mating” to avoid implying any assumptions about the social behaviors or preferences underlying observed mating patterns. We accordingly changed the title of our manuscript to *"Modeling the European Neolithic Expansion: Predominant Within-Group Mating and Limited Cultural Transmission"*.

Third, we emphasize that our approach establishes a baseline relationship between cultural interactions and spatiotemporal genetic ancestry patterns on a broad scale. However, applying our model to specific regions requires incorporating detailed local archaeological data and adapting the methodology to account for factors such as variable interbreeding and expansion rates at regional scales.

Finally, we now highlight that the classification of individuals as "farmers" is not always straightforward in empirical data. For instance, depending on geographic location, some farming societies continued to engage in hunting and maintained hunting grounds, reflecting regional variability in subsistence practices.

These clarifications have been incorporated into the revised manuscript (lines 158-163):

“To evaluate which of our numerical solutions are consistent with this pattern, we assessed the average proportion of EF ancestry after the transition to agricultural practices is complete—i.e., when the population of hunter-gathering individuals is nil. We acknowledge the existence of farming individuals in certain regions and time periods who may also practice elements of hunter-gatherer subsistence. However, for the simplicity of the model, we consider individuals who predominantly farm as farmers.”

And lines 535-549:

“Notably, aDNA studies from Atlantic France and Scandinavia show no evidence of farmer-associated ancestry in hunter-gatherers who overlapped in time with neighboring Neolithic farmers^{75,76}, corroborating our finding of low between-group mating. Additionally, in some regions, evidence suggests that foragers and farmers coexisted for centuries with minimal gene flow. In northeastern France and Germany, some late Neolithic farmers display unexpectedly high levels of hunter-gatherer ancestry (e.g., Mont-Aimé: 50–63%, Tangermünde: ~63%, Blätterhöhle: up to 85%), reflecting prolonged coexistence and limited genetic integration^{61,77–79}. Similarly, in the Iron Gates region, hunter-gatherers and early farmers lived side by side for centuries, exhibiting only minimal admixture despite extended contact⁸⁰.

These patterns suggest persistent but low levels of asymmetric admixture^{61,81} and reveal long-term demographic dynamics, such as the survival of regional hunter-gatherer groups

and potential cultural reversions by farmers to hunter-gatherer subsistence strategies. While our model focuses on the initial farming expansion, integrating these long-term dynamics into our model could offer valuable insights into the diversity and variability of local interactions and population structures following the initial farming expansion.”

Figure 3A: The gradient should point from right to left, like eastern to western farming expansion?

We appreciate the reviewer’s suggestion regarding the direction of the gradient in Figure 3A. Following this feedback, we have adjusted the gradient to run from right to left to better reflect the eastern-to-western farming expansion.

Line 221: Please briefly explain how the models of references 20 and 21 calculate the "front speed." Do they take into account C14 dating?

Ammerman & Cavalli-Sforza used C14 radiocarbon dating with standard errors of 200 years or less from Neolithic sites that showed archaeological evidence of early farming. They considered the earliest date of farming evidence at each site and plotted these dates against the distance from Jericho, which they identified as a potential origin for farming. This allowed them to calculate the average "front speed" as the distance farming expanded per year. Their primary analysis was presented in the 1971 paper (Ammerman & Cavalli-Sforza, 1971) and further discussed in their 1984 book (Ammerman & Cavalli-Sforza, 1984). Pinhasi, Fort, and Ammerman (2005) applied a similar method to estimate the speed of farming expansion (Pinhasi et al., 2005). However, they incorporated more recently dated Neolithic sites and considered multiple potential origin points for farming, extending the original analysis by Ammerman and Cavalli-Sforza.

We now added a short clarification to the text on lines 252-255:

These studies estimate the front speed of farming expansion by using C14 radiocarbon dates from Neolithic sites, plotting the earliest evidence of farming against distances from a proposed origin to calculate the average rate of spread^{22,29,30}.

Line 242: "post-neolithic" when it would be "neolithic expansion" or "neolithization" if you prefer.

To avoid ambiguity, we now state on lines 274-275:

“Previous literature observed a cline of Anatolian ancestry with distance from the farming origin in European Neolithic populations.”

Line 285: "This means some cultural transmission must have occurred, as a fully demic model would result in complete replacement."

I disagree with this phrase. Does "fully demic" not assume admixture can occur? Can it not be fully demic with local admixture? I do not think one thing excludes the other. If it is an assumption of the model, please explain it that way.

We appreciate the reviewer's thoughtful comment regarding the phrasing of "fully demic" models. To clarify, while demic and cultural transmission are not mutually exclusive, a fully demic model assumes complete population replacement with no interaction between incoming and local populations, resulting in total genetic turnover. Our definitions of demic and cultural models are based on the seminal work of Ammerman and Cavalli-Sforza, particularly as outlined in their 1984 book "The Neolithic Transition and the Genetics of Populations in Europe". On pages 82–83, they describe purely cultural, purely demic, and combined demic-cultural models. They emphasize that in a purely demic model, complete genetic replacement occurs because there is no interaction—neither cultural nor genetic—between groups (Ammerman & Cavalli-Sforza, 1984).

Thus, our phrasing, "This means that some level of either vertical or horizontal cultural transmission must have occurred, as a fully demic mode... would result in complete replacement." aligns with the expectations of a purely demic model as defined by Ammerman and Cavalli-Sforza, where the lack of interaction leads to total population replacement.

To address the reviewer's concern, we have now clarified in the main text that in our definition of a "fully demic model," neither vertical nor horizontal cultural transmission occurs, consistent with the definitions provided by Ammerman and Cavalli-Sforza (1984).

Lines 316-318 now read:

"This means some level of either vertical or horizontal cultural transmission must have occurred, as a fully demic model, as defined by Ammerman and Cavalli-Sforza (1984)²⁹, would result in complete ancestry replacement."

Line 396: "While archaeological and genetic research has described the impacts of the Neolithic transition from their respective disciplines, our interdisciplinary approach enables us to understand cultural and behavioral mechanisms invisible to the material evidence. Our mathematical and simulation-based modeling of the Neolithic expansion, for the first time, simultaneously predicts archaeological and genetic patterns linked to one of the most significant transformations in human history."

Do you believe that archaeology does not allow understanding "cultural and behavioral mechanisms"? We would not have studied this phenomenon from a genetic point of view if the archaeological context had not shown such a drastic change.

We thank the reviewer for pointing this out. We agree that archaeology has clearly demonstrated the drastic changes associated with the Neolithic expansion and has highlighted the importance of cultural and behavioral mechanisms. Indeed, the

archaeological record is invaluable for revealing the scale and scope of these transformations. However, the precise reasons behind such cultural changes often remain ambiguous when relying solely on archaeological evidence. For example, disentangling the relative contributions of migration versus the diffusion of ideas is challenging based on material evidence alone.

Similarly, while genetic data can provide compelling evidence for mass migration as a driver of cultural change, it is inherently limited in its ability to quantify the balance between cultural transmission and demic diffusion. This is precisely why we emphasize “cultural and behavioral mechanisms invisible to the material evidence”—these are aspects that become accessible only through an interdisciplinary approach that integrates modeling with both archaeological and genetic data.

To better appreciate the foundational contributions of archaeology and clarify our perspective, we have revised the paragraph as follows 496-506:

“While archaeological research has been instrumental in demonstrating the profound changes associated with the Neolithic transition, including the scale and timing of this transformation, it left unresolved questions about the specific mechanisms driving these changes. For example, archaeological evidence alone has not decisively revealed the relative contributions of migration versus the diffusion of ideas to cultural shifts²¹. Genetic data complements this by highlighting processes such as mass migration^{52,71} but similarly cannot fully quantify the interplay between cultural and demographic factors. Our interdisciplinary approach bridges this gap by using mathematical and simulation-based modeling to integrate insights from both archaeology and genetics. This enables us to explore cultural and behavioral mechanisms that are not directly observable in the material record, offering a more nuanced understanding of the Neolithic expansion as a transformative process in human history.”

Lines 432-434: "Notably, this is corroborated by aDNA studies which do not find any farmer-associated ancestry in hunter-gatherers that overlapped in time with neighboring Neolithic farmers 56,57."

Could the examples I mention at the beginning be such? Tangermünde and Blätterhöhle. I also suggest a review of the genetic studies conducted in Iron Gates, where HG (Koros individual and others) were also found in farmer/nearby farmer contexts. They are exceptions, but I think they need to be mentioned in the discussion.

We thank the reviewer for pointing this out and for suggesting examples such as Tangermünde, Blätterhöhle, and the Iron Gates individuals. We agree that these cases represent interesting exceptions where genetic evidence indicates some level of interaction between hunter-gatherers and farmers in overlapping contexts. To address this, we have revised the discussion, lines 535-549:

“Notably, aDNA studies from Atlantic France and Scandinavia show no evidence of farmer-associated ancestry in hunter-gatherers who overlapped in time with neighboring Neolithic farmers^{75,76}, corroborating our finding of low between-group mating. Additionally, in some regions, evidence suggests that foragers and farmers coexisted for centuries with minimal gene flow. In northeastern France and Germany, some late Neolithic farmers display unexpectedly high levels of hunter-gatherer ancestry (e.g., Mont-Aimé: 50–63%, Tangermünde: ~63%, Blätterhöhle: up to 85%), reflecting prolonged coexistence and limited genetic integration^{61,77–79}. Similarly, in the Iron Gates region, hunter-gatherers and early farmers lived side by side for centuries, exhibiting only minimal admixture despite extended contact⁸⁰.

These patterns suggest persistent but low levels of asymmetric admixture^{61,81} and reveal long-term demographic dynamics, such as the survival of regional hunter-gatherer groups and potential cultural reversions by farmers to hunter-gatherer subsistence strategies. While our model focuses on the initial farming expansion, integrating these long-term dynamics into our model could offer valuable insights into the diversity and variability of local interactions and population structures following the initial farming expansion.”

Thank you for bringing this to our attention, as it adds depth and nuance to the discussion.

Lines 469-470: "Again, this was interpreted as a cultural expansion that was initially mediated by cultural diffusion 52."

This needs clarification, as the phrase would be true only if the Bell Beaker originated in Iberia (which is only one of the theories). If it originated in Central Europe, there is mostly genetic change associated with cultural dispersion. I do not think reference 52 leans towards either option, it just presents both as possible.

Olalde et al. (2018) state in their abstract that migration was not a key mechanism for the spread of the Beaker complex between Iberia and Central Europe: “*We detected limited genetic affinity between Beaker-complex-associated individuals from Iberia and central Europe, and thus exclude migration as an important mechanism of spread between these two regions*” (Olalde et al., 2018). This conclusion was based on genetic evidence showing that most Beaker-associated individuals in Iberia lacked steppe-related ancestry and were genetically similar to preceding Iberian populations, whereas steppe ancestry was widespread in Beaker-associated individuals in Central Europe. In contrast, the further dissemination of the Beaker complex into Britain was associated with a large-scale migration event, due to the near-total replacement of Britain’s gene pool.

However, we acknowledge the reviewer’s observation that Olalde et al. do not propose a clear direction of spread (e.g., Iberia to Central Europe or vice versa), as the geographic origins of the Beaker complex remain debated.

To align the sentence more closely with Olalde et al.’s findings, we revised lines 635-639 to:

“Similar to the farming expansion in southwest Asia, the Bell Beaker cultural phenomenon initially spread across parts of Europe with little impact on genetic ancestry—individuals from Iberia and Central Europe show limited genetic affinity to one another. However, the later expansion of the Beaker complex to Britain led to a massive replacement of nearly all of Britain’s gene pool.”

I hope my comments can help improve the manuscript and I urge the authors to follow the comments of the reviewers who are experts in modelling as this is outside my expertise.

Thank you again for your thoughtful comments and valuable feedback on our manuscript. We truly appreciate the time and effort you dedicated to reviewing our work. Your insights have been incredibly helpful in refining and improving the manuscript, and we are sincerely grateful for your contribution.

References Cited in Responses to Reviewers:

- Alroy, J. (2001). A Multispecies Overkill Simulation of the End-Pleistocene Megafaunal Mass Extinction. *Science*, 292(5523), 1893–1896. <https://doi.org/10.1126/science.1059342>
- Ammerman, A. J., & Cavalli-Sforza, L. L. (1971). Measuring the Rate of Spread of Early Farming in Europe. *Man*, 6(4), 674–688. <https://doi.org/10.2307/2799190>
- Ammerman, A. J., & Cavalli-Sforza, L. L. (1984). *The Neolithic Transition and the Genetics of Populations in Europe*. Princeton University Press. <http://www.jstor.org/stable/j.ctt7zvqz7>
- Curat, M., & Excoffier, L. (2005). The effect of the Neolithic expansion on European molecular diversity. *Proceedings of the Royal Society B: Biological Sciences*, 272(1564), 679–688. <https://doi.org/10.1098/rspb.2004.2999>
- Davison, K., Dolukhanov, P., Sarson, G. R., & Shukurov, A. (2006). The role of waterways in the spread of the Neolithic. *Journal of Archaeological Science*, 33(5), 641–652. <https://doi.org/10.1016/j.jas.2005.09.017>
- Eshed, V., Gopher, A., Gage, T. B., & Hershkovitz, I. (2004). Has the transition to agriculture reshaped the demographic structure of prehistoric populations? New evidence from the Levant. *American Journal of Physical Anthropology*, 124(4), 315–329.

<https://doi.org/10.1002/ajpa.10332>

- Fernandes, R., Grootes, P., Nadeau, M.-J., & Nehlich, O. (2015). Quantitative diet reconstruction of a Neolithic population using a Bayesian mixing model (FRUITS): The case study of Ostorf (Germany). *American Journal of Physical Anthropology*, *158*(2), 325–340. <https://doi.org/10.1002/ajpa.22788>
- Fort, J. (2012). Synthesis between demic and cultural diffusion in the Neolithic transition in Europe. *Proceedings of the National Academy of Sciences*, *109*(46), 18669–18673. <https://doi.org/10.1073/pnas.1200662109>
- Fort, J. (2021). Front propagation and cultural transmission. Theory and application to Neolithic transitions. *Chaos, Solitons & Fractals*, *148*, 111060. <https://doi.org/10.1016/j.chaos.2021.111060>
- Fort, J. (2022). Prehistoric spread rates and genetic clines. *Human Population Genetics and Genomics*, *2*(2). <https://doi.org/10.47248/hpgg2202020003>
- Fort, J., & Pérez-Losada, J. (2024). Interbreeding between farmers and hunter-gatherers along the inland and Mediterranean routes of Neolithic spread in Europe. *Nature Communications*, *15*(1), 7032. <https://doi.org/10.1038/s41467-024-51335-4>
- Fournier, R., Tsangalidou, Z., Reich, D., & Palamara, P. F. (2023). Haplotype-based inference of recent effective population size in modern and ancient DNA samples. *Nature Communications*, *14*(1), 7945. <https://doi.org/10.1038/s41467-023-43522-6>
- Haak, W., Forster, P., Bramanti, B., Matsumura, S., Brandt, G., Tänzer, M., Villems, R., Renfrew, C., Gronenborn, D., Alt, K. W., & Burger, J. (2005). Ancient DNA from the First European Farmers in 7500-Year-Old Neolithic Sites. *Science*, *310*(5750), 1016–1018. <https://doi.org/10.1126/science.1118725>
- Harney, É., Patterson, N., Reich, D., & Wakeley, J. (2021). Assessing the performance of qpAdm: A statistical tool for studying population admixture. *Genetics*, *217*(4), iyaa045. <https://doi.org/10.1093/genetics/iyaa045>

- Isern, N., Fort, J., & de Rioja, V. L. (2017). The ancient cline of haplogroup K implies that the Neolithic transition in Europe was mainly demic. *Scientific Reports*, 7(1), 11229. <https://doi.org/10.1038/s41598-017-11629-8>
- Lazaridis, I., Patterson, N., Anthony, D., Vyazov, L., Fournier, R., Ringbauer, H., Olalde, I., Khokhlov, A. A., Kitov, E. P., Shishlina, N. I., Ailincăi, S. C., Agapov, D. S., Agapov, S. A., Batieva, E., Bauyrzhan, B., Bereczki, Z., Buzhilova, A., Changmai, P., Chizhevsky, A. A., ... Reich, D. (2024). *The Genetic Origin of the Indo-Europeans* (p. 2024.04.17.589597). bioRxiv. <https://doi.org/10.1101/2024.04.17.589597>
- Lipson, M., Szécsényi-Nagy, A., Mallick, S., Pósa, A., Stégmár, B., Keerl, V., Rohland, N., Stewardson, K., Ferry, M., Michel, M., Oppenheimer, J., Broomandkhoshbacht, N., Harney, E., Nordenfelt, S., Llamas, B., Gusztáv Mende, B., Köhler, K., Oross, K., Bondár, M., ... Reich, D. (2017). Parallel palaeogenomic transects reveal complex genetic history of early European farmers. *Nature*, 551(7680), 368–372. <https://doi.org/10.1038/nature24476>
- Marchi, N., Winkelbach, L., Schulz, I., Brami, M., Hofmanová, Z., Blöcher, J., Reyna-Blanco, C. S., Diekmann, Y., Thiéry, A., Kapopoulou, A., Link, V., Piuz, V., Kreutzer, S., Figarska, S. M., Ganiatsou, E., Pukaj, A., Struck, T. J., Gutenkunst, R. N., Karul, N., ... Excoffier, L. (2022). The genomic origins of the world's first farmers. *Cell*, 185(11), 1842-1859.e18. <https://doi.org/10.1016/j.cell.2022.04.008>
- Olalde, I., Brace, S., Allentoft, M. E., Armit, I., Kristiansen, K., Booth, T., Rohland, N., Mallick, S., Szécsényi-Nagy, A., Mittnik, A., Altena, E., Lipson, M., Lazaridis, I., Harper, T. K., Patterson, N., Broomandkhoshbacht, N., Diekmann, Y., Faltyskova, Z., Fernandes, D., ... Reich, D. (2018). The Beaker phenomenon and the genomic transformation of northwest Europe. *Nature*, 555(7695), 190–196. <https://doi.org/10.1038/nature25738>
- Olalde, I., Mallick, S., Patterson, N., Rohland, N., Villalba-Mouco, V., Silva, M., Dulias, K., Edwards, C. J., Gandini, F., Pala, M., Soares, P., Ferrando-Bernal, M., Adamski, N.,

- Broomandkhoshbacht, N., Cheronet, O., Culleton, B. J., Fernandes, D., Lawson, A. M., Mah, M., ... Reich, D. (2019). The genomic history of the Iberian Peninsula over the past 8000 years. *Science*, 363(6432), 1230–1234. <https://doi.org/10.1126/science.aav4040>
- Patterson, N., Isakov, M., Booth, T., Büster, L., Fischer, C.-E., Olalde, I., Ringbauer, H., Akbari, A., Cheronet, O., Bleasdale, M., Adamski, N., Altena, E., Bernardos, R., Brace, S., Broomandkhoshbacht, N., Callan, K., Candilio, F., Culleton, B., Curtis, E., ... Reich, D. (2022). Large-scale migration into Britain during the Middle to Late Bronze Age. *Nature*, 601(7894), 588–594. <https://doi.org/10.1038/s41586-021-04287-4>
- Pinhasi, R., Fort, J., & Ammerman, A. J. (2005). Tracing the Origin and Spread of Agriculture in Europe. *PLOS Biology*, 3(12), e410-.
- Posth, C., Yu, H., Ghalichi, A., Rougier, H., Crevecoeur, I., Huang, Y., Ringbauer, H., Rohrlach, A. B., Nägele, K., Villalba-Mouco, V., Radzeviciute, R., Ferraz, T., Stoessel, A., Tukhbatova, R., Drucker, D. G., Lari, M., Modi, A., Vai, S., Saupe, T., ... Krause, J. (2023). Palaeogenomics of Upper Palaeolithic to Neolithic European hunter-gatherers. *Nature*, 615(7950), 117–126. <https://doi.org/10.1038/s41586-023-05726-0>
- Steele, J., Adams, J., & Sluckin, T. (1998). Modelling Paleoindian dispersals. *World Archaeology*, 30(2), 286–305. <https://doi.org/10.1080/00438243.1998.9980411>
- Villalba-Mouco, V., van de Loosdrecht, M. S., Posth, C., Mora, R., Martínez-Moreno, J., Rojo-Guerra, M., Salazar-García, D. C., Royo-Guillén, J. I., Kunst, M., Rougier, H., Crevecoeur, I., Arcusa-Magallón, H., Tejedor-Rodríguez, C., García-Martínez de Lagrán, I., Garrido-Pena, R., Alt, K. W., Jeong, C., Schiffels, S., Utrilla, P., ... Haak, W. (2019). Survival of Late Pleistocene Hunter-Gatherer Ancestry in the Iberian Peninsula. *Current Biology*, 29(7), 1169-1177.e7. <https://doi.org/10.1016/j.cub.2019.02.006>

LaPolice T.M., Williams M.P. & Huber C.D.
Point by Point Response to Reviewers (Revision #2)

We thank the reviewers and the editor for their time and their continued feedback. Below, we have responded to the comments about our first revision, and made changes in this second revision based on the suggestions.

Reviewer #1

(Remarks to the Authors, Responses from the Authors, and Relevant Excerpts from the Revised Manuscript):

I have reviewed the authors' revisions and the responses to the reviewer comments. I have no further comments or suggestions. The authors have adequately addressed the comments from the reviewers.

We thank the reviewer very much for their time and comments. We are pleased we were able to respond to their comments satisfactorily. Their suggestions have contributed to a significantly improved manuscript, and we appreciate their detailed insight.

Reviewer #2

(Remarks to the Authors, Responses from the Authors, and Relevant Excerpts from the Revised Manuscript):

First, I would like to thank the authors for providing a well-structured and detailed response to my main concerns. Their thorough reply includes meaningful methodological refinements while maintaining the study's original scope. They acknowledge the importance of regional variability in the Neolithic expansion and address this by modifying their model to distinguish between the Mediterranean and continental routes. The incorporation of multiple-route models, adjustments for movement speed, and geographic barriers demonstrate a genuine effort to refine their analysis. Additionally, the authors clarify that while regional variation is significant, their primary objective is to establish a broad baseline for cultural interactions and ancestry estimates. I also appreciate their discussion of the study's limitations and suggestions for future research, including integrating autosomal and mitochondrial data to investigate potential sex-specific migration and interaction patterns.

We appreciate the kind words from the reviewer and are happy we were able to address the valuable points they raised satisfactorily. We thank the reviewer for their detailed feedback which has allowed us to improve the manuscript greatly.

As a non-specialist in statistical modeling, I appreciate the effort to make the manuscript more accessible to researchers from different disciplines. The mathematical framework is now much clearer, and the added references in the figure caption, results, and methods sections help guide readers to the explanations more effectively. However, I suggest ensuring that the explanations in Supplementary Note 1 are as concise and intuitive as possible, particularly for readers without a strong background in mathematical modeling. A brief introductory sentence explaining the general role of reaction-diffusion models in studying population dynamics could further enhance clarity for a broader audience.

In an attempt to improve the accessibility of the writing to readers from multiple disciplines we have included an introductory paragraph to the main text about the general role of reaction-diffusion models in studying population dynamics.

Lines 135-148 now read (changes in red):

“We started by examining the influence of cultural transmission on the dispersal of EF ancestry throughout a Europe inhabited by hunter-gatherers (HGs). We employed a reaction-diffusion model in a one-dimensional (1D) continuum, as a simplified representation of this process. Reaction-diffusion models are commonly used in ecology to describe how populations expand and interact over space and time. These models combine two processes: diffusion, representing random movement or dispersal of individuals, and reaction, which captures local dynamics such as population growth or between-species interaction. In the context of the Neolithic expansion, reaction-diffusion models provide a framework for simulating how farming populations spread geographically and interact with local HG populations. Our model consists of a system of partial differential equations that mirror those utilized in prior models addressing the spread of agriculture (e.g., Fort et al., 2012)³¹ and of the farmer-specific mitochondrial haplogroup K⁵⁰. Importantly, our model extends beyond these by incorporating the genetic ancestry of individuals at a single genetic locus, enabling us to explore how cultural transmission and demographic processes jointly shaped the ancestry landscape of prehistoric Europe.”

We have also rewritten Supplementary Note 1 to make it more intuitive and concise. The new note reads as follows:

“Supplementary Note 1: Comprehensive description of the symbols and components of the equations in Main Text Fig. 1

Our reaction-diffusion model describes the spread of farming by tracking three population groups across space: (1) farmers with Early Farming (EF) ancestry (u), (2) farmers with Western Hunter Gatherer (WHG) ancestry (v), and (3) hunter-gatherers with WHG ancestry (w), each measured as population density (individuals per km). The model is defined by a system of partial differential equations, each with three core components: spatial movement (diffusion), population growth, and cultural transmission (learning).

- Diffusion terms capture the movement of individuals across space, with group-specific diffusion constants (D_u , D_v , D_w) determining how quickly each population spreads across the landscape.
- Logistic growth terms describe how populations locally increase in size over time but are limited by local carrying capacities (K_F for farmers, K_{HG} for hunter-gatherers). Growth slows as population densities approach these environmental limits. The parameters a_u , a_v , and a_w represent the intrinsic growth rates of each population type.
- Cultural transmission terms model how hunter-gatherers adopt farming practices from neighboring farmers. The learning rate (f) reflects how frequently this adoption occurs. Adoption also depends on the local density of farmers relative to the total population, meaning transmission is more likely when farmers are common in the area. A preference parameter (γ) controls whether hunter-gatherers are more likely to learn from farmers than from their peers. Specifically, The expression $((u+v)w)/(u+v+\gamma w)$ models the likelihood of hunter-gatherers with WHG ancestry (w) adopting farming practices, depending on the density of farmers ($u+v$) and the preference factor γ .

Together, these three components allow the model to predict how the farming lifestyle—and the associated genetic ancestry—spreads across Europe through a mixture of migration, growth, and learning.”

The inclusion of a supplementary data file detailing all analyzed individuals, along with metadata, filtering procedures, and qpAdm results, significantly improves transparency in data selection, ensuring clarity and reproducibility. I also appreciate the mention of the AADR database version in both the methods and results sections. However, it is worth noting that reliance on this database introduces a bias, as not all available samples are included.

We appreciate the reviewer’s comment regarding the potential for bias introduced by relying on a single database. While we recognize this concern, we believe that our use of the Allen Ancient DNA Resource (AADR) does not introduce any systematic bias that would affect the conclusions from our farming expansion model. The AADR has a comprehensive and well maintained collection of samples that is highly concordant with another similar, large, community-maintained, aDNA database (see Figure 4 from Schmid et al., *elife*, 2024, below). Nonetheless, we have added a note to the Methods section to acknowledge the reliance on AADR and the implications regarding data coverage.

Lines 922-929 read:

“Finally, we acknowledge that reliance on a single aDNA database (in this case AADR v62.0)⁵⁸ has the potential to introduce bias. However, the AADR⁵⁸ remains one of the most comprehensive, well-curated, and widely used sources of publicly available ancient DNA data. Moreover, recent work by Schmid et al. (2024)¹⁰³ comparing the AADR and the Poseidon database—another large, community-maintained ancient DNA resource—demonstrates substantial overlap between the two, particularly for European samples. This concordance across independently maintained datasets suggests that our sample set reflects the broader landscape of available ancient genomes relevant to our study.”

Figure 4 from Schmid et al., 2024

[See Figure 4 in Schmid Clemens, Ghalichi Ayshin, Lamnidis Thisseas C., Mudiyansele Dhananjaya B. A., Haak Wolfgang, Schiffels Stephan(2024) Poseidon – A framework for archaeogenetic human genotype data management eLife 13:RP98317 <https://doi.org/10.7554/eLife.98317.1>]

PCA = Poseidon Community Archive and PAA = AADR v54.1.p1

While I acknowledge the effort to incorporate regional variability and differentiate between Neolithic expansion routes, I encourage a deeper discussion on how these findings align—or contrast—with existing archaeological evidence from specific regions. Some of the observed variations in learning rates could be explored in greater detail through references to archaeological material culture, settlement patterns, or subsistence strategies. Would the authors consider integrating more direct comparisons with archaeological datasets to further support their interpretations?

We thank the reviewer for this thoughtful suggestion. We agree that integrating more direct comparisons with regional archaeological data would further strengthen the interpretation of our findings. While a full synthesis of archaeological material culture and settlement patterns across Europe is beyond the scope of the present study, in our previous response we expanded our Discussion to include examples where archaeological evidence aligns with our findings—such as prolonged coexistence and limited admixture in regions like the Iron Gates and parts of northern Europe. These cases support the inference of low cultural transmission rates and minimal between-group mating in certain areas. In this current response, we now also highlight the potential for future work to more systematically connect regional variation in learning rates to archaeological indicators of interaction, such as shared tool traditions or shifts in subsistence strategy. Further, we have expanded the

future directions section of our manuscript to incorporate a discussion of the type of archeological and genetic data needed to estimate learning rates within specific local regions.

Lines 588-603 in the Discussion now read (changes in red):

“Future studies should aim to integrate autosomal and mitochondrial data into a unified framework to better understand the influence of sex-specific factors on cultural transmission and migration dynamics. They could also delve into the potential for regional variation of cultural transmission rates and how it could be linked to observed local patterns derived from archaeological studies. For example, analyzing material evidence at archaeological sites along the wavefront could connect our estimated learning rates with indicators of interaction, such as shared tool traditions or shifts in subsistence strategy. However, to assess how learning rates change regionally, two key components are needed. First, local estimates of front speed are necessary, which requires archaeological data documenting the regional pace of farming expansion. Second, more extensive sampling of Early Neolithic ancient DNA in specific regions is essential. This would enable the construction of regional ancestry clines that could be used to estimate localized learning rates from simulations under our model. While ancient genomic data are available for many parts of Neolithic Europe, current sample sizes may be insufficient—once filtered by region and time—to reliably establish local ancestry clines. Expanding aDNA datasets in key geographic regions and generating localized front speed estimates will ultimately allow us to bridge the gap between our large-scale model and finer-scale regional inferences.”

The additional analyses are valuable, but I still have some concerns about the inclusion of Middle and Late Neolithic samples. While learning rate estimates appear stable, the significant drop in the intercept during the Late Neolithic ($p < 0.001$) suggests that temporal variation does have an effect. Could the authors elaborate further on why they believe this does not impact their conclusions? Additionally, discussing the implications of using a broader time range in the main text would improve transparency.

We appreciate the reviewer’s continued attention to this important point. The increase in WHG ancestry in the Middle and Late Neolithic—often referred to as the “hunter-gatherer resurgence”—has been interpreted as resulting from the incorporation of small, previously isolated HG groups that persisted in certain regions following the initial farming expansion. Since our modeling focuses on this initial expansion phase and the degree of interaction (via vertical or horizontal cultural transmission) required to explain observed ancestry clines, we consider the Early Neolithic samples more representative for estimating the learning rate during this key period. Notably, the lowest estimated learning rate (0.0007) is derived from these Early Neolithic samples.

However, we agree that including Middle and Late Neolithic samples increases the temporal breadth of the dataset and may reflect interactions occurring well after the initial

expansion. Although the estimated learning rates for the Middle and Late Neolithic (0.0011 and 0.0013, respectively) are slightly higher, they remain two orders of magnitude lower than previous estimates that were based solely on expansion speed (e.g., ~0.1), and thus still support our conclusion that cultural transmission was limited.

In response to the reviewer's suggestion, we have added a discussion to the main text clarifying that while including later Neolithic samples increases sample size and improves statistical power, it also introduces post-expansion dynamics that may inflate learning rate estimates. We now explicitly note that estimates based on the full Neolithic dataset may reflect a combination of early and late interactions, and are therefore likely to slightly overestimate the amount of cultural transmission that occurred during the initial expansion phase.

Lines 446-449 in the Results now read (changes in red):

"However, there is a significant drop in the intercept from Early to Late ($p < 0.001$) (Supplementary Fig. 10). Despite this, we do not see a considerable change in our estimated learning rate across time periods, with all returning estimates of approximately 0.001."

Lines 604-621 in the Discussion now read (changes in red):

"We also acknowledge that, in addition to geographic variability, temporal changes in ancestry—such as the resurgence of WHG ancestry during the Middle and Late Neolithic⁵⁹—may influence estimates of the learning rate. Our primary modeling goal is to capture dynamics during the initial farming expansion; thus, our estimation procedure assumes that the analyzed individuals are representative of the period immediately following this expansion. To assess the robustness of this assumption, we divided the ancient DNA dataset into Early, Middle, and Late Neolithic subsets (Supplementary Fig. 10) and re-estimated the learning rate for each group independently using our complex landscape model.

As expected, the learning rate estimates increase slightly over time (0.0007, 0.0011, and 0.0013 for the Early, Middle, and Late Neolithic, respectively), consistent with the gradual re-integration of HG ancestry after the expansion phase. While a significant drop in the ancestry intercept is observed in the Late Neolithic ($p < 0.001$), we interpret this as reflecting post-expansion dynamics—such as admixture with remnant HG groups—that are not central to our model's focus. For this reason, we consider the Early Neolithic estimate (0.0007) to be more representative of the initial expansion process. Importantly, even the higher estimates from later periods remain two orders of magnitude lower than previous upper estimates based solely on front speed³¹, supporting our main conclusion that cultural transmission played a minimal role in the spread of farming during the Neolithic."

Finally, while the modifications add regional nuance, the model remains relatively high-level. I understand that more detailed regional modeling would require extensive local archaeological data. However, given the increasing resolution of ancient DNA data, a finer-scale approach may be necessary to fully capture the complexity of regional variability. Would the authors consider discussing how future studies might bridge this gap?

We thank the reviewer for their comments here and throughout the revision process, their valuable comments have allowed us to greatly improve our manuscript.

In addressing this final point from the reviewer, we expanded our discussion to more thoroughly address potential future approaches that could help bridge the gap and account for the regional variability that the author notes. While we appreciate the reviewer's acknowledgement that such study lies beyond the scope of this current project, we agree that it could significantly enhance understanding of the Neolithic Expansion and build on the insights provided by our model.

Lines 588-603 in the Discussion now read (changes in red):

“Future studies should aim to integrate autosomal and mitochondrial data into a unified framework to better understand the influence of sex-specific factors on cultural transmission and migration dynamics. They could also delve into the potential for regional variation of cultural transmission rates and how it could be linked to observed local patterns derived from archaeological studies. For example, analyzing material evidence at archaeological sites along the wavefront could connect our estimated learning rates with indicators of interaction, such as shared tool traditions or shifts in subsistence strategy. However, to assess how learning rates change regionally, two key components are needed. First, local estimates of front speed are necessary, which requires archaeological data documenting the regional pace of farming expansion. Second, more extensive sampling of Early Neolithic ancient DNA in specific regions is essential. This would enable the construction of regional ancestry clines that could be used to estimate localized learning rates from simulations under our model. While ancient genomic data are available for many parts of Neolithic Europe, current sample sizes may be insufficient—once filtered by region and time—to reliably establish local ancestry clines. Expanding aDNA datasets in key geographic regions and generating localized front speed estimates will ultimately allow us to bridge the gap between our large-scale model and finer-scale regional inferences.”

Reviewer #3

(Remarks to the Authors, Responses from the Authors, and Relevant Excerpts from the Revised Manuscript):

I thank the authors for carefully addressing all my comments and for their clear point-by-point response.

We thank the reviewer for their time and thoughtful feedback, and are pleased that we were able to address their comments satisfactorily in our revision. Their insights have contributed to a significantly improved manuscript.

I simply suggest that the authors explain in the methods section why Steppe was included as a source. Although I suggested using this model, it is true that this component is generally assumed to be unnecessary for explaining the ancestry of Neolithic individuals.

We thank the reviewer for their original suggestion to use this model for our analysis. It has improved the overall quality of our ancestry estimations.

As the reviewer described below, we included Steppe as a source population in our analytical framework as a methodological safeguard, based on findings from Chintalapati et al., 2022. Their comprehensive analysis of 109 samples spanning the Late Neolithic, Chalcolithic, and Bronze Age (3000 BCE–750 CE) demonstrated that most individuals carried Steppe pastoralist-related ancestry, with the earliest European appearance of this ancestry component dated to approximately 3200 BCE in Scandinavia. Another source for Steppe ancestry in Neolithic individuals could be modern contamination. To prevent this ancestry component from confounding our analyses of Neolithic population dynamics, we maintained Steppe as a source in our model while implementing a filtering approach that excluded individuals exhibiting greater than 5% Steppe-derived ancestry. This methodological decision ensured that undetected Steppe ancestry would not influence our primary findings regarding pre-Steppe European genetic structure.

In lines 885-890 we now justify our purpose for filtering out Steppe ancestry (changes in red):

“We used qpAdm⁵⁹ from the ADMIXTOOLS2⁹¹ package with parameters ‘allsnps=TRUE’ and ‘afprod = TRUE’ to obtain empirical estimations of EF, Steppe, and WHG ancestry contributions to Neolithic aDNA samples from Europe (Allen Ancient DNA Resource (AADR) v62.0)⁵⁸. Including Steppe as a source population allowed us to filter individuals from our dataset that contained Steppe ancestry or possible modern contamination. I.e., by filtering out these individuals, we were able to capture the population dynamics unique to the farming expansion.”

In many of your models, this [Steppe] component is negligible, and when this happens, it is good practice to show the nested model (i.e., the same qpAdm model without the source

that can be discarded, e.g., because its percentage is lower than its error margin). I do not consider it necessary to perform all possible nested models since the percentage changes would be minimal, but I believe it would be beneficial to explain why Steppe was retained in the model.

At the reviewer's suggestion of it being good practice to report the nested qpAdm model, we have now generated new columns in Supplemental Data File 2 which show the nested model values for each individual. We also confirm that the percentage changes are minimal by calculating the correlation in estimated EF ancestry proportion between the full and the nested model ($r^2 = 0.996$, $p < 0.001$).

We mention this on lines 329-337 which now read (changes in red):

"A complete metadata file for each 1531 analyzed individuals can be found in Supplementary Data File 1. EF, Steppe, and WHG ancestry components were estimated for each site using the software qpAdm⁵⁹ following Patterson et al. 2022⁶⁰ (see Methods). Ancestry estimates for all 1531 analyzed individuals can be found in Supplementary Data File 2. For transparency, in Supplementary Data File 2 we have also included columns showing the nested qpAdm⁵⁹ model for each individual, i.e., assuming only EF and WHG ancestry sources. The correlation between ancestry estimates from the full and the nested model is high ($r^2 = 0.996$, $p < 0.001$), suggesting that the addition of the Steppe component did not affect the EF ancestry estimation."

As described in our previous response above, we have now included a rationale for why the Steppe was included as a source. Thank you for this suggestion- it improves the transparency of our analysis methods.

Below is a plot showing the correlation between the nested model EF ancestry estimates and the full EF ancestry estimate.

In my view, the two main reasons [to include the Steppe as a source] would be:

1. As a validation method to confirm the absence of Steppe ancestry in the Late Neolithic individuals analyzed (serving the same purpose as a PCA).

This point raised by the reviewer was our primary motivation for including the Steppe as a source—it allowed us to filter out individuals with Steppe ancestry from our fitting dataset and ensured that we analyze pre-Steppe European genetic structure (see responses above).

2. To quantify potential modern contamination, as any post-Neolithic human contaminant source would introduce some degree of Steppe ancestry.

Although not our original reason for including Steppe ancestry, we agree that it also helps to identify and remove potential contamination. We have added this point as additional justification in our manuscript:

Lines 885-890 read:

“We used qpAdm⁵⁹ from the ADMIXTOOLS2⁹¹ package with parameters ‘allsnps=TRUE’ and ‘afprod = TRUE’ to obtain empirical estimations of EF, Steppe, and WHG ancestry contributions to Neolithic aDNA samples from Europe (Allen Ancient DNA Resource (AADR) v62.0)⁵⁸. Including Steppe as a source population allowed us to filter individuals

from our dataset that contained Steppe ancestry or possible contamination. I.e., by filtering out these individuals, we were able to capture the population dynamics unique to the farming expansion.”

We observe a general trend of an increasing number of individuals with > 5% Steppe ancestry towards the present, with the Late Neolithic period possessing over 80% of the filtered individuals. These may represent some of the oldest Steppe ancestry in Europe, with previous estimates dating the arrival of Steppe ancestry to Europe to ~3200 BCE (~5150 yBP) (Chintalapati et al., 2022). For the Early Neolithic samples (and perhaps the younger Neolithic periods as well), small levels of Steppe ancestry may be a result of low-level contamination or other estimation errors.

Excluding samples with more than 5% Steppe, as the authors have done, seems like very good practice.

We thank the reviewer for pointing out and verifying it is a valid and useful approach to exclude samples with more than 5% Steppe ancestry. We, again, thank them for their suggestions to improve our analysis of the ancient samples.

Another point I would like to mention, just in case the authors are not aware, is that it might be useful to discuss in the methods section the occurrence of negative EF or WHG components. By using Balkan_N as a source, you standardize the type of farming ancestry used, as it has been observed that this ancestry may vary slightly by region (Lazaridis, 2024). Since it is a generalist source, it tends to yield higher p-values (which is the goal in qpAdm). However, by using Balkan_N, we create a farming ancestry pool that already contains some HG ancestry (similar to what happens with Anatolia, though perhaps to a lesser extent). This is what could lead to negative values for some samples, either for HG or farming ancestry, suggesting that the population source contains more of that component than the target. To summarize, these negative values are informative and can be left as they are. It might be worth mentioning that the geographic origin of farming or regions east of the Balkans is where these negative components appear, as some individuals slightly deviate from the values of Balkan_N chosen as the EF source. Perhaps this could be briefly noted when performing linear regression and observing EF ancestry with increasing distance from the farming origin. In conclusion, your source is not the exact origin, but it is an adequate proxy that improves p-values in Europe. Furthermore, having Anatolia in the outgroups and the Balkans in the sources helps reduce errors. For all these reasons—and because this approach is directly applicable to the future study of the Bell Beaker phenomenon—I strongly believe it is the best strategy to follow.

We thank the reviewer for this insightful and constructive comment regarding the interpretation of negative ancestry estimates in our qpAdm analysis. We are pleased that the reviewer believes we are following the best strategy in our analysis.

To address the reviewer’s comment regarding negative ancestry values, we have now examined the spatiotemporal distribution of negative EF and WHG ancestry values across our dataset (see figures below). In line with the reviewer’s expectations, our analysis shows that negative WHG ancestry estimates are most frequently observed among individuals from the Balkans and the southeastern Mediterranean across all Neolithic time periods. This is consistent with the fact that our EF source population (Balkan_N) already contains some WHG ancestry. Conversely, some individuals from northern and northeastern Europe—particularly in Scandinavia and regions dating to 7,000 years BP and more recent—exhibit slightly negative EF ancestry values. We have now added this finding and expanded our justification for using Balkan_N as the EF source population, echoing the reviewer’s reasoning (Methods, lines 903-921):

“In our qpAdm⁵⁹ analyses, we used Balkan_N as the proxy source population for Early Farmer (EF) ancestry, reflecting previous recommendations⁶⁰. While not the earliest point of Neolithic expansion, Balkan_N serves as a geographically and genetically intermediate farming population that provides broad coverage across European Neolithic contexts. Notably, Balkan_N individuals were selected to possess minimal hunter-gatherer admixture⁶⁰, which improves qpAdm⁵⁹ model fit—typically yielding higher p-values—when modeling admixed Neolithic populations in Europe. To further stabilize ancestry estimates, we included Anatolia_N and a set of deep outgroups in the right populations (see full list in Supplementary Data File 2). As a consequence of using Balkan_N, some ancient samples—especially those from southeastern Europe or regions east of the Balkans—can yield slightly negative WHG ancestry estimates, while samples from northern and northeastern Europe (particularly those younger than 7,000 years BP) may show negative EF ancestry values. In sum, of the samples that passed the plausibility criteria of p-value ≥ 0.01 , Steppe ancestry [0, 0.05] and EF ancestry S.E. ≤ 0.022 , only ~3.8% and ~3.3% were filtered due to negative EF and WHG ancestry estimates, respectively. Under these conditions we followed standard qpAdm⁵⁹ practices and excluded individuals with negative ancestry proportions from downstream analyses. This filtering step ensures that only biologically interpretable ancestry estimates are included in our regressions and simulation-based inference approach and improves model robustness across diverse European Neolithic samples.”

Proportion of otherwise plausible individuals with negative WHG ancestry by date period

Proportion of otherwise plausible individuals with negative EF ancestry by date period

Line 863: The text states "85000," but the correct date is "8500."

This is a good catch by the reviewer, we thank you for noting this issue. We have fixed it and the line (now line 896) reads "8500" rather than "85000".

Line 871: It would be better to say that the PCA reveals broad genetic ancestry rather than genetic relatedness.

We appreciate this point by the reviewer as it better reflects what the PCA depicts. We have changed this line (now line 930-931) accordingly and it now reads:

"We visualized broad genetic ancestry patterns of the analyzed individuals through principal component analysis (PCA)¹⁰⁴"

These are minor comments, and I am satisfied with the thorough revision process carried out by the authors. I congratulate them on their work.

We thank the reviewer very much for their kind words regarding our manuscript and our revisions. We appreciate their detailed comments and believe their suggestions have allowed us to greatly improve our paper.

References Cited in Responses to Reviewers:

Chintalapati, M., Patterson, N., & Moorjani, P. (2022). The spatiotemporal patterns of major human admixture events during the European Holocene. *eLife*, *11*, e77625. <https://doi.org/10.7554/eLife.77625>

Schmid, C., Ghalichi, A., Lamnidis, T. C., Mudiyansele, D. B. A., Haak, W., & Schiffels, S. (2024). Poseidon – A framework for archaeogenetic human genotype data management. *eLife*, *13*. <https://doi.org/10.7554/eLife.98317.1>

LaPolice T.M., Williams M.P. & Huber C.D.
Point by Point Response to Reviewers (Revision #3)

We thank the reviewers and the editor for their time and their continued feedback. Below, we have responded to the comments about our second revision, and made changes in this third revision based on their suggestions.

Reviewer #2

(Remarks to the Authors, Responses from the Authors, and Relevant Excerpts from the Revised Manuscript):

The authors have thoroughly addressed all of my previous comments and suggestions. I am satisfied with the revisions and have no further comments.

We thank the reviewer very much for their time and comments across both previous revisions. We are pleased we were able to satisfactorily respond to their comments. The suggestions have resulted in a significantly improved manuscript and we appreciate their detailed insight.

Reviewer #3

(Remarks to the Authors, Responses from the Authors, and Relevant Excerpts from the Revised Manuscript):

I have no further changes to suggest. Thanks to the authors for all the work done, and congratulations on it.

We thank the reviewer very much for their congratulations and appreciate their detailed feedback throughout both previous revisions. The paper has benefited significantly from their detailed comments.

Reviewer #4

(Remarks to the Authors, Responses from the Authors, and Relevant Excerpts from the Revised Manuscript):

This is a nice piece of work, with important new results. In my opinion, it should be published as soon as possible in an interdisciplinary journal such as *Nature Comm.* because it is of interest to a wide audience (geneticists, archaeologists, mathematical

modelers, ecologists, etc.). This study is important because it is the first one (as far as I know) that uses genome-wide data to estimate the percentage of early farmers that interbred with hunter gatherers (HGs) in an agricultural expansion.

We thank the reviewer for their kind words, recommendation, and recognition of the novelty of the work. We agree that our manuscript is targeted at a wide audience and touches on many disciplines that all surround this pivotal transition in time. We appreciate the reviewer's time and detailed comments and will seek to address in the following response each valuable point the reviewer raises.

In this manuscript the observed European Neolithic ancestry cline is compared to clines obtained by means of simulations. The simulated cline that agrees best with the observed one is used to estimate the percentage of early farmers that interbred with hunter-gatherers (HGs). The most important point is that this comparison and estimation are done for the first time using the whole-genome ancestry, in contrast to previous studies that considered mtDNA and Y-chromosome haplogroup frequency clines. Obviously the whole genome has more information than haplogroups, so it is important to simulate a whole-genome ancestry cline. As far as I know, this was not done before this manuscript, in which the authors have found a solution (namely, to consider a tiny subset of the genome in the simulations). It could be argued that the inferred percentage of farmers that interbred with HGs is essentially the same by using either of both approaches, i.e., whole-genome ancestry on one hand (~ 0.001 per year, i.e., $\sim 2.5\%$ per generation from this manuscript) and a neutral haplogroup on the other (1%-8% per generation from Refs.50,69, see my minor comment below on lines 524-526 of the manuscript). It could be also argued that this agreement is unsurprising because all neutral parts of the genome should be influenced in the same way by cultural transmission, dispersal and reproduction (i.e., the 3 processes at work in the propagation of the Neolithic wave of advance). But this agreement can only be tested with a whole-genome study such as that reported in this study. And this is indeed necessary because the whole genome has more information, as mentioned above, so we have find out whether the implications are similar or not to those from partial information (haplogroups). For these reasons, I find this paper really important and timely.

We appreciate the reviewer's commentary regarding our use of genomic simulations and acknowledge the points they raise regarding the use of different types of genetic information to understand the Neolithic ancestry cline. We especially thank the reviewer for noting the importance and timely nature of the work. In the following revision we sought to address all the comments raised by the reviewer who brings additional expertise to the review. These changes have improved the manuscript significantly.

The authors have already done an utmost effort in response to 3 other reviewers. I suggest only an important, major change and minor ones. The minor ones are mainly to make the paper clearer for non-biologists and to include proper comparisons to previous results, which are largely omitted in the manuscript (so it is impossible to distinguish

what is new from what had been already done by other methods). In spite of the length of this report, I believe that the necessary work will not be huge.

We thank the reviewer for their acknowledgement of our previous work on the first two revisions to make the manuscript as strong as possible. We are committed to making the work clear for readers from all disciplines. Further, we appreciate the reviewer's expertise regarding the comparison to previous work as it certainly makes the manuscript stronger and more transparent.

MAJOR POINT:

I have only one major concern. In my opinion, the datapoints in Fig. 5B do not measure the same quantity as the simulations (blue line in the same figure). So it is not clear [to me] that both estimates of the "EF ancestry proportion" can be plotted in the same figure. However, this may not be a serious problem after all (if both ancestry estimates measure approximately the same quantity). I will not ask the authors to repeat all of their simulations (which take 4-5 days each according to their README file in the github server). But at least a check of the validity of their results should be made, as explained below. It could be simply added, e.g. as Supp. Note 3 & Supp. Fig. 17.

We are thankful for this thoughtful and constructive comment. We agree that the two approaches—ancestry inference using qpAdm (for the empirical data) and ancestry estimation based on ancestry-specific markers (in the simulations)—differ in how they define and quantify ancestry. As the reviewer notes, this raises a valid question about the comparability of the two estimates. In response, we have conducted additional analyses to assess the consistency between these methods. These results, which incorporate the reviewer's suggestions, are presented in our response below.

In the simulations, it seems that all farmers have initially all $247 \cdot 2 = 494$ markers equal to "1", and all HGs have all 494 markers equal to "0" (README file). According to p. 34 the authors estimate the "EF ancestry proportion" from a simulation run by counting the number of "1" markers for each individual and dividing by 494. After grouping farmers in bins, they obtain the blue line in Fig. 5B. It seems to me that this procedure is very different from the qpAdm software, which yields the datapoints in the same figure. Indeed, qpAdm requires reference populations and outgroups (to obtain the datapoints in Fig. 5B), whereas counting the proportion of "1"s (blue line) does not. So it seems clear that the datapoints and blue line in Fig. 5B do not measure exactly the same quantity. The datapoints in Fig. 5B will be different if we choose different populations as sources and/or outgroups, whereas the blue line will be the same. So the blue line that best agrees with the datapoints will also be different if we choose different populations as sources and/or outgroups. Thus the inferred percentage of early farmers that interbred with HGs will also be different (and this is the main result concerning human behavior). Therefore, in my opinion at least one check is necessary to make sure that the percentage of early farmers inferred by the authors is similar to that inferred by some procedure that makes it possible to estimate exactly the same "ancestry" both for the

data (datapoints in Fig. 5B) and the simulations (blue line in Fig. 5B). If qpAdm could be applied to 494 markers, the solution would be to repeat Fig. 5B by applying qpAdm both to 494 positions of the ancient farmers (datapoints) and to the 494 positions of the simulated farmers. This should be done with the same source and outgroup populations in both cases (simulations or blue line in Fig. 5B on one hand, and real individuals or datapoints in Fig. 5B on the other). In this new simulation (just for this check), I believe that you cannot assume that all farmers have initially all 247·2=494 markers equal to "1", neither that all HGs have all 494 markers equal to "0". In full rigor, this does not seem to correspond strictly to the real initial conditions, i.e., those implied by the ancient DNA data (21 farmers and 19 HGs, from p. 41). Instead, for each initial individual I suggest to use his/her true, real genome from the ancient DNA data, i.e., a number (1, 2, 3 or 4) corresponding to each nucleotide (T, C, G or A) for the 494 markers that you consider, i.e., one per million of human chromosome 1 (line 737). To ensure reproducibility, please add a new Supp. Data file with the 494 markers (nucleotide position and base) of the 21+19+618 individuals and the qpAdm ancestries obtained from them. If your simulation begins with e.g. 210 farmers, 10 of them should have the same 494 nucleotides as the ancient farmer I2532.AG (Supp. Data File 1), 10 other farmers should have the genome of the ancient farmer I2533.AG, etc. If over the landscape there are initially e.g. 1,900 HGs, 100 of them should have the genome of the HG I4971.AG, etc. If qpAdm does not work with 494 markers, I suggest the following procedure (but the authors may prefer a different one). Consider 494 nucleotide positions (one per million of human chromosome 1, from line 737) that satisfy the following condition. For each nucleotide position, none of the 21 farmers of the EF source population has the same nucleotide as any of the 18 HGs of the WHG source population (this makes it easy to estimate ancestry, as explained below, and can be justified because markers shared by some farmers and HGs can provide less information on acculturation than markers that are different for farmers and HGs). Each marker will be e.g. 1, 2, 3 or 4 (i.e., base T, C, G or A). To ensure reproducibility, please add a new Supp. Data file with the 494 markers of the 21+19+618 individuals (nucleotide position and base). For each of the 618 farmers, count how many of the 494 positions have the same nucleotide as at least one of the initial 21 farmers (let us call this number f) and how many of the 494 positions have the same nucleotide as at least one of the 18 initial hunter-gatherers (h). Then compute the farmer ancestry proportion of this farmer as $f/(f+h)=f/494$. Repeating the same procedure for the other 617 farmers and plotting these datapoints as a function of distance should lead to an observed cline (similar to that of the datapoints in Fig. 5B). Please include these ancestries for the 618 individuals in the new Supp. Info. File mentioned above. Next run a simulation as described in the last two sentences of the previous paragraph (so that the initial genetic conditions agree with the aDNA data). For each of the final simulated farmers, estimate her/his farmer ancestry proportion as $f/(f+h) = f/494$. Due to cultural transmission, some of the HG nucleotides will appear in the final farmers, and their proportion will increase with distance, so this farmer ancestry $f/(f+h)=f/494$ (proportion of the initial farmer nucleotides) will decrease with increasing distance (new blue line, analogous to that in Fig. 5B). In this way you can repeat Fig. 5B without using qpAdm, neither for the simulated nor for the 618 real ancient farmers. This new figure, analogous

to Fig. 5B, should not necessarily replace Fig. 5B. For example, you can include it as Supp. Fig. 17 (if you do not want to repeat all simulations). From this new figure, you can find out the learning rate. Admittedly, this is a very simple approach from a conceptual point of view. But the main point is that it applies the same methodology to the simulated farmers and the real ancient farmers, so that both ancestries can be plotted in a figure and compared. In fact, such a procedure is not so different to what you have done for the simulated farmers. The main points are that in this procedure the initial conditions agree with the aDNA data and that you do not have to use different definitions of ancestry for the simulated and observed clines, so you can plot both of them in the same figure and compare them (because they surely measure exactly the same quantity). Alternatively, you can follow a different procedure as long as it uses the same methodology to compute the ancestry for the simulated farmers and the real ancient farmers. It would be also nice to compare a little bit these ancestries for the 618 farmers to the corresponding ones from qpAdm (is the average % difference between them small? Is there a high correlation between them? Are both clines similar?). Independently of whether you apply the method in the last paragraph of that in the previous one, the new simulated and observed clines (obtained using the same definition of EF ancestry) should be used to find out learning rate, which should be compared to that found by the authors (about 0.001 per year). In case the authors considered it totally impossible to apply a single procedure to find out both the datapoints and the blue curve in Fig. 5B, I would still recommend publication of this paper in *Nature Comm.* (because it would nevertheless be an important contribution as a first step to solve this problem), with the only condition that the authors admit in the main text that such a test would be necessary to make sure that the datapoints and blue line in Fig. 5B measure approximately the same ancestry, with a sufficient degree of approximation so that the estimation of the learning rate is reliable.

We appreciate the reviewer's central concern that our simulations and empirical analyses use different approaches to estimate ancestry: The simulations estimate ancestry directly using ancestry-specific markers, while the empirical data relies on qpAdm, which infers ancestry proportions from genome-wide allele frequency differences. Thus, it is a valid question whether both approaches capture the same underlying signal.

We argue that our approach is accurate and appropriate, based on four main considerations:

1. Two recent studies co-authored by members of our team (Flegontova et al., 2025; Williams et al., 2024) have extensively evaluated the accuracy and robustness of qpAdm. In the specific case of estimating Neolithic farmer and hunter-gatherer (HG) ancestry in European populations, the approach is expected to perform well. This is because (i) the genetic differentiation between the two sources is relatively high ($F_{ST} \sim 0.09$), and (ii) the archaeological and temporal context supports the test assumptions of qpAdm—namely, that there is no gene flow from the test population into one of the proxy

source populations after admixture, since the source populations were sampled earlier in time. In short, all key recommendations laid out in Flegontova et al. (2025) for avoiding model violations are satisfied in our analysis, and qpAdm should yield accurate estimates of ancestry proportions.

2. The simulation approach, which uses ancestry-specific markers to directly compute ancestry proportions, is a standard and widely accepted method in simulation studies (e.g., Kim et al., 2018)—it provides a controlled way to track ancestry in synthetic genomes.

3. We acknowledge that the choice of proxy source population can influence ancestry estimates. However, we now demonstrate that using Anatolian early farmers instead of Balkan early farmers (EF) yields nearly identical ancestry clines and learning rate estimates. In response to a previous reviewer’s suggestion, we adopted the Balkan EF as a more appropriate proxy. However, we now emphasize in the revised Methods section that our results are robust to this choice (lines 1028-1033):

“To assess the robustness of our ancestry estimates to the choice of source population, we repeated the qpAdm⁶² analysis using Anatolian early farmers (n = 26) instead of Balkan early farmers (n = 21) as the proxy for Neolithic ancestry. We found that the resulting ancestry proportions produced similar spatial clines and learning rate estimates (see Supplementary Fig. 18). This suggests that our inferences are not sensitive to the specific choice of early farmer reference group, and supports the robustness of the qpAdm-based ancestry estimates.”

4. We carefully considered the reviewer’s suggestion to bring the two approaches into closer alignment. While the first option—running qpAdm on only 247 markers—is not feasible (qpAdm has insufficient power at such a low marker number; Williams et al., 2024), we pursued the second suggestion. Specifically, across all variants in our dataset, we identified a total of 53 variants that are fixed for alternate alleles in the EF (n=21) and HG (n=18) samples. We then used these variants as ancestry markers to estimate EF ancestry in the 618 Neolithic individuals, counting the number of EF-specific alleles per individual (as suggested by the reviewer). This mirrors the ancestry estimation in our simulations. Reassuringly, the marker-based estimates are strongly correlated with the qpAdm estimates ($r = 0.56$, $p < 0.001$), and the resulting cline of EF ancestry is similar (see figures below). Most importantly, when we used this cline to estimate the learning rate via least-squares fitting, we obtained a rate of 0.00103 per year—closely matching the 0.000823 per year estimate derived from the qpAdm-based cline. Thus, even without using qpAdm, the learning rate inferred from ancestry-specific markers is highly consistent with our previous estimate.

Conclusion:

While qpAdm estimates ancestry indirectly through genome-wide allele frequency differences, and our simulations use a direct marker-based approach, both methods yield consistent estimates of the spatial learning rate (~0.001 per year). We now explicitly discuss these differences and the steps we took to validate the consistency of both approaches in the revised manuscript (Discussion, lines 654-665):

"A potential limitation of our study is that the simulated and empirical ancestry estimates are derived using different approaches: The simulations use ancestry-specific markers to directly compute ancestry, while the empirical data relies on qpAdm⁶², which infers ancestry from genome-wide allele frequency differences. While these methods are conceptually distinct, we show that they yield consistent results. Specifically, we identified 53 ancestry-informative markers that are fixed between early farmer and hunter-gatherer individuals in our samples and used them to directly estimate Neolithic ancestry across our dataset. These marker-based estimates are strongly correlated with qpAdm⁶²-based ancestry estimates ($r = 0.56$, $p < 0.001$; Supplementary Fig. 15), and the resulting cline of farming ancestry yields a nearly identical learning rate estimate (0.00103 per year) to that obtained using qpAdm⁶² (0.000823 per year). This convergence across methods suggests that our main results are robust to how ancestry is defined and quantified. We provide the ancestry marker set and derived ancestry values for all individuals in the Supplementary Data 3."

We also provide the list of 53 ancestry-specific markers, including allele states for all 657 EF/HG/Neolithic individuals and the resulting ancestry estimates, in the supplementary material.

MINOR POINTS:

Lines 84-87: "Previous estimations have suggested that cultural effect could account for anywhere between 0% and ~40% of the expansion speed of farming³¹, with others showing a maximum cultural effect of 21% to be consistent with the observed front speed³³." For readers to understand these results, I suggest to add: "These differences are mainly due to assumptions on the dispersal behaviour of early farmers and mode of cultural transmission."

At the suggestion of the reviewer, we have changed lines 84-88 to read (changes in red):

"Previous estimations have suggested that cultural effect could account for anywhere between 0% and ~40% of the expansion speed of farming³⁴, with others showing a maximum cultural effect of 21% to be consistent with the observed front speed³³. **These differences are mainly due to assumptions on the dispersal behavior of early farmers and mode of cultural transmission.**"

Also in line 84: "between 0% and ~40% of the expansion speed of farming³¹...". Here ref. 51 is more appropriate than 31.

We have changed the citation number to represent the more appropriate reference. We thank the reviewer for pointing this out (note citation numbers have changed and it is no longer 51 it is 34).

This line now reads (changes in red):

"Previous estimations have suggested that cultural effect could account for anywhere between 0% and ~40% of the expansion speed of farming³⁴..."

Lines 126-127: "We conclude that there must have been near complete within-group mating in farming and hunter gathering groups." For some readers it would be clearer to add: "(i.e., very few farmer-HG matings)."

This point raised by the reviewer increases the clarity of this sentence and we appreciate the suggestion. We have now added the suggested wording to the manuscript:

Lines 126-128 now read (changes in red):

"We conclude that there must have been near complete within-group mating in farming and hunter gathering groups **(i.e., very few farmer-HG matings).**"

Lines 146-147: "prior models addressing the spread ... of the farmer-specific mitochondrial haplogroup K50. Importantly, our model extends beyond these by incorporating the genetic ancestry of individuals at a single genetic locus, enabling us to explore...". I believe that Refs. 50, 51, 69 also incorporate the genetic ancestry of

individuals at a single locus (indeed, mtDNA behaves as a single locus). A slightly modified write-up would solve this inconsistency, e.g.: "prior models addressing the spread ... of the farmer-specific mitochondrial haplogroup K50. Our basic model, similarly to previous ones^{50,51,69}, incorporates the genetic ancestry of individuals at a single genetic locus or marker, enabling us to explore ..."

We have changed the text based on the suggestion from the reviewer. Lines 145-150 now read (changes in red):

"Our model consists of a system of partial differential equations that mirror those utilized in prior models addressing the spread of agriculture (e.g., Fort et al., 2012)³¹ and of the farmer-specific mitochondrial haplogroup K⁵¹. Our basic model, similarly to previous ones^{34,51,56}, incorporates the genetic ancestry of individuals at a single genetic locus or marker, enabling us to explore how cultural transmission and demographic processes jointly shaped the ancestry landscape of prehistoric Europe."

Figure 1A: In my opinion $(u+v+w)/KF$ and $(u+v+w)/KHG$ in these 3 equations and the mathematica code should be replaced by $(u+v)/KF + w/KHG$ (see, e.g., Eq. (11) in Ref. 35). However, it is possible that this would not change Fig. 1B much and, in any case, no quantitative comparison to the observed cline is made for this basic (reaction-diffusion one-dimensional) model. So I will let the authors to choose between introducing this change or not.

The logistic growth component of our model serves as a simple heuristic to reflect the assumption that the hunter-gatherer lifestyle supports a much lower population density than farming. In our formulation, population density is defined as the sum of both farmers and hunter-gatherers, without distinguishing between them. This setup naturally leads to the expansion of the farming population displacing the hunter-gatherer population—capturing a key dynamic we aim to model. If we were to follow the reviewer's suggestion and replace the logistic terms in all three equations with " $(u + v)/KF + w/KHG$ ", the differential growth advantage of farmers would disappear, and no displacement of hunter-gatherers would occur.

That said, we also tested alternative formulations where farmers and hunter-gatherers occupy partially distinct niches and thus compete less directly for resources. Specifically, we modified the terms to " $(w + c(u + v))/KHG$ " in equation 3 and " $(c w + u + v)/KF$ " in equations 1 and 2, with "c" ranging from 0 to 1 to represent the degree of competition. We found that only when "c" is very small—i.e., when competition is nearly absent—do the resulting ancestry clines change significantly. Based on these findings, and in accordance with the reviewer's suggestion, we opted to to keep the current setup of the equations. The exact formulation of the logistic terms appears to have only a minor impact on the overall ancestry dynamics, and we do not use this model for our inference of learning rate.

For the basic (reaction-diffusion one-dimensional) model, the caption to Fig. 1 says that these clines are for [the first value of time such that] $w=0$, i.e., when all HGs have acculturated (by the way, for clarity this could be also mentioned in the main text). But line 715 says that the final time is 150 generations. If this is the first value of time when $w=0$ (approximately, as it depends on position), it should be mentioned in both places. If not, please correct.

This is a good point by the reviewer. The final time shown in the plot is indeed 150 generations, but the farming population has effectively reached fixation by around 100 generations. We tested plotting the clines at 100 generations and found that the results are identical. We'll clarify this in both the main text and the figure caption to avoid confusion:

Figure 1 caption now reads (changes in red):

“Figure 1: One-dimensional model

(A) Our mathematical model utilizes a reaction-diffusion framework to simulate the spread of EF ancestry across a one-dimensional European landscape initially populated by hunter-gatherers. It comprises a set of partial differential equations that describe the movement (diffusion) of individuals, their population growth (logistic), and the impact of cultural transmission on genetic ancestry. For a comprehensive description of the symbols and components of the equations, see Supplementary Note 1. The model tracks three distinct groups: farmers with EF ancestry, farmers with WHG ancestry, and hunter gatherers with WHG ancestry, each affected by diffusion constants, growth rates, and carrying capacities. EF ancestry is calculated as $u/(u+v+w)$, when farming is fixed and $w=0$. The cultural transmission aspect is modeled through parameters that dictate the learning rate of new practices (f) and a bias toward adopting farming methods from either farmers or hunter-gatherers (γ). (B) Relationship between the ratio of the learning rate to the bias parameter ($C = f/\gamma$) and the proportion of EF ancestry in the model, averaged across the landscape. For this one-dimensional model, we calculated the EF ancestry proportion at generation 100, shortly after all HGs have been acculturated. This plot illustrates that the final proportion of EF ancestry after the Neolithic expansion is predominantly determined by the ratio C , rather than the individual values of f or γ . Data points indicate model outcomes across a range of learning rates (f) and bias parameters (γ).”

Lines 163-165 in the Results section now read (changes in red):

“We solved the model equations numerically to investigate the resulting change in farming ancestry, shortly after all HGs have been acculturated, over a time span of 100 generations and across the 3000 km transect.”

Lines 761-764 in the Methods section now read (changes in red):

“We set up the partial differential equations model in Mathematica 13.3⁸⁸ and numerically solved it over a time span of 100 generations (i.e., ~2500 years when assuming a generation time of 25 years), which was shortly after all HGs had been acculturated, using the Mathematica function *NDSolve*.”

Lines 271-273: "We conclude that, although the front speed is informative about the step size parameter of the model, it does not provide fine-scale insight into the degree of cultural transmission." It is important to compare this point to previous work, e.g.: "This conclusion agrees with the wide ranges of the cultural effect calculated from the front speed by using intergenerational dispersal distances and probabilities measured for ethnographic preindustrial farming populations⁵¹ instead of a normal distribution as assumed in the present paper."

We have modified the text based on the suggestions from the reviewer. Lines 274-281 now read (changes in red):

“We conclude that, although the front speed is informative about the step size parameter of the model, it does not provide fine-scale insight into the degree of cultural transmission. This conclusion agrees with the wide ranges of the cultural effect calculated from the front speed by using intergenerational dispersal distances and probabilities measured for ethnographic preindustrial farming populations³⁴. This is notable because it highlights the importance of co-analyzing ancestry and front speed in a combined model. In subsequent analyses, we explore the relationship between cultural transmission and genetic ancestry patterns and evaluate if ancestry patterns allow the estimation of cultural transmission parameters.”

Lines 278-283: For the agent-based model, please mention the value(s) of time for the final simulated farmers (i.e., those for which you have computed the blue line in Fig. 5B). In principle, for the comparison between the blue line and the datapoints to be valid, the simulation results should correspond (as closely as possible) to the same time as the radiocarbon dates of the 618 ancient farmers. So I would have perhaps preferred to group the datapoints in Fig. 5B in regions and apply the mean radiocarbon date of the ancient farmers in each region to the simulations (blue line). But there is no need to make any of these changes, because the results could be much the same. However, please check that the simulated values (blue line in Fig. 5B) are essentially the same for the whole range of radiocarbon dates. If this is so, please mention it in the main text. Otherwise, please try to find a solution (that proposed 3-4 lines above or another one).

This is a good point raised by the reviewer. However, we are confident that the comparison between the point estimates from ancient individuals and the simulated ancestry cline is valid. We note here that our empirical qpAdm filtering strategy employed excludes individuals exhibiting negative farming ancestry (and negative ancestry generally) thereby constraining the computed empirical ancestry cline to

individuals possessing both EF and WHG source ancestries. This filtering constraint aligns with our agent-based model simulations, wherein estimates derived from final simulated farmers likewise represent admixtures of EF and WHG sources.

These filtered ancient individuals we use for comparison indeed span a wide portion of the Neolithic, covering a time range of 3,379 years. This range encompasses nearly the full duration of the simulated expansion, rather than corresponding to a single snapshot in time as in the simulation. Despite this, our filtering maintains the alignment with the simulated data points.

As we discuss in lines 634-653 and demonstrate in Supplementary Figure 11, fitting the model to subsets of the ancient samples drawn from different Neolithic time periods has minimal effect on the inferred learning rate. This suggests that our results are robust to variation in sampling time within the Neolithic.

Further, as shown in Supplementary Figure 11, the simulated ancestry cline remains remarkably consistent throughout the simulation, regardless of the year chosen for ancestry sampling. The main difference across sampled years is the distance that farming ancestry has progressed across the landscape, not the shape or slope of the cline itself. Thus, drift appears to have only a minimal effect on the established ancestry cline over the Neolithic time period.

For our analysis of the simulated individuals' ancestry, we selected the cline from the end of the expansion—approximately 3,500 years into the simulation—when farming had spread across the entire landscape and hunter-gatherers were no longer present. (The exact number of years was variable depending on parameters of the simulation). This cline is shown as the blue line in Figure 5B and represents the final state of the expansion, which also aligns with the expected expansion speed. In the specific case of Figure 5B the blue line represents 3666 years after the start of the expansion.

Finally, to further assess whether the cline might change after the expansion is complete—such as through backward diffusion of farming ancestry into previously established regions—we extended the simulation for an additional ~1500 years beyond the point when farmers had become ubiquitous across the landscape. Consistent with the trends shown in Supplementary Figure 11 (which sampled during the expansion), the cline remained stable during this extended period after the expansion. This further increases our confidence that the comparison between the simulated cline and the ancient samples is valid and robust.

In the figure below, we show multiple clines plotted in the same fashion as the blue line on Figure 5B. These represent samples taken the year farming was fixed (which in this simulation happened after 3663 years) and then samples every 200 years, for ~1500 years. The clines remain nearly identical, as expected from the trends shown in Supplementary Figure 11.

For color-blinded people, it would help a lot if you could please use the yellow and orange colors for the two upper (or two lower) curves in Figs. 4a and especially 4b.

We have changed the 8 figures in the main text and supplement that use this color palette to include yellow and orange for the two upper curves as the reviewer suggests. Figure 4, for example, has been changed (below) based on the reviewer's suggestion:

Also, Fig. 4b would be much clearer if it were possible to invert the order of the legend colors and values, so that 1 corresponds to the upper line and 0.7 to the lower line.

We thank the reviewer for this suggestion that allows for better interpretation of the plots. We have now inverted the colors and for the 8 figures that use this color palette, in all cases, the parameters that yield the highest retention of EF ancestry following the farming expansion are orange and yellow—regardless of their numerical value. See Figure 4b in the comment above.

Lines 310-312: It seems to me that this has been already discussed in line 288. Please remove one of both, or cite the relevant figure in line 312 (Fig. 4a again?).

The initial goal of the point regarding the learning rate was to point out the analogous effect between within-group mating and learning rate on ancestry patterns. We see the redundancy the reviewer pointed out and have made the following changes (red) to make the text more succinct. We also added references to the relevant figures as the reviewer suggested:

“We found that low levels of between-group mating (i.e. HG with farmer) lead to similar ancestry clines (Fig. 4b) as the simulations with variable learning rates (Fig. 4a). This shows that both vertical and horizontal cultural transmission are analogous in their genetic effects. We further saw a rapid loss of EF ancestry when we decreased the within-group mating rate (i.e., increased the between-group rate), such that with within-group mating rates less than 90%, EF ancestry quickly declined to zero toward the far end of the range (Fig. 4B) analogous to what we observed with learning rates greater than 0.01 (Fig. 4A).”

Line 317: "we conclude that cultural transmission must have only played a limited role in the farming expansion." This should be compared to other studies, e.g.: "This conclusion is also supported qualitatively by the major genetic [Bramanti et al. 2009] and genomic [Skoglund et al. 2012] turnover in Europe at the time of the Neolithic transition, as well as quantitatively by the low intensities of cultural transmission implied by the shapes of mitochondrial and Y-chromosome clines in Neolithic Europe^{50,51,69}."

The suggestion from the reviewer provides valuable support for our conclusions. We have altered the main text in accordance with the reviewer's suggestion.

Lines 318-325 now read (changes in red):

“In sum, both horizontal and vertical cultural transmission can generate spatial clines in EF ancestry after the farming expansion, with higher rates of cultural transmission leading to steeper clines and lower final proportions. Given that the proportion of EF ancestry estimated from aDNA in post-Neolithic Europe is considerably high, we conclude that cultural transmission must have only played a limited role in the farming expansion. This conclusion is also supported qualitatively by the major genetic⁵⁹ and genomic⁶⁰ turnover in Europe at the time of the Neolithic transition, as well as

quantitatively by the low intensities of cultural transmission implied by the shapes of mitochondrial and Y-chromosome clines in Neolithic Europe^{34,51,56}.”

Figure 5B: In my printed version, I can barely see the 95% CI bars around the datapoints, because the bars are very thin. Can this be improved?

We thank the reviewer for this suggestion regarding how the figure appears while printed. We have made the error bars both wider and more opaque in this revision so that they can be seen better, particularly when printed. We also made the pink regression line and the blue simulation line wider so they stand out from the points and now darker and wider updated error bars.

The revised Figure 5 can be seen below:

Lines 369-375 and Methods, lines 948-955: Both paragraphs are impossible to understand for me. Please re-write and expand (possibly in a Supp. Info. note). How is the likelihood of each sample (lines 17) computed from the observed an expected EF proportions (line 951)? Why should it be multiplied across sites (line 373), i.e., why should the log-likelihood be summed up across sites (line 952)? Where can we find the corresponding mathematical derivations? How are the standard error rates of EF from qpAdm taken into account (line 372)? Why is the standard error (line 375) computed as the square root of the negative inverse of the second derivative in Supp. Fig. 5 (line 955)?

We acknowledge that the descriptions in the original manuscript lacked the necessary detail for a full understanding of the statistical procedures used. Thus, we have now created a new, detailed Supplementary Note, "Supplementary Note 3: Statistical Inference of the Cultural Learning Rate," which provides comprehensive explanations of the statistical framework, including all relevant mathematical derivations. Below, we briefly respond to the reviewer's specific questions.

Regarding how the likelihood of each sample is computed from observed and expected

Early Farmer (EF) proportions—this is based on the assumption that the empirical EF ancestry proportion for a sample (y_i ; estimated with qpAdm) is drawn from a normal distribution. The mean of this distribution is the EF ancestry predicted by our model for a given learning rate f , and its standard deviation is the sample-specific standard error from the qpAdm analysis ($y_i \sim N(\mu_i(f), \sigma_i^2)$). The likelihood for sample i is the probability density of observing the empirical data (y_i) if the true mean were the model's prediction $\mu_i(f)$, calculated as:

$$L_i(f|y_i, \sigma_i) = \frac{1}{\sigma_i \sqrt{2\pi}} \exp\left(-\frac{(y_i - \mu_i(f))^2}{2\sigma_i^2}\right)$$

The multiplication of likelihoods across samples (or summing of log-likelihoods) is justified by the assumption that estimation errors of EF ancestry are independent across different archaeological samples. If errors are independent, their joint probability is the product of individual probabilities. For numerical stability and mathematical convenience, we work with log-likelihoods, transforming this product into a sum:

$$LL(f) = \sum \log(L_i(f)).$$

The standard error rates of EF ancestry from qpAdm are incorporated by using the sample-specific standard error (σ_i) as the standard deviation parameter in the normal distribution assumed for the observed EF ancestry at that site, as shown in the likelihood formula above. A larger σ_i results in a flatter likelihood for that sample, giving it less weight in the overall estimation, while a smaller σ_i leads to a sharper peak and more influence.

Finally, the standard error of the maximum likelihood estimate (MLE) is computed as the square root of the negative inverse of the second derivative of the log-likelihood function. This is a standard result from likelihood theory related to Fisher Information. In our case, to compute the second derivative, the log-likelihood function is approximated by a quadratic fit, $LL(f) \approx af^2 + bf + c$, leading to $SE[\hat{f}] \approx \sqrt{-1/(2a)}$. Intuitively, the curvature of the log-likelihood function at its peak (captured by the second derivative, $2a$) reflects the precision of the MLE. A sharper peak (larger negative $2a$) implies a smaller variance and standard error. This method relies on the asymptotic properties of MLEs and the adequacy of the quadratic approximation to the log-likelihood surface near its maximum, as shown in Supplementary Fig. 5.

A full derivation, with more mathematical details than the present response text, is now in Supplementary Note 3 which we reference in the Results section on lines 381-385:

“Likelihood values across samples were multiplied, assuming that estimation errors are independent across sites. A quadratic polynomial was then fit to the likelihood values to derive the maximum likelihood estimate and standard error of the learning parameter (Supplementary Fig. 5; details in Supplementary Note 3).”

And in the Methods on lines 1016-1017:

“...derivative (Supplementary Fig. 5; (see Supplementary Note 3 for full derivation of the statistical approach)”.

We once again thank the reviewer for their insightful comments, which have significantly helped us to improve the clarity and rigor of our manuscript.

Is this method really necessary? Please explain why you do not use a simpler method to estimate the value and range of the learning rate, e.g. just to minimize the sum of the magnitudes of the differences between the observed and simulated ancestries over samples (the error of the learning rate could be also estimated, e.g., by repeating estimations with values taken from the qpAdm error range, i.e., from the bars in Fig. 5B). Would the results be so different to justify the use of a method that is unknown to many readers of an interdisciplinary journal? If you are not sure, please write that such an alternative, simpler method could also work. Otherwise, many readers (including me) can be confused.

We agree that a simpler alternative could also work. In fact, we tested a standard least-squares approach—which is arguably the most common non-parametric method for curve fitting—and found that it yields nearly identical estimates of the learning rate ($f = 0.000823$) compared to our likelihood-based method ($f = 0.000798$). See also the figures below. This is not surprising, as both approaches are known to converge to the same estimate under standard regression assumptions.

We chose to retain our likelihood-based framework because it offers the advantage of explicitly accounting for heterogeneity in the uncertainty of the ancestry estimates (derived from qpAdm), which we believe is important in this context. We also now provide a detailed explanation of our method and its justification in Supplementary Note 4, which we hope makes it more accessible to readers unfamiliar with this general approach.

Lines 376-378: I believe that the words in UNDERLINED CAPITAL LETTERS should be added: "... to an estimated learning rate of 0.000798 [0.000787, 0.000808] FOR 100% WITHIN-GROUP MATING (Fig. 6; Supplementary Fig. 5a). Using the same approach and the simulated clines in Fig. 4b, we also derived an estimated within-group mating parameter of 0.9835 [0.9833, 0.9837] FOR ZERO LEARNING RATE (Supplementary Fig. 5b)."

We have added these clarifying statements that the reviewer suggests to these lines (385-388) (changes in red):

"... to an estimated learning rate of 0.000798 [0.000787, 0.000808] for 100% within-group mating (Fig. 6; Supplementary Fig. 5A). Using the same approach and the simulated clines in Fig. 4b, we also derived an estimated within-group mating parameter of 0.9835 [0.9833, 0.9837] with a learning rate of zero (Supplementary Fig. 5B)."

Lines 457-461: "In sum, despite very different landscape and expansion models, we consistently estimate a yearly learning rate of ~0.1% (Fig. 6). Given this low rate, it suggests that at the tip of the wave-front, only 1 in 1,000 farmers converted 1 HG each year (see Methods). This estimate is robust to modeling assumptions and two orders of magnitude smaller than previously estimated upper limits of the learning rate based on the speed of the farming expansion alone^{31,33}." Your result should be also related to other previous estimations of the learning rate, e.g.: "Our learning rate of ~0.1% per year or ~2.5% per generation of 25 years is consistent also with the range 1%-8% per generation for the learning (and/or interbreeding) rate previously estimated by comparing clines of mitochondrial and Y-chromosome haplogroups to spatial simulations with a generation time of 32 years^{50,69}."

In response to the reviewer's comment we have modified the text based on their suggested wording.

Lines 467-475 now read (changes in red):

“In sum, despite very different landscape and expansion models, we consistently estimate a yearly learning rate of $\sim 0.1\%$ (Fig. 6). Given this low rate, it suggests that at the tip of the wave-front, only 1 in 1,000 farmers converted 1 HG each year (see Methods). This estimate is robust to modeling assumptions and two orders of magnitude smaller than previously estimated upper limits of the learning rate based on the speed of the farming expansion alone^{31,33}. Further, our learning rate of $\sim 2.5\%$ per 25-year generation which we calculate using genomic data is within the learning (and/or interbreeding) range of 1%-8% per generation previously estimated by comparing clines of mitochondrial and Y-chromosome haplogroups to spatial simulations with a generation time of 32 years^{51,56}.”

Lines 503-505: "geographic ancestry patterns remain largely unchanged during the expansion—while still resulting in a process that is predominantly demic in nature (cultural effect < 50%)." A more detailed explanation on this result and its relationship to previous ones is in order, e.g.: "The reason is that for low values of the learning (and/or interbreeding) rate f , the spread rate varies very slowly with f so the spread is overwhelmingly demic^{31,51} but, in sharp contrast, the genetic ancestry cline varies very fastly with f ^{50,69} so there is not a genetic ancestry turnover (except near the origin of the spread) e.g. for $f \approx 0.1$ per generation⁵⁰. We stress that such a lack of genetic turnover for a mainly demic process (cultural effect < 50%) does not happen for the spread of the Neolithic in Europe (in this case, Fig. 5b implies a genetic turnover with $f \approx 0.001$ per year, i.e., $f \approx 0.025$ per generation." In my opinion, the last sentence is important because otherwise many readers may understand that this point refers to the spread of the Neolithic in Europe.

These lines 515-525 have been changed based on the reviewer's clarifying suggestion and now reads (changes in red):

“However, our results indicate that there is a broad range of cultural transmission rates (i.e., learning rates f) that prevent significant genetic ancestry turnover—i.e., geographic ancestry patterns remain largely unchanged during the expansion—while still resulting in a process that is predominantly demic in nature (cultural effect < 50%). The reason for this is that for low values of f , the expansion rate varies slowly with f so the spread is overwhelmingly demic^{31,34} but, in contrast, the genetic ancestry cline varies rapidly with f ^{51,56} so there is not a genetic ancestry turnover (except near the origin of the spread) e.g. for $f \approx 0.1$ per generation⁵¹. We note, however, a lack of genetic turnover for a mainly demic process (cultural effect < 50%) does not happen for the spread of the Neolithic in Europe (in this case, Fig. 5D implies a genetic turnover with $f \approx 0.001$ per year) but may be applicable to other cultural expansions classified as not being demic in nature.”

Lines 524-526: "This rate (ABOUT 0.1% OF FARMERS CONVERTING A HG PER YEAR) is significantly more precise than inferences from earlier studies based solely on the speed of the expansion, which estimated that up to 10% of farmers each year were converting HGs, assuming a 25-year generation time³¹." This is another statement that should be related to previous work by adding a sentence, e.g.: "Our result of ~0.1% per year or ~2.5% per generation is consistent also with the range 1%-8% previously estimated from Neolithic haplogroup clines by taking into account the uncertainties in the parameter values^{50,69}."

We have made the changes to wording as suggested by the reviewer to both clarify and better relate to previous literature.

Lines 544-548 now read (changes in red):

"This rate of about 0.1% of farmers converting a hunter gatherer per year is significantly more precise than inferences from earlier studies based solely on the speed of the expansion, which estimated that up to 10% of farmers each year were converting HGs, assuming a 25-year generation time³¹. Our result of ~0.1% per year or ~2.5% per generation is consistent also with the range 1%-8% previously estimated from Neolithic mitochondrial haplogroup clines^{51,56}."

Lines 529-532: "We estimate that the cultural effect is negligibly small (0.5%), whereas prior studies have reported estimates up to 40%. However, these prior estimates fail to consider the distinct ancestry patterns that would result from such high levels of cultural transmission." This text should be replaced to provide a full, accurate comparison to previous work, e.g.: "We estimate that the cultural effect is negligibly small (0.5%). The first published estimations of this effect were based on analyzing the front speed and found the range 0%-40%^{31,51}. Those estimates were later refined by analyzing haplogroup ancestry patterns, leading to the range 0.7%-2.3%^{50,51}. That range is reasonably similar to our result of 0.5% because we have made use of whole-genome ancestries and assumed a normal distribution for the dispersal of individuals." (NOTE: you cite the range 0.7%-2.3% correctly in line 104 but surprisingly not here)

We thank the reviewer for pointing this out and we have now addressed the point accordingly based on the wording suggestions from the reviewer. We have also added the additional citations from lines 104 to this section.

Lines 551-556 now read (changes in red):

"We estimate that the cultural effect is negligibly small (0.5%), whereas prior studies have reported estimates up to 40%^{31,33}. However, these prior estimates failed to consider the distinct ancestry patterns that would result from such high levels of cultural transmission. As such, those estimates were later refined using haplogroup ancestry

patterns, leading to the range 0.7%-2.3%^{34,51}. This narrowed range aligns more closely with our results which make use of whole-genome ancestries.”

Lines 532-537: "Moreover, previous approaches are often underpinned by assumptions about the movement of people and how culture was transmitted. For example, they utilize the distance between spousal birthplaces to calculate dispersal ranges^{31,32,72}, or studies of modern missionary trips to determine how readily HGs convert to farming³¹, assuming that these quantities are also applicable to prehistoric populations. Our approach does not rely on such assumptions, instead fitting parameters to empirical observations of front speed and ancestry." Again, please replace this text to provide an accurate, fair comparison to previous work by recognizing that your model also makes assumptions and taking into account that missionary trips were used only in the earliest estimation³¹ but later disregarded⁵¹. In other words, in my opinion replacing this text by something similar to the following one would be much better: "We caution that our model and previous ones are based on some assumptions. For example, unfortunately there are no data on individual dispersal distances in prehistory, but such data are absolutely necessary in any spatial model of Neolithic spread. Previous authors^{29,31,72} used parent-child birthplace and spousal birthplace distances measured for present preindustrial populations. Those distance distributions do not resemble a normal distribution, as assumed by us, but comparing both approaches is of interest and justified by the lack of prehistoric dispersal distances. Another source of uncertainty in the earliest models³¹ was the learning or interbreeding rate, but recently the use of genetics^{50,69} and genomics (present paper) has made it possible to estimate this parameter directly from prehistoric data. Similarly, it has been proposed that prehistoric dispersal distances could be measured in the future by identifying parent-child pairs buried in different places^{32,51}."

These clarifications from the reviewer are important and we appreciate the reviewer's well written and well thought out wording suggestion.

We have replaced this section to reflect the reviewer's suggestion.

Lines 557-567 now read:

"We caution that our model and previous models are based on some assumptions. For example, there is no data on individual dispersal distances in prehistory, but such data are absolutely necessary in any spatial model of Neolithic spread. Previous authors^{29,31,74} used parent-child birthplace and spousal birthplace distances measured for present preindustrial populations. Those distance distributions do not resemble a normal distribution, as assumed by us, but comparing both approaches is of interest and justified by the lack of prehistoric dispersal distances. Another source of uncertainty in the earliest models³¹ was the learning or interbreeding rate, but recently the use of genetics^{51,56} and genomics (present paper) has made it possible to estimate this parameter directly from prehistoric data. Similarly, it has been proposed that prehistoric

dispersal distances could be measured in the future by identifying parent-child pairs buried in different places^{32,34}."

Line 544: "Overall, our modeling implies that mating between cultural groups must have been remarkably rare (<3% between-group mating; Supplementary Fig. 5b)". Comparison to previous results is needed, e.g., ", in agreement with previous work^{50,69}".

We have changed this line based on their suggested wording.

Line 574-576 now reads (changes in red):

"Overall, our modeling implies that mating between cultural groups must have been remarkably rare (<3% between-group mating; Supplementary Fig. 5B), **in agreement with previous work^{51,56}**."

Lines 583-587: "Two factors may account for this discrepancy. First, our analysis utilized a larger sample size and the finer resolution of genome-wide ancestry estimates, which likely allowed us to detect subtle differences in learning rates that mitochondrial data may not capture. Second, because mitochondrial haplotypes are maternally inherited, they reflect only female ancestry patterns, potentially missing sex specific cultural interactions and movement dynamics." The second factor seems clearly wrong, because Ref.69 shows that a Y-chromosome cline (Fig. 5) is consistent with the same value as the mitochondrial clines (Fig. 3). Moreover, in my opinion the claim of a discrepancy is unjustified, because in your Fig. 6 you find a learning rate of about $0.00125/\text{yr} \cdot 25\text{yr}/\text{gen} = 0.031/\text{gen} = 3.1\%$ per generation for the South (Mediterranean individuals) and about $0.0009/\text{yr} \cdot 25\text{yr}/\text{gen} = 0.0225/\text{gen} = 2.3\%$ per generation for the North (continental individuals) without performing a sensitivity analysis of the parameter values (growth rate, generation time, carrying capacities...) over their complete ethnographic ranges, but both results are within the range 1%-8% found independently for the Mediterranean and the continental routes in the sensitivity analysis over those parameter values reported in the last 2 paragraphs and Secs. S5-S6 of Ref.69. So I see no discrepancy and my suggestion would be to change the text above by something that properly summarizes these results, e.g.: "However, there is not discrepancy if we take into account that our Fig. 6 implies a learning rate of about $0.00125/\text{yr} = 3.1\%$ per generation of 25 years for the South (Mediterranean individuals) and about $0.0009/\text{yr} = 2.3\%$ per generation for the North (continental individuals), and both values are within the range found previously from mitochondrial and Y-chromosome haplogroup clines, namely 1%-8% (as estimated by sensitivity analyses of the parameter values independently for the Mediterranean and continental clines) 69."

In response to this comment, we have revised our statement to be more cautious (lines 612-617):

"Our analysis leveraged a larger sample size and the finer resolution provided by genome-wide ancestry estimates, which likely enabled us to detect subtle differences in learning rates that may be overlooked with mitochondrial data. However, additional analyses are needed to assess the robustness of our findings to other model parameters, such as generation time and population growth rate, which may also differ between the Mediterranean and continental routes."

Lines 609-611: To increase reproducibility of the results, the final time(s) for each simulation (early, middle and late Neolithic) should be given.

This is a good point that increases the reproducibility of our study. We have changed the *empirical* dates that correspond to the time periods mentioned in the lines the reviewer suggested (now 638-642) as well as when we discuss this point in the Results (lines 448-450), and also in the figure heading of Supplementary Figure 10. We previously only mentioned this in the methods.

To clarify, the timing of the *simulation* data sampling remains the same (happening once farming has become ubiquitous) regardless of which empirical time period to which we fit. As previously mentioned in our response comment on pages 11-13 (see above), the timing of the simulation sampling does not affect the resulting distance cline.

The line the reviewer mentioned (now 638-642) reads as follows (changes in red):

"To assess the robustness of this assumption, we divided the ancient DNA dataset into Early (greater than 6500ybp), Middle (6500-5500ybp), and Late (less than 5500ybp) Neolithic subsets (Supplementary Fig. 10) and re-estimated the learning rate for each group independently using our complex landscape model."

The Results lines 448-450 read as follows (changes in red):

"We separated the 618 ancient individuals into bins corresponding to the Early (greater than 6500ybp), Middle (6500-5500ybp), and Late (less than 5500ybp) Neolithic."

Supplementary Figure 10 now mentioned this at the top of the plot (see below):

Lines 618-621: I suggest to add the text in UNDERLINED CAPITAL LETTERS in order to compare properly to previous work: "Importantly, even the higher estimates from later periods (0.0006-0.0014 PER YEAR OR 1.5%-3.5% PER GENERATION OF 25 YEARS, FROM FIG. 6) remain two orders of magnitude lower than previous upper estimates based solely on front speed³¹ AND WITHIN THE RANGE 1%-8% PER GENERATION IMPLIED BY HAPLOGROUP CLINES⁶⁹, supporting our main conclusion that cultural transmission played a minimal role in the spread of farming during the Neolithic."

We have made the changes suggested by the reviewer which provide more clarity regarding the context of our findings within previous literature.

Lines 648-653 now read (changes in red):

"Importantly, even the higher estimates from later periods (0.0006-0.0014 per year or 1.5%-3.5% per generation of 25 years, Fig. 6) remain two orders of magnitude lower than previous upper estimates based solely on front speed³¹, and within the range 1%-8% per generation implied by haplogroup clines⁵⁶, supporting our main conclusion that cultural transmission played a minimal role in the spread of farming during the Neolithic."

Lines 648-650: "Our findings challenge this binary framework, introducing a third, intermediate scenario: a predominantly demic expansion that occurs without significant ancestry turnover." This should be related to previous results, e.g.: "This third scenario is also seen in previous simulations of haplogroup clines, where some learning rates (e.g., about 10% per generation) lead to waves of advance that are mainly cultural genetically (no major genetic replacement except near the origin of the spread)⁵⁰ but mainly demic archaeologically (no significant increase in the spread rate relative to the purely demic case)³¹.

The following lines have been added based on the reviewer's suggestion.

Lines 702-706 now read:

"This third scenario is also seen in previous simulations of haplogroup clines, where some learning rates (e.g., about 10% per generation) lead to waves of advance that are mainly cultural genetically (no major genetic replacement except near the origin of the spread)⁵¹, but mainly demic archaeologically (no significant increase in the spread rate relative to the purely demic case)³¹.

Line 680: "—we significantly improve on previous estimates that suggest up to 21-40% cultural effect^{31,33}." Again, please replace this sentence by another one that describes all previous work faithfully, for example: "—we estimate a cultural effect of about 0.5%,

which improves previous estimates that did not consider the whole genome but haplogroup frequencies and led to the range 0.7%-2.3% for the cultural effect^{50,51}."

We appreciate the reviewer's continued attention to contextualizing our work within previous results. We intend to represent the previous body of work completely and faithfully and their comments have helped us refine our commentary on previous work.

In accordance with this- lines 726-728 now read:

"—we estimate a cultural effect of about 0.5%, which improves previous estimates that did not consider the whole genome but haplogroup frequencies and led to the range 0.7%-2.3% for the cultural effect^{34,51}."

Line 703: "The cultural transmission model was originally developed by Fort (2012)³¹, but without accounting for the ancestry of individuals." This is incomplete and misleading, so I suggest e.g.: "The cultural transmission model was originally developed by Fort (2012)³¹, and accounting for the ancestry of individuals at a single genetic marker by Isern et al. (2017)⁵⁰." The authors can decide if an additional sentence should be perhaps included to further clarify the previous point, e.g.: "In these models horizontal and vertical transmission are mathematically equivalent, with the only difference that the intensity of cultural transmission (C or f in the present paper) is the number of HGs converted into farmers per early farmer and generation (in horizontal transmission)²¹ or the proportion of early farmers who interbreed with a HG (in vertical transmission)^{xx}." Here Ref. XX stands for Phys. Rev. E 83, 056124 (2011), Eqs. (46) and (54).

Lines 749-751 have been modified to reflect the reviewer's comment (changes in red):

"The cultural transmission model was originally developed by Fort (2012)³¹, and accounting for haplogroup frequency at a single genetic marker by Isern et al. (2017)⁵¹."

Lines 729-731: "In each subsequent year, the ancestry proportion of new offspring is determined via sexual recombination of the individual's parental genomes". How is this exactly applied? Please explain it clearly and precisely for non-biologists (here or in the Supp. Info.), so that they do not have to look it up in Ref. 56 or other sources. Also, I suppose that rather than "the ancestry proportion of new offspring" it should be "the simulated genome of new offspring". How is it determined from those of the parents? E.g., for each simulated genome position do you choose at random the corresponding one from one of both parents? Also, please state explicitly if the software distinguished between men and women or any two simulated individuals can mate.

Lines 792-796 now have a short explanation of how SLiM models recombination, which is expanded upon in Supplemental Note 4. We now reference this new supplemental

note during the section of the main text the reviewer comments upon. The main text (lines 792-796) read as follows (changes in red):

“The genomes of new individuals are generated via simulated sexual recombination of the two parents as part of the built-in SLiM reproduction function⁵⁷, assuming a recombination rate of 1 cM/Mb. This function takes the two parental genomes and simulates crossing over between the two gametes at the specified recombination rate⁵⁷ and is explained in more detail in Supplementary Note 4.”

New Supplementary Note 4 reads as follows:

“Supplementary Note 4: Description of recombination and genome modeling in SLiM

We use the “crossover breakpoints” recombination model in SLiM, which is SLiM’s standard method for simulating genetic recombination via gametic crossing over⁸. In this model, recombination events are introduced during simulated meiosis by specifying a recombination rate per base pair. We chose a rate of 1 centimorgan per megabase (1 cM/Mb), which reflects the average recombination rate across the human genome⁹. This value corresponds to a 1% chance of a crossover event occurring during meiosis per one million base pairs. Accordingly, this leads to a 1% chance of crossover between our marker mutations that we assume to be one million base pairs apart from each other. As described in the SLiM manual⁸, this recombination rate is used when generating parental gametes to randomly place crossover breakpoints along the chromosome. To generate a gamete, SLiM¹⁰ begins copying from one of the two parental homologous chromosomes. Upon reaching a recombination breakpoint, it switches to copying from the other homolog. Thus, these breakpoints simulate crossover events in meiosis, in which genetic material is exchanged between the parental homologs. The result of this process is a recombined haploid gamete, composed of segments inherited from both parental homologs. SLiM¹⁰ then uses these gametes from each parent to assemble the diploid genome of the offspring. Of note, for simplicity our model does not assume male or female individuals, thus an individual can, in theory, mate with any other individual with the offspring number being drawn from a Poisson distribution (offspring number \sim Poisson($\lambda = 0.1$)).”

Line 736: I believe that the words in UNDERLINED CAPITAL LETTERS should be added for clarity: " To determine the ancestry proportion SIMULATED GENOME of each individual, ONE neutral marker mutations are IS initialized every megabase along the 247Mb simulated chromosome (human chromosome 1) of all farmers. THEREFORE, SINCE WE CONSIDER DIPLOID INDIVIDUALS, EACH ONE HAS A SIMULATED GENOME OF 247*2=494 CHARACTERS. ALL INITIAL FARMERS HAVE ALL 494 CHARACTERS EQUAL TO "1". These markers “tag” EF ancestry along the chromosome of each individual. ALL HUNTER-GATHERERS HAVE ALL CHARACTERS EQUAL TO "0". Accordingly, FOR FARMERS AFTER THE INITIAL TIME the proportion of EF ancestry is calculated by counting the number of "1" markers and dividing by the total number of marker locations (I.E., 494)."

We have changed this section of the methods for clarity based on the reviewer's suggestions.

Lines 784-791 in the Methods now read (changes in red):

"To determine the ancestry proportion of each simulated individual, one neutral marker mutation is initialized every megabase along the 247Mb simulated chromosome (human chromosome 1) of all farmers. Therefore, since we consider diploid individuals, each individual has a simulated genome of $247 * 2 = 494$ markers. All initial farmers have all 494 markers which are set to equal "1". These markers "tag" EF ancestry along the chromosome of each individual. All initial hunter-gatherers have all markers equal to "0". Accordingly, the proportion of EF ancestry is calculated in all individuals subsequent to year zero by counting the number of markers equal to one and dividing by the total number of marker locations (i.e., by 494)."

Lines 740-741: "The genomes of new individuals are generated via simulated sexual recombination of the two parents as part of the built-in SLiM reproduction function⁵⁶."
Same comment as above for lines 729-731: How is this exactly applied? Please explain it clearly and precisely for non-biologists (here or in the Supp. Info.), so that they do not have to look it up in Ref. 56 or other sources.

Lines 792-796 now have a short explanation of how SLiM models recombination, which is expanded upon in Supplemental Note 4. We now reference this new supplemental note during the section of the main text the reviewer comments upon. The main text (lines 792-796) read as follows (changes in red):

"The genomes of new individuals are generated via simulated sexual recombination of the two parents as part of the built-in SLiM reproduction function⁵⁷, assuming a recombination rate of 1 cM/Mb. This function takes the two parental genomes and simulates crossing over between the two gametes at the specified recombination rate⁵⁷ and is explained in more detail in Supplementary Note 4."

Please see above for our new Supplementary Note 4 (above) that now explains in detail how SLiM models recombination.

Line 742: "assuming a recombination rate of 1 cM/Mb". Please explain for non-biologists so that they do not have to look up for this: What does "recombination rate" mean? How is this rate applied? Where does this value come from? Is it well-established? Does its uncertainty or error affect the results?

This value means there is a 1 in 100 or 1% chance of a crossover event happening during meiosis per 1 million base pairs (or one megabase). This value is a well established average human genomic recombination rate (Li & Freudenberg, 2009).

Because this value selected is an average across the genome, the local variability due to recombination hot and cold spots along the genome should not affect the results as this is a global average across the genome and we consider marker mutations with a large distance between each other. We have included new Supplemental Note 4 which explains what the 1cM/Mb value means.

Lines 792-796 now have a short explanation of how SLiM models recombination, which is expanded upon in Supplemental Note 4. We now reference this new supplemental note during the section of the main text the reviewer comments upon. The main text (lines 792-796) read as follows (changes in red):

“The genomes of new individuals are generated via simulated sexual recombination of the two parents as part of the built-in SLiM reproduction function⁵⁷, assuming a recombination rate of 1 cM/Mb. This function takes the two parental genomes and simulates crossing over between the two gametes at the specified recombination rate⁵⁷ and is explained in more detail in Supplementary Note 4.”

Please see above for our new Supplementary Note 4 (above) that now explains in detail how SLiM models recombination.

Lines 760-769: The density-dependent scaling of mortality rates is not a problem, but at first sight it seems rather hypothetical. If there is any ethnographic or ecological information to support it (even for non-human populations), please include citations.

We now reference in this section three papers that provide support for density dependent mortality in Neolithic populations. The first is an archaeological investigation of an early farming site that discusses the impact of increasing population density within a farming community. It finds several costs to population health as a result of the increasing population density. These costs to health include increases in: Disease exposure, interpersonal conflict, and different labor demands surrounding greater needs for the acquisition of various resources. The second is a simulation paper which models the population dynamics during the Holocene in Europe. This study concluded that to produce the expected population dynamics from this time, density dependent social processes such as interpersonal conflict were required. Lastly we cite a paper that found that increased population density was correlated with a reduction in body size and juvenile mortality.

Lines [XX-XX] now read (changes in red):

“To model density-dependent competition—based on the idea that increased density leads to higher health-related mortality costs^{14,90,91}—we implemented a density-dependent scaling of the 'equilibrium' mortality curve. This scaling serves as the

mechanism by which the population growth slows and ultimately stabilizes as it approaches the carrying capacity K .”

Also lines 760-769: I am a little confused, for the following reason. At first I understood that in the agent-based model, logistic reproduction is not applied and the density-dependent scaling of mortality is the mechanism for the population density of farmers to stop growing when it reaches its carrying capacity (if this is so, please mention it so that the reader can understand why you apply the density-dependent scaling of mortality). But according to line 791, the "fertility rate of 0.1 is calculated based on the age-specific mortality curve, such that the population size is maintained when the population is at carrying capacity (i.e., deaths are compensated by births)." Does this imply that the density-dependent scaling of mortality is not needed for the population density of farmers to stop growing when it reaches its carrying capacity? If yes, why do you assume the density-dependent scaling of mortality? Would the results change without assuming it? Please explain carefully.

Yes, the reviewer is correct that the density-dependent scaling of mortality is the key mechanism in our model that causes farmer population growth to slow and stabilize at carrying capacity. We have now made this mechanism more explicit (see our response above). The fertility rate is fixed at 0.1 and does not vary with density. As a result, births occur at a constant rate, while mortality decreases at lower densities—leading to more births than deaths when the population is below carrying capacity. This produces logistic-like population growth. The density-dependent mortality component is therefore essential for limiting population growth as density increases.

We now clarify this in the revised text (lines 830-834):

“We assume a constant fertility rate of 0.1 per mature individual per year (see *Reproduction and within-group mating* section below). The age-specific mortality curve is scaled by population density such that, at carrying capacity, the number of deaths balances the number of births, maintaining a stable population size. At lower densities, mortality decreases, allowing the population to grow toward the carrying capacity.”

Also lines 760-769: If for some reason the density-dependent scaling of mortality is needed, an important question arises: Would essentially the same cline be found by e.g. assuming, instead of density-dependent scaling of mortality rates, simply that there is no net reproduction of farmers when their population density reaches their carrying capacity? I note that it would take longer for the HGs to disappear because in addition to acculturation, you reasonably assume that density-dependent scaling of mortality has to be applied also to them (lines 776-780). But a cline would be generated. Would it be essentially the same as with density-dependent scaling of mortality rates? Knowing this would clarify if there is no need to assume a density-dependent scaling of mortality rates, But even if there is no need, I do not suggest to omit it - I am asking you only to clarify whether it is absolutely necessary or not. And in case it is, whether a simpler approach may also work.

We thank the reviewer for this insightful comment. Indeed, our implementation of density-dependent scaling of mortality is one way to ensure that the population growth of farmers slows as it approaches carrying capacity. As the reviewer suggests, similar qualitative behavior—i.e., the formation of a cline—could be obtained by imposing a hard constraint on net reproduction once the carrying capacity is reached (e.g., by setting net growth to zero beyond a certain density).

However, we chose to model density dependence via mortality rather than capping reproduction directly for two reasons. First, scaling the age-specific mortality curve provides a smooth and biologically interpretable mechanism by which crowding effects (e.g., due to disease or competition for resources) influence survival, in line with ethnographic and demographic evidence (Kondor et al., 2023; Larsen et al., 2019; Walker & Hamilton, 2008). Second, this approach allows individual survival to vary smoothly with population density, rather than imposing a hard cutoff on growth at carrying capacity.

We agree that the qualitative features of the cline would likely persist under either implementation. However, we find that the density-dependent mortality formulation offers a more realistic and mechanistic representation of how density regulates population size, and it is essential in our model to achieve convergence to carrying capacity without explicitly enforcing it.

As already mentioned above (comment on Lines 278-283), in the reaction-diffusion 1-dimensional model you calculate clines when the HGs disappear ($w=0$) but I do not know at what time you calculate clines in the agent based model. Please make sure to explain this.

For our agent based simulation, we selected the cline from the end of the expansion when farming populations had spread across the entire landscape and hunter-gatherers were no longer present. The exact number of years was variable depending on parameters of the simulation but this came generally ~3,500 years into the simulation. In the specific case of Figure 5b the blue line represents 3666 years after the start of the expansion.

This is mentioned already in the figure captions and now we have better clarified that the ancestry value from the agent based model was taken when the HGs disappear by explicitly stating this, rather than only saying it was once all individuals are practicing farming:

Lines 290-293 now read (changes in red):

“We found all learning parameters result in a cline of EF ancestry—estimated once all individuals are practicing farming (i.e., when there are no longer any hunter-gatherer individuals)—that decreases with increasing distance from the farming origin.

However, we are confident that the comparison between the point estimates from ancient individuals and the simulated ancestry cline is valid regardless of the timing when the simulated cline is measured. As mentioned in the response above and shown in Supplementary Figure 11, the ancestry cline remains remarkably consistent throughout the simulation, regardless of the year chosen for ancestry sampling. The main difference across sampled years is the distance that farming has progressed across the landscape, not the shape or slope of the cline itself, thus we chose the cline when farming had progressed across the full landscape.

Line 765: What is "the expected local K within this area"?

The expected local K is the expected number of people inside a given local radius based on the carrying capacity.

Lines 817-823 now read (changes in red):

“The local population pressure is calculated as the ratio of the number of neighbors within a radius of 30 km surrounding each individual, relative to the expected local K (or the expected number of individuals in this area calculated using the carrying capacity). We derive each individual's 'experienced' mortality rate by multiplying its 'equilibrium' mortality rate by the local population pressure, which allows the population to grow until it hits the carrying capacity K. Each cultural group has a unique K— K_F for farmers, and K_{HG} for HGs, which allows farming to support higher population densities.”

Line 789: A mating range of 10 km is surprising because usually the mating distance is much larger, according to ethnographic data of pre-industrial populations. I am not asking to change this, but you could mention if for a range of e.g. 50 km the conclusions do not change.

The reviewer makes a good point in noting that ethnographic data suggests larger spousal mating distances than 10km over a generation. However, in our model, the mating range refers to the *annual* distance an individual can travel in search of a mate, not the generational distance an individual may have traveled before mating. Given that our model assumes a standard deviation of the annual movement distribution of 5 km in each x and y direction (i.e. an average of ~6.3 km traveled per year), a 50 km mating distance would imply individuals can search about ten times farther than they typically move in a year, which we find unrealistic.

We selected a mating range of 10 km to account for the possibility of longer mating excursions—greater than average annual movement—but still within a biologically

realistic range given our movement parameters. In contrast, the ethnographic estimates cited by the reviewer most likely reflect generational dispersal distances (e.g., the distance between spouses' birthplaces) rather than yearly mobility. Because our model assumes each reproductive-age individual may mate each year, the mating range defines the annual spatial window in which they—or a prospective mate—could find each other if they mated within the year.

Nonetheless, we tested a scenario with a 50 km mating range per year and found our model was generally robust to this change. The greater mating distance accelerated the expansion rate, which was then offset by reducing the step size from 5 km to 4.2 km to maintain a comparable expansion speed of ~1km/year. Under these conditions, the model produced an estimated learning rate of 0.000512 [0.000505, 0.00052], which is similar in magnitude to the estimation of 0.000798 [0.000787, 0.000808] from the agent-based model with the simple square landscape and our baseline parameters.

Line 781: It would be nice to mention the "runtime and memory requirements". 4-5 hours and 30 Gb RAM? (README file in github server). If you did not use only a PC but also run simulations in parallel in a cluster, please include also the runtime, memory, etc.

This is a valuable point the reviewer raises and we have changed the text accordingly to represent the text in the README file written up in our code availability GitHub page for this project.

Lines 839-843 now read (changes in red):

“To reduce the **significant** runtime and memory requirements of the agent-based simulation, we downscaled the number of individuals by a factor of five. Thus, we divided all carrying capacities mentioned above, as well as the starting population density, by 5 for all agent-based simulation runs. **This results in an average runtime of 4-5 days on a computing cluster using 20Gb or less of RAM per individual simulation.**”

Of Note: The reviewer comment above says 4-5 hours, but accurate runtime as described in our GitHub that the reviewer noted was actually 4-5 days. Our text above reflects this value, rather than the reviewer's comment.

Line 790: It is surprising (from ethnography) that "the number of offspring" per year and individual can be higher than 1.

Given a fertility rate of 0.1 (the rate parameter of the Poisson distribution), the probability of an individual having two or more offspring in a single pregnancy is approximately 4.9%. This seems broadly consistent with present-day twin birth rates, which are around 3% (Osterman et al., 2023).

Also line 790: To facilitate reproducibility of the results, please include this Poisson distribution in the Supp. Info.

In addition to the mention of the Poisson distribution in the main text Methods, we now explicitly include the distribution that describes the number of offspring per successfully mating mature individual: Offspring number \sim Poisson($\lambda = 0.1$) in Supplemental Note 4.

Lines 808-810: "Thus, for each HG in the simulation at a given year, the transitioning to farming is a random event with probability calculated as the product of the local proportion of farmers and a simulation-wide learning rate f ." To avoid confusion, please add: "(assuming $\gamma = 1$, as justified in the Results section)."

This is a good clarification from the reviewer. We have changed the text accordingly and it now reflects the reviewer's good suggestion.

Lines 868-871 now read (changes in red):

"Thus, for each HG in the simulation at a given year, the transitioning to farming is a random event with probability calculated as the product of the local proportion of farmers and a simulation-wide learning rate f (assuming $\gamma = 1$, as justified in the first section of *Results*)."

Line 906: Please explain "provides broad coverage". Does it mean few negative ancestries?

In this context "broad coverage" refers to the utility of the Balkan_N individuals as a proxy source of early farmer ancestry across a wide temporal and geographical range.

For increased clarity, we have changed the text on lines 966-968 to read (changes in red):

"...Balkan_N serves as a geographically and genetically intermediate farming population that provides broad utility as a proxy source across European Neolithic contexts."

Line 910: "outgroups... (see full lines Supplementary Data File 2)." It seems that it should be Supplementary Data File 1.

This is correct- we thank the reviewer for pointing this out.

The line now reads (changes in red):

"To further stabilize ancestry estimates, we included Turkey_N and a set of deep outgroups in the right populations (see full list in Supplementary Data File 1)."

Line 911: For clarity, a possible reason could be included, e.g. that suggested by reviewer #3: "As a consequence of using Balkan_N, WHICH HAS SOME HG ANCESTRY, some ancient samples... can yield slightly negative WHG ancestry estimates ..."

We appreciate this clarifying point from the reviewer that will make the text more accessible. We have modified our text accordingly.

Lines 972-976 now read (changes in red):

"As a consequence of using Balkan_N estimates, we included Turkey_Nestry, some ancient samples—especially those from southeastern Europe or regions east of the Balkans—can yield slightly negative WHG ancestry estimates, while old samples from northern and northeastern Europe (particularly those older than 7,000 years BP) may show negative EF ancestry values."

Line 966: Just for clarity: "Finally, note that to derive the expected EF ancestry proportion from our simulations AS A FUNCTION OF DISTANCE, we had to bin simulated individuals..."

We thank the reviewer for this clarifying suggestion. We have changed the text to include their suggestion.

Lines 1033-1036 now read (changes in red):

"Finally, note that to derive the expected EF ancestry proportion from our simulations as a function of distance, we had to bin simulated individuals into different distance bins to compute their average ancestry proportion as a function of distance."

Supp. Fig. 12, caption: I suggest to add: "Compare to Fig. 7 in the main paper."

At the suggestion of the reviewer we have modified the figure caption for Supplementary Figure 12 to read (changes in red):

"Supplementary Figure 12: Mean final EF ancestry in the population as a function of the cultural effect given various within-group mating rates

Cultural effect describes the percent contribution of cultural transmission to the front speed in addition to a baseline speed of a fully demic model. This figure shows the cultural effect under various within-group mating parameters (as compared to the same plot but for learning rate parameters in main text Fig. 7), and the mean EF ancestry proportion across the simulated population upon the conclusion of the simulation when farming has become ubiquitous. Simulations were run with no peer-to-peer learning and only parent-to-child vertical cultural transmission via various within-group mating rates (right). For comparison to empirical data, the dashed gray line represents the mean proportion of EF ancestry from our aDNA estimates."

Supp. Fig. 13, caption: I suggest to add: "Compare to Supp. Fig. 11."

This is also a valuable suggestion from the reviewer which will clarify the purpose of this additional plot. This figure caption has been changed to read (changes in red):

"Supplementary Figure 13: Interaction of learning and within-group mating parameters without the assumption that children of farmers only become farmers

These plots are results from simulations where the offspring of a farmer and a HG has a 50/50 chance of becoming either a HG or a farmer. Other simulations in the study assume that any non-within-group matings between HGs and farmers always result in a farming offspring (see Supplementary Figure 11 for comparison). The plots in this present figure do not assume farming offspring. The panels show EF ancestry proportion in the population, over the course of the simulation, plotted over distance from origin (km). Lines show the progression of the expansion over time with different colored lines plotted in 500 year increments. Plot faceted on the x-axis by learning rate and on the y-axis by within-group mating parameters tested."

Supp. Info. File 2: one of the sources is "WHGA". Similarly, in Supp. File 1 the individuals labelled "WHGA" are the HG source population. If this is correct, please introduce the acronym WHGA in line 891, etc. in the main paper (or change WHGA into WHG in Supp. Files 1 and 2).

We have changed the references in the supplemental files to maintain consistency with the manuscript. They are now referred to as EF and WHG rather than EEF and WHGA throughout.

Supp. Info. Files 1 and 2: Similarly, I see that "EEF" refers to the EF population. If this is correct, please introduce the acronym EEF in line 891, etc. in the main paper (or change EEF into EF in Supp. Files 1 and 2).

We have changed the references in the supplemental files to maintain consistency with the manuscript. They are now referred to as EF and WHG rather than EEF and WHGA throughout.

Supp. File 1: I can only find 16 WHGAs (not 18 as stated in line 892) and 20 EEFs (not 21 as stated in line 893). Please correct or clarify, to ensure that all results can be reproduced.

This is indeed correct—three individuals were missing from Supplementary File 1. Thank you for catching this. We've now added their information (one EEF (EF) and two WHGA (WHG)) to the revised file.

Supp. Data File 2: Please include the distance of each individual from the origin (horizontal axis in Fig. 5B) to facilitate the reproducibility of the results. Also, please remember to mention it in the file description (Supp. Info. pdf, p. 25).

In an effort to increase reproducibility, we have added this column to the supplementary file, as suggested by the reviewer.

In spite of the length of this report, this paper gives the solution to an important problem, so it will be very useful for many researchers and should be published after incorporating the suggestions in this report.

We have taken into consideration the valuable feedback from the reviewer and addressed the points they raised. **We truly appreciate the reviewer's time and detailed commentary** that has allowed us to improve the manuscript and better contextualize our results. **Lastly, we thank the reviewer for their repeated support of our work throughout the review, and for their recommendation for publication of this manuscript at *Nature Communications*.**

References Cited in Responses to Reviewers:

- Flegontova, O., Işıldak, U., Yüncü, E., Williams, M. P., Huber, C. D., Kočí, J., Vyazov, L. A., Changmai, P., & Flegontov, P. (2025). Performance of qpAdm-based screens for genetic admixture on graph-shaped histories and stepping stone landscapes. *Genetics*, *230*(1), iyaf047. <https://doi.org/10.1093/genetics/iyaf047>
- Kim, B. Y., Huber, C. D., & Lohmueller, K. E. (2018). Deleterious variation shapes the genomic landscape of introgression. *PLOS Genetics*, *14*(10), e1007741. <https://doi.org/10.1371/journal.pgen.1007741>
- Kondor, D., Bennett, J. S., Gronenborn, D., Antunes, N., Hoyer, D., & Turchin, P. (2023). Explaining population booms and busts in Mid-Holocene Europe. *Scientific Reports*, *13*(1), 9310. <https://doi.org/10.1038/s41598-023-35920-z>
- Larsen, C. S., Knüsel, C. J., Haddow, S. D., Pilloud, M. A., Milella, M., Sadvari, J. W., Pearson, J., Ruff, C. B., Garofalo, E. M., Bocaege, E., Betz, B. J., Dori, I., & Glencross, B. (2019). Bioarchaeology of Neolithic Çatalhöyük reveals fundamental transitions in health,

- mobility, and lifestyle in early farmers. *Proceedings of the National Academy of Sciences*, 116(26), 12615–12623. <https://doi.org/10.1073/pnas.1904345116>
- Li, W., & Freudenberg, J. (2009). Two-parameter characterization of chromosome-scale recombination rate. *Genome Research*, 19(12), 2300–2307. <https://doi.org/10.1101/gr.092676.109>
- Osterman, M. J. K., Hamilton, B. E., Martin, J. A., Driscoll, A. K., & Valenzuela, C. P. (2023). Births: Final Data for 2021. *National Vital Statistics Reports: From the Centers for Disease Control and Prevention, National Center for Health Statistics, National Vital Statistics System*, 72(1), 1–53.
- Walker, R. S., & Hamilton, M. J. (2008). Life-History Consequences of Density Dependence and the Evolution of Human Body Size. *Current Anthropology*, 49(1), 115–122. <https://doi.org/10.1086/524763>
- Williams, M. P., Flegontov, P., Maier, R., & Huber, C. D. (2024). Testing times: Disentangling admixture histories in recent and complex demographies using ancient DNA. *Genetics*, 228(1), iyae110. <https://doi.org/10.1093/genetics/iyae110>

LaPolice T.M., Williams M.P. & Huber C.D.

Point by Point Response to Reviewers (Revision #4)

We thank the reviewers and the editor for their time and their continued feedback. Below, we have responded to the comments about our second revision, and made changes in this fourth revision based on their suggestions.

Reviewer #4

(Remarks to the Authors, Responses from the Authors*, and Relevant Excerpts from the Revised Manuscript): *Line numbers in response text relate to tracked changes version

This second version has properly taken into account the single major point in my previous report. I believe that for this reason, we can now be confident on the validity of the results. Although personally I prefer the new approach based on the 53 markers (because then empirical and simulated ancestries are calculated in the same way), using qpAdm instead to calculate the empirical ancestries yields approximately the same learning rate. So the approach using qpAdm is also sound. Moreover the new version is substantially clearer and more complete than the previous one. So I am very happy to recommend publication in Nature Comm. of this important, very useful and timely paper.

We are pleased that we were able to address the reviewer's main comment satisfactorily and ensure the robustness of our approach. We appreciate their recommendation and feedback as it has improved the manuscript's clarity and robustness. We thank the reviewer again for their continued time and detailed feedback.

The authors might perhaps want to consider the following *optional* points, which I believe would not require much work and would make the paper even clearer and more complete. The line numbers are those from the version with tracked changes.

1. Caption to Fig. 1: "by the ratio C" should be "by the ratio C/ γ ".

This is referring to the ratio $C = f/\gamma$. We have made the caption for Figure 1B clearer; it now reads (changes in red):

“(B) Relationship between the ratio of the learning rate to the bias parameter ($C = f/\gamma$) and the proportion of EF ancestry in the model, averaged across the landscape. For this one-dimensional model, we calculated the EF ancestry proportion at generation 100, shortly after all HGs have been acculturated. This plot illustrates that the final proportion of EF ancestry after the Neolithic expansion is predominantly determined by the ratio $C =$

f/γ , rather than the individual values of f or γ . Data points indicate model outcomes across a range of learning rates (f) and bias parameters (γ).”

2. The logistic terms in the set of 3-population coupled equations in Fig. 1A are apparently not discussed. They contain $1-(u+v+w)/K_F$ for farmers and $1-(u+v+w)/K_{HG}$ for HGs. This is different from other coupled logistic terms between farmers and HGs that have been considered in the literature, e.g. $1-(u+v)/K_F$ for farmers and $1-w/K_{HG}$ for HGs (Aoki, Shida & Shigesada, Theor. Popul. Biol. 1996; Aoki, Human Popul. Genet. Genom. 2024), or $1-(u+v)/K_F-w/K_{HG}$ both for farmers and HGs (Isern & Fort, New J. Phys. 2010). Can the authors include some justification of their logistic terms and bibliographical references about them in their Supplementary Note 1? I believe that a valid justification is simply that competition between farmers and HGs leads to an additional mortality for each of them. It would be very useful to give this justification (or another one) as well as to include references with derivations, justifications and empirical checks. Also, the recent work by Cortell-Nicolau et al., PNAS 2025 should be cited. They consider $1(u+v+c_1*w)/K_F$ for farmers and $1-(c_2*(u+v)+w)/K_{HG}$ for HGs. Similarly, in the mathematica file by the authors $c_1=c_2=c$ and they use $c=1$. Can the authors include some justification of using $c_1=1$ and $c_2=1$? The last paragraph in p. 9 of their response to my suggestions can be an excellent addition and solve this.

We thank the reviewer for the very helpful suggestion. We have added a justification for the form of the logistic terms in Supplementary Note 1, explaining that they reflect direct competition between farmers and hunter-gatherers for shared resources. We now cite relevant work (Aoki et al. 1996; Isern & Fort 2010; Aoki 2024; Cortell-Nicolau et al. 2025) and clarify that our formulation assumes equal contribution to density dependence. We also describe our tests of alternative formulations using a competition coefficient c , including the case $c_1 = c_2 = 1$ used in our code. As noted in our response, these changes have minimal effect on ancestry dynamics unless competition is nearly absent. The requested clarification has been added to the Supplementary Note 1:

Justification of Logistic Terms

In our model, the logistic terms $1 - (u + v + w) / K_F$ (for farmers) and $1 - (u + v + w) / K_{HG}$ (for hunter-gatherers) assume that farmers and hunter-gatherers compete for shared ecological resources that determine the local carrying capacity. This formulation reflects symmetric density dependence, where all individuals contribute equally to local crowding, resulting in reduced reproduction or increased mortality as the total population density approaches the carrying capacity, K_F or K_{HG} .

This approach contrasts with alternative formulations in the literature where partial niche overlap is modeled using differential weighting of group contributions, such as:

- $1 - (u + v) / K_F, 1 - w / K_{HG}$ (Aoki et al., 1996; Aoki, 2024)^{1,2}
- $1 - (u + v) / K_F - w / K_{HG}$ (Isern & Fort, 2010)³

- $1 - (u + v + c_1 * w) / K_F, 1 - (c_2 * (u + v) + w) / K_{HG}$ (Cortell-Nicolau et al., 2025)⁴

To evaluate the sensitivity of our results to this assumption, we tested a generalized form with a competition coefficient c (between 0 and 1) controlling the strength of inter-group competition: $1 - (c * w + u + v) / K_F$ for farmers, and $1 - (w + c * (u + v)) / K_{HG}$ for hunter-gatherers. We found that only when c is very small (i.e., when competition or niche overlap is nearly absent) do the ancestry clines differ meaningfully from the baseline model. Accordingly, and for simplicity, we set $c = 1$ in our default formulation (i.e., full competition), consistent with the parameterization used in our Mathematica code.

3. Line 195: "that the final EF ancestry after the Neolithic expansion..." → "that the final spatially-averaged EF ancestry after the Neolithic expansion..."

We have changed lines 203-206 based on the reviewer's suggestion to increase clarity (changes in red):

"By altering the values of f and γ , we observed that the final **spatially-averaged** EF ancestry after the Neolithic expansion correlates most strongly with the ratio $C = f/\gamma$, rather than the individual parameters f or γ , even when f varies over four orders of magnitude (see Fig. 1B)."

4. Line 198: "falling below 50% when the ratio reaches 0.1 - an outcome that is..." → "falling below 50% when the ratio reaches 0.1 per generation - an outcome that is..."

We have adjusted lines 206-209 at the reviewer's suggestion (changes in red):

"Moreover, an increase in the f/γ ratio strongly correlates with a decrease in EF ancestry, falling below 50% when the ratio reaches 0.1 **per generation**—an outcome that is highly inconsistent with the predominant EF ancestry proportion estimated from aDNA for Neolithic Europe⁴⁷."

5. Many estimates of the learning rate (mean values and ranges) in the main text have no units. All of them should be reported with units (e.g. yr-1 or "per year"). For instance, in line 328: "leaning rates larger than 0.03 result in..." → "leaning rates larger than 0.03 per year result in..."

This is a good point by the reviewer. Previously, we only indicated when the unit was different (i.e., per generation values), but it is good practice to include "per year" for all mentions of learning rate values. We have gone through and added this throughout the manuscript.

6. Line 353: "We further saw a rapid loss of EF ancestry when we decreased the within-group mating rate (i.e., increased the between group rate), such that with

within-group mating rates less than 90%, EF ancestry quickly declined to zero toward the far end of the range (Fig. 4B), analogous to what we observed with learning rates greater than 0.01 (Fig. 4A). "→" We further saw a rapid loss of EF ancestry when we decreased the within-group mating rate (i.e., increased the between group rate), such that with within-group mating rates less than 90%, EF ancestry quickly declined to zero toward the far end of the range (Fig. 4B). This has been previously observed using haplogroup frequencies (Fig. 3 in 51). It is analogous to what we observed with learning rates greater than 0.01 per year (Fig. 4A)."

We have added this reference to previous literature in lines 291-296 at the reviewer's suggestion (changes in red):

"We further saw a rapid loss of EF ancestry when we decreased the within-group mating rate (i.e., increased the between-group rate), such that with within-group mating rates less than 90%, EF ancestry quickly declined to zero toward the far end of the range (Fig. 4B). This has been previously observed using mtDNA haplogroup frequencies⁵¹, and is analogous to what we observed with learning rates greater than 0.01 per year (Fig. 4A)."

7. Line 497: I believe that refs. 70,71 did not show that the Neolithic spread faster along the Mediterranean coast than inland. This was shown by Zilhao, PNAS 2001 and Isern et al., PNAS 2017.

The original 70 and 71 we cited are related to the claim earlier in the same sentence that "prior studies have indicated that the agricultural expansion was not uniform", rather that they show faster spread on the coast directly. However, for clarity we have taken the reviewer's suggestion to cite Isern et al., *PNAS* 2017 as a replacement reference 70. We have elected to keep reference 71 the same as it discusses different expansion axes and expansion heterogeneity between routes (i.e., relates to both claims).

8. Line 556: "This estimate is robust to modeling assumptions and one order of magnitude smaller than previously estimated upper limits..." → "This estimate is robust to modeling assumptions and substantially smaller than previously estimated upper limits..." (to avoid an inconsistency with the Discussion section).

We have updated lines 424-426 per the reviewer's suggestion (changes in red):

"This estimate is robust to modeling assumptions and substantially smaller than previously estimated upper limits of the learning rate based on the speed of the farming expansion alone^{31,33}"

9. Line 581: "For our empirically estimated learning rate of approximately 0.001, the cultural effect is close to zero (0.5%), suggesting..." → "For our empirically estimated learning rate of approximately 0.001 per year, the cultural effect is close to zero (0.5%)."

This is similar to the range estimated using haplogroup data³⁴ (0.7-2.3%), both results suggesting..."

We have updated lines 449-453 to relate back to previous literature as suggested by the reviewer (changes in red):

"For our empirically estimated learning rate of approximately 0.001 per year, the cultural effect is close to zero (0.5%). This is similar to the range estimated using mtDNA haplogroup data³⁴ (0.7-2.3%), both results suggesting that this level of learning does not substantially increase the front speed relative to a pure demic model."

10. Line 640: "estimated that up to 10% of farmers each year were converting HGs, assuming a 25-year generation time³¹" → "estimated that up to 10% of farmers each year were converting a HG, if we apply our value of a 25-year generation time³¹"

We appreciate the reviewer's suggestion. However, we prefer to retain the original phrasing—"assuming a 25-year generation time"—as it more clearly and concisely conveys that the estimate depends on a generational timescale, without overemphasizing our specific choice.

11. Line 739: "However, additional analyses are needed to assess the robustness of our findings to other model parameters, such as generation time and population growth rate, which may also differ between the Mediterranean and continental routes." The underlined text is difficult to defend. In contrast, differences in the dispersal behaviour are much better established: the archaeological data imply that along the western Mediterranean coast the distance moved per generation was between 240 and 427 km, i.e. between 5 and 8 times longer than along the inland route [Fort, AAS 2022]. More importantly, taking into account the error in all other parameter values will inflate the estimated error in the learning rate (as I wrote in my previous report, this was observed for the cross-mating rate in Ref. 56). So I suggest to change this sentence into: "However, additional analyses are needed to assess the robustness of our findings to other model parameters. Indeed, a sensitivity analysis over the ranges of all parameters would be needed to see if our southern and northern estimations for the learning rate overlap or not."

We have changed lines 573-576 based on the reviewer's suggestion (changes in red):

"However, additional analyses are needed to assess the robustness of our findings to other model parameters. In particular, a sensitivity analysis over the ranges of all parameters would be needed to see if our southern and northern estimations for the learning rate overlap or not."

12. Line 796 says that the learning rate f obtained using qpAdm is 0.000823 per year. Why not 0.00798 per year, as in line 456? (the leftmost value in Fig. 6, from Supplementary Fig. 5A). Where does the value 0.000823 come from and which of the models in the horizontal

axis of Fig. 6 has been used to obtain this estimate and that from the 53 markers (0.00103 per year)?

We thank the reviewer for pointing out this inconsistency. The value of 0.000823 per year was obtained using a least-squares fitting approach, which we applied to both datasets (marker-derived ancestry vs. qpAdm-based estimates) to facilitate direct comparison. It is not possible to apply the likelihood approach outlined in Supplementary Note 3 to the marker-derived ancestry, as this does not provide a standard error of the individual ancestry estimates (necessary for the likelihood approach). However, we recognize that this introduced confusion, since the rest of the manuscript focuses on estimates derived using the likelihood approach. To maintain consistency and clarity, we have updated the text on lines 629 to use the 0.00798 estimate. The implications remain the same.

13. The reproduction part of the agent-based simulations could be clearer: The authors write (line 805): "we computed an equilibrium fertility rate of 0.1, for each mature individual over the age of 11. I.e., this fertility rate compensates for yearly deaths, which allows the population size to stay constant over time". But as the wave-of-advance propagates, the population has to grow, so maybe add "in regions where it has reached the carrying capacity". They also write (line 849): "When a suitable mate is found, then the number of offspring is sampled from a Poisson distribution with a fertility rate of 0.1." Not clear enough for any reader to reproduce the results easily. Please include a Supplementary Note in the Supp. Info. with the equation of this Poisson distribution (rather than just "offspring number \sim Poisson($\lambda = 0.1$)" as in Supp. Note 4), a plot of it (I suppose that the number of offspring or children will correspond to the horizontal axis), a plot of the cumulative probability, and an explanation similar to this one (if it is correct): "We generate a number at random between 0 and 1. If it is e.g. 0.56, we see from the plot of the cumulative probability that it corresponds to an offspring of ... children."

We thank the reviewer for their comment– we have made the changes (red) to lines 784-797 accordingly.

“...we computed an equilibrium fertility rate of 0.1 for each mature individual over the age of 11. I.e., this fertility rate compensates for yearly deaths, which allows the population size to stay constant over time in regions where it has reached the carrying capacity.”

To clarify, offspring numbers are sampled directly from a Poisson distribution with rate $\lambda = 0.1$ using the pseudorandom number generator implemented in SLiM. This means that for each mating event, the number of offspring is drawn as a non-negative integer according to the Poisson probability mass function. We have revised (red text) the main text on lines 839-842 to make this clearer:

“When a suitable mate is found, then the number of offspring is sampled from a Poisson distribution with rate parameter $\lambda = 0.1$ using SLiM’s built-in pseudorandom number

generator. This produces a non-negative integer number of offspring per mating event, drawn according to the Poisson probability mass function.”

Given that the Poisson distribution is a well-known and widely used model for discrete stochastic events such as offspring number, and that we now clearly describe its use in both the main text and Supplementary Note 4, we feel that an additional plot of the distribution is not necessary for reproducibility or clarity. We therefore prefer to keep the supplementary material streamlined and have not included a plot.

14. Line 934: "one neutral marker mutation is mutation are initialized every megabase along the 247Mb..."→"one neutral marker mutation is initialized at every megabase along the 247Mb..."

We thank the reviewer for this catch of our typographical error- we have changed it to fix this sentence:

Lines 760-762 now read:

“To determine the ancestry proportion of each simulated individual, one neutral marker mutation is initialized every megabase along the 247Mb simulated chromosome (human chromosome 1) of all farmers.”

15. Line 954: "We generated an age-specific 'equilibrium' mortality rate that corresponds to the age-at death distribution observed in the age-at-death data (Supplementary Table 1; Supplementary Table 2; Supplementary Fig 7)". First, please mention if you apply it both to farmers and HGs. Second, it is not clear [to me] how these tables and figure were computed. Please include the observed age-at-death data and explain carefully the method in a Supplementary Note, so that any reader can easily repeat the calculations and reproduce these tables and figure.

We thank the reviewer for this helpful comment. We now clarify in the main text (red) on lines 778-780 that the same age-specific mortality schedule is applied to both farmers and hunter-gatherers.

“We generated an age-specific 'equilibrium' mortality rate that is applied to both farmers and hunter-gatherers, which corresponds to the age-at-death distribution observed in the age-at-death data^{42,55} (Supplementary Data File 3; Supplementary Data File 4; Supplementary Note 5; Supplementary Fig 7), i.e., in equilibrium, this mortality rate leads to the age-at-death distribution observed in the osteological data^{42,55}.”

To improve clarity and reproducibility, we have added a detailed explanation as a new Supplementary Note describing how the age-specific mortality rates were derived from published Neolithic age-at-death data. Specifically, we use the observed age-at-death distribution as an estimate of the stable age structure under demographic equilibrium.

From this, we compute the conditional probability of death in each age bin given survival to that bin. These probabilities are then used as the mortality rate in the agent-based model.

We have added this explanation to the supplement under Supplementary Note 5:

Supplementary Note 5: Method for Estimating Age-Specific Mortality Rates and the Equilibrium Fertility Rate

Age-Specific Mortality Rates

We used published age-at-death data derived from Neolithic skeletal remains, binned into discrete yearly age categories (see Supplementary Data File 3; Supplementary Data File 4). These data reflect the relative frequency of individuals who died in each age year and can be interpreted as a stable age-at-death distribution under the assumption of demographic equilibrium. To estimate age-specific mortality rates from this distribution, we followed these steps:

Step 1: Normalize the age-at-death counts to get a probability distribution that sums to 1 across all age bins.

Step 2: Estimate the conditional probability of death in each age bin, given survival to that bin. This is calculated as: $q_x = d_x / (d_x + d_{x+1} + d_{x+2} + \dots)$

where:

- q_x is the probability of dying in age bin x , given survival to the start of that bin
- d_x is the number of observed deaths in age bin x
- the denominator is the sum of all deaths in age bin x and older

Step 3: Apply these probabilities as age-specific mortality rates in the agent-based model. These rates are used identically for both farmers and hunter-gatherers.

This approach approximates an "equilibrium" mortality schedule under the assumption of a stable age distribution and no strong age-related preservation bias. While simplified, it provides a plausible mortality pattern consistent with Neolithic skeletal data.

Equilibrium Fertility Rate

To ensure demographic equilibrium in our agent-based model, we derived a fertility rate that balances mortality such that the population remains stable over time (i.e., zero net growth).

Step 1: Using the observed age-at-death distribution from Neolithic skeletal data (see Supplementary Data File 3; Supplementary Data File 4), we estimated the probability of surviving to each age, denoted as s_x . This was calculated as:

s_x = (number of individuals who survived to age x) divided by (total number of individuals)

Here, the number surviving to age x is computed as the sum of all deaths in age x and older. These survival probabilities approximate the age structure of a stable population under the observed mortality pattern.

Step 2: We define the reproductive age range as individuals aged 12 and older, consistent with assumptions in anthropological demography for early farming populations.

Step 3: Under equilibrium conditions, each individual must, on average, produce one surviving offspring to replace themselves. Therefore, the equilibrium fertility rate f_r is calculated as:

$$f_r = 1 / (\text{sum of } s_x \text{ for } x \geq 12)$$

This gives the per-individual per-year fertility rate required to maintain a stable population size, assuming all mature individuals have equal fertility and mortality remains constant over time. In our case, for the mortality rates in Supplementary Data File 3, the sum of survival probabilities for ages 12 and up is approximately 10, leading to:

$$f_r = 1 / 10 = 0.1$$

Step 4: This fertility rate is applied in the simulation by drawing the number of offspring from a Poisson distribution with rate (λ) = 0.1. This is done for each individual aged 12 or older who has a suitable mating partner. This implementation ensures that, on average, births and deaths balance out, maintaining demographic equilibrium over time.

16. Line 957: " Using these 'equilibrium' mortality rates for each age, we computed an equilibrium fertility rate of 0.1, for each mature individual above the age of 11". Please include the detailed calculation in a Suppl. Note. How do you apply this fertility of 0.1 in the Poisson distribution of offspring? (point 13 above).

We thank the reviewer for this helpful comment. We have added a detailed explanation of the fertility rate calculation to the new Supplementary Note 5. Please see our response to comment 15 above. Briefly, we estimated age-specific survival probabilities from the observed age-at-death distribution and calculated the equilibrium fertility rate as

the reciprocal of the sum of survival probabilities for individuals aged 12 and older. This yields a fertility rate of 0.1, which we apply by sampling the number of offspring from a Poisson distribution with rate (λ) = 0.1 for each individual in this age group who has a mating partner. This implementation ensures demographic stability over time.

17. Line 963: morality → mortality.

We thank the reviewer for catching this typographical error. We have fixed this accordingly.

Lines 789-793 now read (changes in red):

“Our primary age-specific mortality rates (Supplementary Data File 3) correspond to the age-at-death distribution observed by Papathanasiou (2005)⁴² but to assess if different mortality curves impact our results, we sought out an alternative mortality curve based on Eshed et al., (2004)⁵⁵ (Supplementary Fig. 7; Supplementary Data File 4).”

18. Line 999: "density-dependent competition takes into account all individuals, independent of cultural identity. As a result, HG populations gradually decline over time as the population pressure from close by farming populations becomes too large for their smaller KHG". I do not understand. From lines 971-981 I thought that density-dependent competition was just an increase of farming populations with densities below K_F , a decrease of farming populations above K_F , and similarly an increase of HG populations below K_{HG} and a decrease of HG populations above K_{HG} . If this is wrong and the numbers of farmers and HGs within a radius of 30 km surrounding each individual are added up, what carrying capacity is used to calculate the population pressure? K_F , K_{HG} , K_F+K_{HG} , or K_F if the individual is a farmer and K_{HG} if he/she is a HG? Moreover, my understanding of the process was that HG populations decline only because of cultural transmission. If for the agent model there is an additional cause of HG decline (density dependent competition), for clarity please mention that this is in contrast with the reaction-diffusion model, where the only cause of HG decline is cultural transmission.

We thank the reviewer for this important clarification request. In the agent-based model, density-dependent competition is implemented symmetrically: the total local population density, summed over both farmers and hunter-gatherers within a 30 km radius, is compared to the individual's group-specific carrying capacity (K_F for farmers, K_{HG} for HGs) to determine survival probability. This means that farmers and HGs both contribute to local crowding, and HGs may decline even in the absence of cultural transmission if population pressure from nearby farmers exceeds what K_{HG} can sustain.

We clarify in the revised text that this is consistent with the reaction–diffusion model, which also incorporates shared density dependence between groups. In both models,

population pressure contributes to the decline of HGs, alongside cultural transmission. We have added the following clarifying sentence (red) to the main text on lines 796-807:

“To model density-dependent competition—based on the idea that increased density leads to higher health-related mortality costs^{14,90,91}—we implemented a density-dependent scaling of the 'equilibrium' mortality curve. In our agent-based model, density-dependent competition is determined by the total local population density, regardless of cultural identity, and this is compared to the carrying capacity specific to each group. This approach is consistent with our reaction diffusion model, in which both farmers and hunter-gatherers contribute equally to population pressure. To this end, all age-dependent mortality rates were scaled down in regions with population density below K, and scaled up in regions with population density above K. The local population pressure is calculated as the ratio of the number of neighbors within a radius of 30 km surrounding each individual, relative to the expected local K (or the expected number of individuals in this area calculated using the carrying capacity).”

19. Line 1083: Please mention that in future work it would be useful to apply dispersal distances and probabilities from ethnographic pre-industrial populations, which are never normal distributions but are known to have an important effect on the spread rate and the cultural effect (e.g., Bancells and Fort, AAS 2024, tables 1-2).

We thank the reviewer for the suggestion. We chose to focus the Methods section on the core features of our model and to keep it streamlined. While we recognize the value of alternative dispersal formulations, we feel that an extended discussion in the Methods would disrupt the flow and focus of the modeling description.

20. Supplementary Note 3: This explanation on the use MLE is very detailed and the authors should be commended for writing it. However: (1) As they explain, it is based on several assumptions. (2) As they also note, MLE gives more weight to more precise estimates but this effect is likely negligible because the error bars in Fig. 5B are rather similar. (3) In any case, it would be intuitively clearer for most readers (who are not versed on MLE) to find the best value of f simply by minimizing the sum of the errors (differences between the datapoints and the blue line in Fig. 5B). This does not mean that the MLE results are not interesting, but in any case both approaches should be compared. I suggest to include a new Supplementary Note (right after Supp. Note 3) with the two last paragraphs in p. 17 of their response to my suggestions and the two figures in the upper p. 18. I suggest to include also Fig. 6 in the main paper and the 11 values and error bars of the learning rate f obtained by minimizing the sum of the errors. An error range for each of these 11 values of f can be obtained by bootstrap resampling (i.e., choosing a value from the error bar of each data point in Fig. 5B). Perhaps the values of f will be very similar but the error bars (with 95% confidence level) will be larger than those from MLE. It is important to know. Some conclusions (such as different values of f in southern and northern Europe) depend on how large the error bars are.

We thank the reviewer for the thoughtful suggestion and for the positive feedback on Supplementary Note 3. However, we have chosen not to implement the proposed comparison with a least-squares approach for the following reasons:

The maximum likelihood framework we use is well-established and widely applied in genomics and statistical modeling. It provides both point estimates and confidence intervals, and is known to be robust to deviations from normality due to the central limit theorem.

As the reviewer notes, our data show relatively homogeneous measurement uncertainty across points. Under standard theory, assuming normally distributed measurement error, the MLE and least-squares approaches yield the same point estimates. This is consistent with our previous comparison of the two methods, which showed near-identical estimates. We therefore see no benefit in presenting both.

Similarly, we expect little difference in confidence intervals. The relatively narrow intervals reflect the strong relationship between ancestry clines and learning rate, as well as the large number of aDNA samples. A permutation- or bootstrap-based approach would not, in our view, offer additional interpretive value.

In fact, our current Fig. 6 demonstrates that modeling assumptions—such as landscape shape and preferred movement directions—have a much greater influence on estimated learning rates than the uncertainty arising from measurement error. We therefore believe that focusing on model structure and its implications (as we do in Fig. 6) provides deeper insight than switching estimation frameworks.

Finally, while we briefly note that learning rates may differ along the Mediterranean versus Continental routes, we also clarify in the Discussion that investigating regional differences is beyond the scope of the current study—further sensitivity analyses would be needed to make strong claims about such patterns. That said, our central conclusion, that the learning rate is on the order of 1/1000 per year, remains robust across methods and scenarios.

Given these points, we hope the reviewer understands our decision not to implement the suggested changes. We believe that Supplementary Note 3 offers a clear, replicable, and statistically sound description of our estimation approach.

21. Also in Supp. Note 3, the error from MLE is based on a variance, so is its confidence level (CL) only 68%? Perhaps this is why the error bars in Fig. 6 are so narrow? Should this error be multiplied by 1.96 to obtain the usual confidence level of 95%? If so, the error bars in Fig. 6 and the text should ideally be given with 95% CL. In any case, the % of CL should be given.

We thank the reviewer for raising this point. As noted in the figure caption, the error bars in Fig. 6 represent 95% confidence intervals. The interval limits are calculated as the point estimate plus (upper limit) or minus (lower limit) two times the standard error (SE), where the SE is the square root of the variance of the maximum likelihood estimate, as described in Supplementary Note 3V. This approximation is standard and corresponds closely to a 95% confidence level.

22. Please report the CL of the error bars from qpAdm in the datapoints in Fig. 5b (also in the main paper, line 1234). I believe that it should be the same CL as that used in Fig. 6.

We previously reported the CL in the figure caption for 5B, however we have now added it to the main text, as well, based on the reviewers suggestion. Lines 331-336 now read (changes in red):

“Consistent with previous aDNA studies of the Neolithic⁴⁷, when plotting the estimated ancestry against direct distance from the farming origin (black points with error bars representing 95% CI; Fig. 5B), a clear trend of decreasing ancestry with increasing distance can be observed (dashed pink line, Fig. 5B; solid black line, Supplementary Fig. 4; slope = $-7.554537e-05$, $r^2 = 0.366$, p-value < 0.001).”

We previously reported the CL in the figure caption for 6, however we have now added it to the main text ahead of the first mention of figure 6, as well, based on the reviewers suggestion. Lines 350-352 now read (changes in red):

“This approach leads to an estimated learning rate and 95% CI of 0.000798 [0.000787, 0.000808] per year for 100% within-group mating (Fig. 6; Supplementary Fig. 5A).”

23. Supplementary Note 4: For non-biologist readers, please explain the meaning of "parental homologous chromosomes". Also, there is "a 1% chance of a crossover event occurring per one million base pairs." My interpretation is that on average, there is a crossover event (i.e., a "change of the parent to be copied") per 100 million base pairs. But since the genomes simulated by the authors have only 494 "1"s or "0"s, how is this implemented in practice? I would have simply expected about half of these 494 values (chosen at random) to be copied from one parent and the rest from the other. If they are not chosen at random, please explain why. Perhaps because it is known experimentally that long strings are copied from each parent? Perhaps every marker belongs to a different megabase, so the first 100 values are copied from one parent, the next 100 from the other parent, the next 100 from the first parent, the next 100 from the second parent, and the final 94 from the first parent? It would help a lot if you could explain this in detail.

We thank the reviewer for this helpful comment. We have clarified the terminology and implementation details based on our response to the reviewer below in Supplementary Note 4 (changes in red) to improve accessibility for non-biologist readers.

By "parental homologous chromosomes," we refer to the two copies of a chromosome that each individual inherits—one from each parent. In SLiM, each individual carries two homologous chromosomes, and during reproduction, one recombinant chromosome is passed to the offspring, formed by copying segments from the two parental homologs with recombination.

Regarding the recombination rate: although the simulated genome is simplified and consists of only 494 binary markers, SLiM treats this sequence as embedded in a continuous chromosome of defined physical length—in our case, 247 megabases. The recombination rate is set to 1% per megabase, meaning that on average, one crossover occurs per 100 megabases per meiosis. Recombination breakpoints are randomly placed along this physical chromosome, and the binary markers are distributed along it with equal spacing. Therefore, recombination in SLiM results in long contiguous blocks of markers inherited from one or the other parent, as observed empirically in real genomes.

This design allows us to simulate realistic patterns of recombination without needing to model the full genomic sequence.

Supplementary Note 4: Description of recombination and genome modeling in SLiM

We use the "crossover breakpoints" recombination model in SLiM¹², which is SLiM's standard method for simulating genetic recombination via gametic crossing over¹³. In this model, recombination events are introduced during simulated meiosis by specifying a recombination rate per base pair. We chose a rate of 1 centimorgan per megabase (1 cM/Mb), which reflects the average recombination rate across the human genome¹⁴. This value corresponds to a 1% chance of a crossover event occurring during meiosis per one million base pairs. Accordingly, this leads to a 1% chance of crossover between our marker mutations that we assume to be one million base pairs apart from each other. As described in the SLiM manual¹³, this recombination rate is used when generating parental gametes to randomly place crossover breakpoints along the chromosome. In SLiM¹², each individual carries two homologous chromosomes, and during reproduction, one recombinant chromosome is passed to the offspring, formed by copying segments from the two parental homologs with recombination.

To generate a gamete, SLiM¹² begins copying from one of the two parental homologous chromosomes (i.e., one of the two copies of a chromosome that each individual inherits—one from each parent). Upon reaching a recombination breakpoint, it switches to copying from the other homolog. Thus, these breakpoints simulate crossover events in meiosis, in which genetic material is exchanged between the parental homologs. The result of this process is a recombined haploid gamete, composed of segments inherited from both parental homologs. SLiM¹² then uses these gametes from each parent to assemble the diploid genome of the offspring. SLiM¹² treats this sequence as embedded in a continuous chromosome of defined physical length—in our case, 247

megabases. Therefore, recombination in SLiM¹² results in long contiguous blocks of markers inherited from one or the other parent, as observed empirically in real genomes.

Lastly, of note, for simplicity our model does not assume male or female individuals, thus an individual can, in theory, mate with any other individual with the offspring number being drawn from a Poisson distribution (offspring number \sim Poisson($\lambda = 0.1$)).

This important paper should be definitely published in Nature Comm., even if the authors refuse to introduce these changes. But in my view, they would make their nice work clearer and sounder.

We appreciate the reviewers recommendation for publication in *Nature Communications* and their detailed feedback on our work. It has improved the paper significantly.

Review on LaPolice et al, Modelling the European Neolithic expansion..., submitted to *Nature Comm.*

This is a nice piece of work, with important new results. In my opinion, it should be published as soon as possible in an interdisciplinary journal such as *Nature Comm.* because it is of interest to a wide audience (geneticists, archaeologists, mathematical modelers, ecologists, etc.). This study is important because it is the first one (as far as I know) that uses genome-wide data to estimate the percentage of early farmers that interbred with hunter-gatherers (HGs) in an agricultural expansion.

In this manuscript the observed European Neolithic ancestry cline is compared to clines obtained by means of simulations. The simulated cline that agrees best with the observed one is used to estimate the percentage of early farmers that interbred with hunter-gatherers (HGs). The most important point is that this comparison and estimation are done for the first time using the whole-genome ancestry, in contrast to previous studies that considered mtDNA and Y-chromosome haplogroup frequency clines. Obviously the whole genome has more information than haplogroups, so it is important to simulate a whole-genome ancestry cline. As far as I know, this was not done before this manuscript, in which the authors have found a solution (namely, to consider a tiny subset of the genome in the simulations). It could be argued that the inferred percentage of farmers that interbred with HGs is essentially the same by using either of both approaches, i.e., whole-genome ancestry on one hand (≈ 0.001 per year, i.e., $\approx 2.5\%$ per generation from this manuscript) and a neutral haplogroup on the other (1%-8% per generation from Refs.^{50,69}, see my minor comment below on lines 524-526 of the manuscript). It could be also argued that this agreement is unsurprising because all neutral parts of the genome should be influenced in the same way by cultural transmission, dispersal and reproduction (i.e., the 3 processes at work in the propagation of the Neolithic wave of advance). But this agreement can only be tested with a whole-genome study such as that reported in this study. And this is indeed necessary because the whole genome has more information, as mentioned above, so we have find out whether the implications are similar or not to those from partial information (haplogroups). For these reasons, I find this paper really important and timely.

The authors have already done an utmost effort in response to 3 other reviewers. I suggest only an important, major change and minor ones. The minor ones are mainly to make the paper clearer for non-biologists and to include proper comparisons to previous results, which are largely omitted in the manuscript (so it is impossible to distinguish what is new from what had been already done by other methods). In spite of the length of this report, I believe that the necessary work will not be huge.

MAJOR POINT

I have only one major concern. In my opinion, the datapoints in Fig. 5B do not measure the same quantity as the simulations (blue line in the same figure). So it is not clear [to me] that both estimates of the "EF ancestry proportion" can be plotted in the same figure. However, this may not be a serious problem after all (if both ancestry estimates measure approximately the same quantity). I will not ask the authors to repeat all of their simulations (which take 4-5 days each according to their README file in the github server). But at least a check of the validity of their results should be made, as explained below. It could be simply added, e.g. as Supp. Note 3 & Supp. Fig. 17.

In the simulations, it seems that all farmers have initially all $247 \cdot 2 = 494$ markers equal to "1", and all HGs have all 494 markers equal to "0" (README file). According to p. 34 the authors estimate the "EF ancestry proportion" from a simulation run by counting the number of "1" markers for each individual and dividing by 494. After grouping farmers in bins, they obtain the blue line in Fig. 5B. It seems to me that this procedure is very different from the qpAdm software, which yields the datapoints in the same figure. Indeed, qpAdm requires reference populations and outgroups (to obtain the datapoints in Fig. 5B), whereas counting the proportion of "1"s (blue line) does not. So it seems clear that the datapoints and blue line in Fig. 5B do not measure exactly the same quantity. The datapoints in Fig. 5B will be different if we choose different populations as sources and/or outgroups, whereas the blue line will be the same. So the blue line that best agrees with the datapoints will also be different if we choose different populations as sources and/or outgroups. Thus the inferred percentage of early farmers that interbred

with HGs will also be different (and this is the main result concerning human behavior). Therefore, in my opinion at least one check is necessary to make sure that the percentage of early farmers inferred by the authors is similar to that inferred by some procedure that makes it possible to estimate exactly the same "ancestry" both for the data (datapoints in Fig. 5B) and the simulations (blue line in Fig. 5B).

If qpAdm could be applied to 494 markers, the solution would be to repeat Fig. 5B by applying qpAdm **both** to 494 positions of the ancient farmers (datapoints) and to the 494 positions of the simulated farmers. This should be done with the same source and outgroup populations in both cases (simulations or blue line in Fig. 5B on one hand, and real individuals or datapoints in Fig. 5B on the other). In this new simulation (just for this check), I believe that you cannot assume that all farmers have initially all $247 \cdot 2 = 494$ markers equal to "1", neither that all HGs have all 494 markers equal to "0". In full rigor, this does not seem to correspond strictly to the real initial conditions, i.e., those implied by the ancient DNA data (21 farmers and 19 HGs, from p. 41). Instead, for each initial individual I suggest to use his/her true, real genome from the ancient DNA data, i.e., a number (1, 2, 3 or 4) corresponding to each nucleotide (T, C, G or A) for the 494 markers that you consider, i.e., one per million of human chromosome 1 (line 737). To ensure reproducibility, please add a new Supp. Data file with the 494 markers (nucleotide position and base) of the 21+19+618 individuals and the qpAdm ancestries obtained from them. If your simulation begins with e.g. 210 farmers, 10 of them should have the same 494 nucleotides as the ancient farmer I2532.AG (Supp. Data File 1), 10 other farmers should have the genome of the ancient farmer I2533.AG, etc. If over the landscape there are initially e.g. 1,900 HGs, 100 of them should have the genome of the HG I4971.AG, etc.

If qpAdm does not work with 494 markers, I suggest the following procedure (but the authors may prefer a different one). Consider 494 nucleotide positions (one per million of human chromosome 1, from line 737) that satisfy the following condition. For each nucleotide position, none of the 21 farmers of the EF source population has the same nucleotide as any of the 18 HGs of the WHG source population (this makes it easy to estimate ancestry, as explained below, and can be justified because markers shared by some farmers and HGs can provide less information on acculturation than markers that are different for farmers and HGs). Each marker will be e.g. 1, 2, 3 or 4 (i.e., base T, C, G or A). To ensure reproducibility, please add a new Supp. Data file with the 494 markers of the 21+19+618 individuals (nucleotide position and base). For each of the 618 farmers, count how many of the 494 positions have the same nucleotide as at least one of the initial 21 farmers (let us call this number f) and how many of the 494 positions have the same nucleotide as at least one of the 18 initial hunter-gatherers (h). Then compute the farmer ancestry proportion of this farmer as $f/(f+h) = f/494$. Repeating the same procedure for the other 617 farmers and plotting these datapoints as a function of distance should lead to an observed cline (similar to that of the datapoints in Fig. 5B). Please include these ancestries for the 618 individuals in the new Supp. Info. File mentioned above. Next run a simulation as described in the last two sentences of the previous paragraph (so that the initial genetic conditions agree with the aDNA data). For each of the final simulated farmers, estimate her/his farmer ancestry proportion as $f/(f+h) = f/494$. Due to cultural transmission, some of the HG nucleotides will appear in the final farmers, and their proportion will increase with distance, so this farmer ancestry $f/(f+h) = f/494$ (proportion of the initial farmer nucleotides) will decrease with increasing distance (new blue line, analogous to that in Fig. 5B). In this way you can repeat Fig. 5B **without** using qpAdm, neither for the simulated **nor** for the 618 real ancient farmers. This new figure, analogous to Fig. 5B, should not necessarily replace Fig. 5B. For example, you can include it as Supp. Fig. 17 (if you do not want to repeat all simulations). From this new figure, you can find out the learning rate. Admittedly, this is a very simple approach from a conceptual point of view. But the main point is that it applies the same methodology to the simulated farmers **and** the real ancient farmers, so that both ancestries can be plotted in a figure and compared. In fact, such a procedure is not so different to what you have done for the simulated farmers. The main points are that in this procedure the initial conditions agree with the aDNA data and that you do not have to use different definitions of ancestry for the simulated and observed clines, so you can plot both of them in the same figure and compare them (because they surely measure exactly the same quantity). Alternatively, you can follow a different procedure as long as it uses the same methodology to compute the ancestry for the simulated farmers **and** the real ancient farmers. It would be also nice to compare a little bit these ancestries for the 618

farmers to the corresponding ones from qpAdm (is the average % difference between them small? Is there a high correlation between them? Are both clines similar?).

Independently of whether you apply the method in the last paragraph of that in the previous one, the new simulated and observed clines (obtained using the same definition of EF ancestry) should be used to find out learning rate, which should be compared to that found by the authors (about 0.001 per year).

In case the authors considered totally impossible to apply a single procedure to find out both the datapoints and the blue curve in Fig. 5B, I would still recommend publication of this paper in *Nature Comm.* (because it would nevertheless be an important contribution as a first step to solve this problem), with the only condition that the authors admit in the main text that such a test would be necessary to make sure that the datapoints and blue line in Fig. 5B measure approximately the same ancestry, with a sufficient degree of approximation so that the estimation of the learning rate is reliable.

MINOR POINTS

Lines 84-87: "Previous estimations have suggested that cultural effect could account for anywhere between 0% and ~40% of the expansion speed of farming³¹, with others showing a maximum cultural effect of 21% to be consistent with the observed front speed³³." For readers to understand these results, I suggest to add: "These differences are mainly due to assumptions on the dispersal behaviour of early farmers and mode of cultural transmission."

Also in line 84: "between 0% and ~40% of the expansion speed of farming³¹...". Here ref. 51 is more appropriate than 31.

Lines 126-127: "We conclude that there must have been near complete within-group mating in farming and hunter-gathering groups." For some readers it would be clearer to add: "(i.e., very few farmer-HG matings)."

Lines 146-147: "prior models addressing the spread ... of the farmer-specific mitochondrial haplogroup K⁵⁰. Importantly, our model extends beyond these by incorporating the genetic ancestry of individuals at a single genetic locus, enabling us to explore...". I believe that Refs. 50, 51, 69 also incorporate the genetic ancestry of individuals at a single locus (indeed, mtDNA behaves as a single locus). A slightly modified write-up would solve this inconsistency, e.g.: "prior models addressing the spread ... of the farmer-specific mitochondrial haplogroup K⁵⁰. Our basic model, similarly to previous ones^{50,51,69}, incorporates the genetic ancestry of individuals at a single genetic locus or marker, enabling us to explore ..."

Figure 1A: In my opinion $(u+v+w)/K_F$ and $(u+v+w)/K_{HG}$ in these 3 equations and the *mathematica* code should be replaced by $(u+v)/K_F + w/K_{HG}$ (see, e.g., Eq. (11) in Ref. ³⁵). However, it is possible that this would not change Fig. 1B much and, in any case, no quantitative comparison to the observed cline is made for this basic (reaction-diffusion one-dimensional) model. So I will let the authors to choose between introducing this change or not.

For the basic (reaction-diffusion one-dimensional) model, the caption to Fig. 1 says that these clines are for [the first value of time such that] $w=0$, i.e., when all HGs have acculturated (by the way, for clarity this could be also mentioned in the main text). But line 715 says that the final time is 150 generations. If this is the first value of time when $w=0$ (approximately, as it depends on position), it should be mentioned in both places. If not, please correct.

Lines 271-273: "We conclude that, although the front speed is informative about the step size parameter of the model, it does not provide fine-scale insight into the degree of cultural transmission." It is important to compare this point to previous work, e.g.: "This conclusion agrees with the wide ranges of the cultural effect calculated from the front speed by using intergenerational dispersal distances and probabilities measured for ethnographic preindustrial farming populations⁵¹ instead of a normal distribution as assumed in the present paper."

Lines 278-283: For the agent-based model, please mention the value(s) of time for the final simulated farmers (i.e., those for which you have computed the blue line in Fig. 5B). In principle, for the comparison between the blue line and the datapoints to be valid, the simulation results should correspond (as closely as possible) to the same time as the radiocarbon dates of the 618 ancient farmers. So I would have perhaps preferred to group the datapoints in Fig. 5B in regions and apply the mean radiocarbon date of the ancient farmers in each region to the simulations (blue line). But there is no need to make any of these changes, because the results could be much the same. However, please check that the simulated values (blue line in Fig. 5B) are essentially the same for the whole range of radiocarbon dates. If this is so, please mention it in the main text. Otherwise, please try to find a solution (that proposed 3-4 lines above or another one).

For color-blinded people, it would help a lot if you could please use the yellow and orange colors for the two upper (or two lower) curves in Figs. 4a and specially 4b. Also, Fig. 4b would be much clearer if it were possible to invert the order of the legend colors and values, so that 1 corresponds to the upper line and 0.7 to the lower line.

Lines 310-312: It seems to me that this has been already discussed in line 288. Please remove one of both, or cite the relevant figure in line 312 (Fig. 4a again?).

Line 317: "we conclude that cultural transmission must have only played a limited role in the farming expansion." This should be compared to other studies, e.g.: "This conclusion is also supported qualitatively by the major genetic [Bramanti et al. 2009] and genomic [Skoglund et al. 2012] turnover in Europe at the time of the Neolithic transition, as well as quantitatively by the low intensities of cultural transmission implied by the shapes of mitochondrial and Y-chromosome clines in Neolithic Europe^{50,51,69}."

Figure 5B: In my printed version, I can barely see the 95% CI bars around the datapoints, because the bars are very thin. Can this be improved?

Lines 369-375 and Methods, lines 948-955: Both paragraphs are impossible to understand for me. Please re-write and expand (possibly in a Supp. Info. note). How is the likelihood of each sample (lines 17) computed from the observed and expected EF proportions (line 951)? Why should it be multiplied across sites (line 373), i.e., why should the log-likelihood be summed up across sites (line 952)? Where can we find the corresponding mathematical derivations? How are the standard error rates of EF from qpAdm taken into account (line 372)? Why is the standard error (line 375) computed as the square root of the negative inverse of the second derivative in Supp. Fig. 5 (line 955)? Is this method really necessary? Please explain why you do not use a simpler method to estimate the value and range of the learning rate, e.g. just to minimize the sum of the magnitudes of the differences between the observed and simulated ancestries over samples (the error of the learning rate could be also estimated, e.g., by repeating estimations with values taken from the qpAdm error range, i.e., from the bars in Fig. 5B). Would the results be so different to justify the use of a method that is unknown to many readers of an interdisciplinary journal? If you are not sure, please write that such an alternative, simpler method could also work. Otherwise, many readers (including me) can be confused.

Lines 376-378: I believe that the words in UNDERLINED CAPITAL LETTERS should be added: "... to an estimated learning rate of 0.000798 [0.000787, 0.000808] FOR 100% WITHIN-GROUP MATING (Fig. 6; Supplementary Fig. 5a). Using the same approach and the simulated clines in Fig. 4b, we also derived an estimated within-group mating parameter of 0.9835 [0.9833, 0.9837] FOR ZERO LEARNING RATE (Supplementary Fig. 5b)."

Lines 457-461: "In sum, despite very different landscape and expansion models, we consistently estimate a yearly learning rate of ~0.1% (Fig. 6). Given this low rate, it suggests that at the tip of the wave-front, only 1 in 1,000 farmers converted 1 HG each year (see Methods). This estimate is robust to modeling assumptions and two orders of magnitude smaller than previously estimated upper limits of the learning rate based on the speed of the farming

expansion alone^{31,33}." Your result should be also related to other previous estimations of the learning rate, e.g.: "Our learning rate of $\approx 0.1\%$ per year or $\approx 2.5\%$ per generation of 25 years is consistent also with the range 1%-8% per generation for the learning (and/or interbreeding) rate previously estimated by comparing clines of mitochondrial and Y-chromosome haplogroups to spatial simulations with a generation time of 32 years^{50,69}."

Lines 503-505: "geographic ancestry patterns remain largely unchanged during the expansion—while still resulting in a process that is predominantly demic in nature (cultural effect < 50%)." A more detailed explanation on this result and its relationship to previous ones is in order, e.g.: "The reason is that for low values of the learning (and/or interbreeding) rate f , the spread rate varies very slowly with f so the spread is overwhelmingly demic^{31,51} but, in sharp contrast, the genetic ancestry cline varies very fastly with f ^{50,69} so there is not a genetic ancestry turnover (except near the origin of the spread) e.g. for $f \approx 0.1$ per generation⁵⁰. We stress that such a lack of genetic turnover for a mainly demic process (cultural effect < 50%) does not happen for the spread of the Neolithic in Europe (in this case, Fig. 5b implies a genetic turnover with $f \approx 0.001$ per year, i.e., $f \approx 0.025$ per generation." In my opinion, the last sentence is important because otherwise many readers may understand that this point refers to the spread of the Neolithic in Europe.

Lines 524-526: "This rate (ABOUT 0.1% OF FARMERS CONVERTING A HG PER YEAR) is significantly more precise than inferences from earlier studies based solely on the speed of the expansion, which estimated that up to 10% of farmers each year were converting HGs, assuming a 25-year generation time³¹." This is another statement that should be related to previous work by adding a sentence, e.g.: "Our result of $\approx 0.1\%$ per year or $\approx 2.5\%$ per generation is consistent also with the range 1%-8% previously estimated from Neolithic haplogroup clines by taking into account the uncertainties in the parameter values^{50,69}."

Lines 529-532: "We estimate that the cultural effect is negligibly small (0.5%), whereas prior studies have reported estimates up to 40%. However, these prior estimates fail to consider the distinct ancestry patterns that would result from such high levels of cultural transmission." This text should be replaced to provide a full, accurate comparison to previous work, e.g.: "We estimate that the cultural effect is negligibly small (0.5%). The first published estimations of this effect were based on analyzing the front speed and found the range 0%-40%^{31,51}. Those estimates were later refined by analyzing haplogroup ancestry patterns, leading to the range 0.7%-2.3%^{50,51}. That range is reasonably similar to our result of 0.5% because we have made use of whole-genome ancestries and assumed a normal distribution for the dispersal of individuals." (NOTE: you cite the range 0.7%-2.3% correctly in line 104 but surprisingly not here)

Lines 532-537: "Moreover, previous approaches are often underpinned by assumptions about the movement of people and how culture was transmitted. For example, they utilize the distance between spousal birthplaces to calculate dispersal ranges^{31,32,72}, or studies of modern missionary trips to determine how readily HGs convert to farming³¹, assuming that these quantities are also applicable to prehistoric populations. Our approach does not rely on such assumptions, instead fitting parameters to empirical observations of front speed and ancestry." Again, please replace this text to provide an accurate, fair comparison to previous work by recognizing that your model also makes assumptions and taking into account that missionary trips were used only in the earliest estimation³¹ but later disregarded⁵¹. In other words, in my opinion replacing this text by something similar to the following one would be much better: "We caution that our model and previous ones are based on some assumptions. For example, unfortunately there are no data on individual dispersal distances in prehistory, but such data are absolutely necessary in any spatial model of Neolithic spread. Previous authors^{29,31,72} used parent-child birthplace and spousal birthplace distances measured for present preindustrial populations. Those distance distributions do not resemble a normal distribution, as assumed by us, but comparing both approaches is of interest and justified by the lack of prehistoric dispersal distances. Another source of uncertainty in the earliest models³¹ was the learning or interbreeding rate, but recently the use of genetics^{50,69} and genomics (present paper) has made it possible to estimate this parameter directly from prehistoric data. Similarly, it has been proposed that prehistoric dispersal distances could be measured in the future by identifying parent-child pairs buried in different places^{32,51}."

Line 544: " Overall, our modeling implies that mating between cultural groups must have been remarkably rare (<3% between-group mating; Supplementary Fig. 5b)". Comparison to previous results is needed, e.g., ", in agreement with previous work^{50,69}".

Lines 583-587: "Two factors may account for this discrepancy. First, our analysis utilized a larger sample size and the finer resolution of genome-wide ancestry estimates, which likely allowed us to detect subtle differences in learning rates that mitochondrial data may not capture. Second, because mitochondrial haplotypes are maternally inherited, they reflect only female ancestry patterns, potentially missing sex-specific cultural interactions and movement dynamics." The second factor seems clearly wrong, because Ref.⁶⁹ shows that a Y-chromosome cline (Fig. 5) is consistent with the same value as the mitochondrial clines (Fig. 3). Moreover, in my opinion the claim of a discrepancy is unjustified, because in your Fig. 6 you find a learning rate of about $0.00125/\text{yr} \cdot 25\text{yr}/\text{gen} = 0.031/\text{gen} = 3.1\%$ per generation for the South (Mediterranean individuals) and about $0.0009/\text{yr} \cdot 25\text{yr}/\text{gen} = 0.0225/\text{gen} = 2.3\%$ per generation for the North (continental individuals) without performing a sensitivity analysis of the parameter values (growth rate, generation time, carrying capacities...) over their complete ethnographic ranges, but both results are within the range 1%-8% found independently for the Mediterranean and the continental routes in the sensitivity analysis over those parameter values reported in the last 2 paragraphs and Secs. S5-S6 of Ref.⁶⁹. So I see no discrepancy and my suggestion would be to change the text above by something that properly summarizes these results, e.g.: "However, there is not discrepancy if we take into account that our Fig. 6 implies a learning rate of about $0.00125/\text{yr} = 3.1\%$ per generation of 25 years for the South (Mediterranean individuals) and about $0.0009/\text{yr} = 2.3\%$ per generation for the North (continental individuals), and both values are within the range found previously from mitochondrial and Y-chromosome haplogroup clines, namely 1%-8% (as estimated by sensitivity analyses of the parameter values independently for the Mediterranean and continental clines)⁶⁹."

Lines 609-611: To increase reproducibility of the results, the final time(s) for each simulation (early, middle and late Neolithic) should be given.

Lines 618-621: I suggest to add the text in UNDERLINED CAPITAL LETTERS in order to compare properly to previous work: "Importantly, even the higher estimates from later periods (0.0006-0.0014 PER YEAR OR 1.5%-3.5% PER GENERATION OF 25 YEARS, FROM FIG. 6) remain two orders of magnitude lower than previous upper estimates based solely on front speed³¹ AND WITHIN THE RANGE 1%-8% PER GENERATION IMPLIED BY HAPLOGROUP CLINES⁶⁹, supporting our main conclusion that cultural transmission played a minimal role in the spread of farming during the Neolithic."

Lines 648-650: "Our findings challenge this binary framework, introducing a third, intermediate scenario: a predominantly demic expansion that occurs without significant ancestry turnover." This should be related to previous results, e.g.: "This third scenario is also seen in previous simulations of haplogroup clines, where some learning rates (e.g., about 10% per generation) lead to waves of advance that are mainly cultural genetically (no major genetic replacement except near the origin of the spread)⁵⁰ but mainly demic archaeologically (no significant increase in the spread rate relative to the purely demic case)³¹."

Line 680: "—we significantly improve on previous estimates that suggest up to 21-40% cultural effect^{31,33}." Again, please replace this sentence by another one that describes all previous work faithfully, for example: "—we estimate a cultural effect of about 0.5%, which improves previous estimates that did not consider the whole genome but haplogroup frequencies and led to the range 0.7%-2.3% for the cultural effect^{50,51}."

Line 703: "The cultural transmission model was originally developed by Fort (2012)³¹, but without accounting for the ancestry of individuals." This is incomplete and misleading, so I suggest e.g.: "The cultural transmission model

was originally developed by Fort (2012)³¹, and accounting for the ancestry of individuals at a single genetic marker by Isern et al. (2017)⁵⁰."

The authors can decide if an additional sentence should be perhaps included to further clarify the previous point, e.g.: "In these models horizontal and vertical transmission are mathematically equivalent, with the only difference that the intensity of cultural transmission (C or f in the present paper) is the number of HGs converted into farmers per early farmer and generation (in horizontal transmission)²¹ or the proportion of early farmers who interbreed with a HG (in vertical transmission)^{xx}." Here Ref. XX stands for *Phys. Rev. E* 83, 056124 (2011), Eqs. (46) and (54).

Lines 729-731: "In each subsequent year, the ancestry proportion of new offspring is determined via sexual recombination of the individual's parental genomes". How is this exactly applied? Please explain it clearly and precisely for non-biologists (here or in the Supp. Info.), so that they do not have to look it up in Ref. 56 or other sources. Also, I suppose that rather than "the ancestry proportion of new offspring" it should be "the simulated genome of new offspring". How is it determined from those of the parents? E.g., for each simulated genome position do you choose at random the corresponding one from one of both parents?

Also, please state explicitly if the software distinguished between men and women or any two simulated individuals can mate.

Line 736: I believe that the words in UNDERLINED CAPITAL LETTERS should be added for clarity: "~~To determine the ancestry proportion~~ SIMULATED GENOME of each individual, ONE neutral marker mutations ~~are~~ IS initialized every megabase along the 247Mb simulated chromosome (human chromosome 1) of all farmers. THEREFORE, SINCE WE CONSIDER DIPLOID INDIVIDUALS, EACH ONE HAS A SIMULATED GENOME OF 247*2=494 CHARACTERS. ALL INITIAL FARMERS HAVE ALL 494 CHARACTERS EQUAL TO "1". These markers "tag" EF ancestry along the chromosome of each individual. ALL HUNTER-GATHERERS HAVE ALL CHARACTERS EQUAL TO "0". Accordingly, FOR FARMERS AFTER THE INITIAL TIME the proportion of EF ancestry is calculated by counting the number of "1" markers and dividing by the total number of marker locations (I.E., 494)."

Lines 740-741: "The genomes of new individuals are generated via simulated 740 sexual recombination of the two parents as part of the built-in SLiM reproduction function⁵⁶." Same comment as above for lines 729-731: How is this exactly applied? Please explain it clearly and precisely for non-biologists (here or in the Supp. Info.), so that they do not have to look it up in Ref. 56 or other sources.

Line 742: "assuming a recombination rate of 1 cM/Mb". Please explain for non-biologists so that they do not have to look up for this: What does "recombination rate" mean? How is this rate applied? Where does this value come from? Is it well-established? Does its uncertainty or error affect the results?

Lines 760-769: The density-dependent scaling of mortality rates is not a problem, but at first sight it seems rather hypothetical. If there is any ethnographic or ecological information to support it (even for non-human populations), please include citations.

Also lines 760-769: I am little confused, for the following reason. At first I understood that in the agent-based model, logistic reproduction is not applied and the density-dependent scaling of mortality is the mechanism for the population density of farmers to stop growing when it reaches its carrying capacity (if this is so, please mention it so that the reader can understand why you apply the density-dependent scaling of mortality). But according to line 791, the "fertility rate of 0.1 is calculated based on the age-specific mortality curve, such that the population size is maintained when the population is at carrying capacity (i.e., deaths are compensated by births)." Does this imply that the density-dependent scaling of mortality is not needed for the population density of farmers to stop growing when it reaches its carrying capacity? If yes, why do you assume the density-dependent scaling of mortality? Would the results change without assuming it? Please explain carefully.

Also lines 760-769: If for some reason the density-dependent scaling of mortality is needed, an important question arises: Would essentially the same cline be found by e.g. assuming, instead of density-dependent scaling of mortality rates, simply that there is not net reproduction of farmers when their population density reaches their carrying capacity? I note that it would take longer for the HGs to disappear because in addition to acculturation, you reasonably assume that density-dependent scaling of mortality has to be applied also to them (lines 776-780). But a cline would be generated. Would it be essentially the same as with density-dependent scaling of mortality rates? Knowing this would clarify if there is no need to assume a density-dependent scaling of mortality rates, But even if there is no need, I do not suggest to omit it - I am asking you only to clarify whether it is absolutely necessary or not. And in case it is, whether a simpler approach may also work.

As already mentioned above (comment on Lines 278-283), in the reaction-diffusion 1 dimensional model you calculate clines when the HGs disappear ($w=0$) but I do not know at what time you calculate clines in the agent-based model. Please make sure to explain this.

Line 765: What is "the expected local K within this area"?

Line 789: A mating range of 10 km is surprising because usually the mating distance is much larger, according to ethnographic data of pre-industrial populations. I am not asking to change this, but you could mention if for a range of e.g. 50 km the conclusions do not change.

Line 781: It would be nice to mention the "runtime and memory requirements". 4-5 hours and 30 Gb RAM? (README file in github server). If you did not use only a PC but also run simulations in parallel in a cluster, please include also the runtime, memory, etc.

Line 790: It is surprising (from ethnography) that "the number of offspring" per year and individual can be higher than 1.

Also line 790: To facilitate reproducibility of the results, please include this Poisson distribution in the Supp. Info.

Lines 808-810: "Thus, for each HG in the simulation at a given year, the transitioning to farming is a random event with probability calculated as the product of the local proportion of farmers and a simulation-wide learning rate f ." To avoid confusion, please add: "(assuming $\gamma = 1$, as justified in the Results section)."

Line 906: Please explain "provides broad coverage". Does it mean few negative ancestries?

Line 910: "outgroups... (see full lines Supplementary Data File 2)." It seems that it should be Supplementary Data File 1.

Line 911: For clarity, a possible reason could be included, e.g. that suggested by reviewer #3: "As a consequence of using Balkan_N, WHICH HAS SOME HG ANCESTRY, some ancient samples... can yield slightly negative WHG ancestry estimates ..."

Line 966: Just for clarity: "Finally, note that to derive the expected EF ancestry proportion from our simulations AS A FUNCTION OF DISTANCE, we had to bin simulated individuals..."

Supp. Fig. 12, caption: I suggest to add: "Compare to Fig. 7 in the main paper."

Supp. Fig. 13, caption: I suggest to add: "Compare to Supp. Fig. 11."

Supp. Info. File 2: one of the sources is "WHGA". Similarly, in Supp. File 1 the individuals labelled "WHGA" are the HG source population. If this is correct, please introduce the acronym WHGA in line 891, etc. in the main paper (or change WHGA into WHG in Supp. Files 1 and 2).

Supp. Info. Files 1 and 2: Similarly, I see that "EEF" refers to the EF population. If this is correct, please introduce the acronym EEF in line 891, etc. in the main paper (or change EEF into EF in Supp. Files 1 and 2).

Supp. File 1: I can only find 16 WHGAs (not 18 as stated in line 892) and 20 EEFs (not 21 as stated in line 893). Please correct or clarify, to ensure that all results can be reproduced.

Supp. Data File 2: Please include the distance of each individual from the origin (horizontal axis in Fig. 5B) to facilitate the reproducibility of the results. Also, please remember to mention it in the file description (Supp. Info. pdf, p. 25).

In spite of the length of this report, **this paper gives the solution to an important problem, so it will be very useful for many researchers and should be published** after incorporating the suggestions in this report.

Review on LaPolice et al, Modelling the European Neolithic expansion..., submitted to *Nature Comm.*

This second version has properly taken into account the single major point in my previous report. I believe that for this reason, we can now be confident on the validity of the results. Although personally I prefer the new approach based on the 53 markers (because then empirical and simulated ancestries are calculated in the same way), using *qpAdm* instead to calculate the empirical ancestries yields approximately the same learning rate. So the approach using *qpAdm* is also sound. Moreover the new version is substantially clearer and more complete than the previous one. So **I am very happy to recommend publication in *Nature Comm.* of this important, very useful and timely paper.**

The authors might perhaps want to consider the following **optional** points, which I believe would not require much work and would make the paper even clearer and more complete. The line numbers are those from the version with tracked changes.

1. Caption to Fig. 1: "by the ratio C" should be "by the ratio C/γ ".
2. The **logistic terms** in the set of 3-population coupled equations in Fig. 1A are apparently not discussed. They contain $1-(u+v+w)/KF$ for farmers and $1-(u+v+w)/KHG$ for HGs. This is different from other coupled **logistic** terms between farmers and HGs that have been considered in the literature, e.g. $1-(u+v)/KF$ for farmers and $1-w/KHG$ for HGs (Aoki, Shida & Shigesada, *Theor. Popul. Biol.* 1996; Aoki, *Human Popul. Genet. Genom.* 2024), or $1-(u+v)/KF-w/KHG$ both for farmers and HGs (Isern & Fort, *New J. Phys.* 2010). Can the authors include some justification of their **logistic terms** and bibliographical references about them in their Supplementary Note 1? I believe that a valid justification is simply that competition between farmers and HGs leads to an additional mortality for each of them. It would be very useful to give this justification (or another one) as well as to include references with derivations, justifications and empirical checks. Also, the recent work by Cortell-Nicolau et al., *PNAS* 2025 should be cited. They consider $1-(u+v+c1*w)/KF$ for farmers and $1-(c2*(u+v)+w)/KHG$ for HGs. Similarly, in the mathematica file by the authors $c1=c2=c$ and they use $c=1$. Can the authors include some justification of using $c1=1$ and $c2=1$? The last paragraph in p. 9 of their response to my suggestions can be an excellent addition and solve this.
3. Line 195: "that the final EF ancestry after the Neolithic expansion..." → " that the final **spatially-averaged** EF ancestry after the Neolithic expansion..."
4. Line 198: "falling below 50% when the ratio reaches 0.1 - an outcome that is..." → "falling below 50% when the ratio reaches 0.1 **per generation** - an outcome that is..."
5. Many estimates of the learning rate (mean values and ranges) in the main text have no units. All of them should be reported with units (e.g. yr^{-1} or "per year"). For instance, in line 328: "leaning rates larger than 0.03 result in..." → " leaning rates larger than 0.03 **per year** result in..."
6. Line 353: "We further saw a rapid loss of EF ancestry when we decreased the within-group mating rate (i.e., increased the between group rate), such that with within-group mating rates less than 90%, EF ancestry quickly declined to zero toward the far end of the range (Fig. 4B), analogous to what we observed with learning rates greater than 0.01 (Fig. 4A)." → "We further saw a rapid loss of EF ancestry when we decreased the within-group mating rate (i.e., increased the between group rate), such that with within-group mating rates less than 90%, EF ancestry quickly declined to zero toward the far end of the range (Fig. 4B). **This has been previously observed using haplogroup frequencies (Fig. 3 in ⁵¹)**. It is analogous to what we observed with learning rates greater than 0.01 **per year** (Fig. 4A)."
7. Line 497: I believe that refs. ^{70,71} did not show that the Neolithic spread faster along the Mediterranean coast than inland. This was shown by Zilhao, *PNAS* 2001 and Isern et al., *PNAS* 2017.

8. Line 556: "This estimate is robust to modeling assumptions and one order of magnitude smaller than previously estimated upper limits..." → "This estimate is robust to modeling assumptions and substantially smaller than previously estimated upper limits..." (to avoid an inconsistency with the Discussion section).

9. Line 581: "For our empirically estimated learning rate of approximately 0.001, the cultural effect is close to zero (0.5%), suggesting..." → "For our empirically estimated learning rate of approximately 0.001 per year, the cultural effect is close to zero (0.5%). This is similar to the range estimated using haplogroup data³⁴ (0.7-2.3%), both results suggesting..."

10. Line 640: "estimated that up to 10% of farmers each year were converting HGs, assuming a 25-year generation time³¹" → "estimated that up to 10% of farmers each year were converting a HG, if we apply our value of a 25-year generation time³¹"

11. Line 739: "However, additional analyses are needed to assess the robustness of our findings to other model parameters, such as generation time and population growth rate, which may also differ between the Mediterranean and continental routes." The underlined text is difficult to defend. In contrast, differences in the dispersal behaviour are much better established: the archaeological data imply that along the western Mediterranean coast the distance moved per generation was between 240 and 427 km, i.e. between 5 and 8 times longer than along the inland route [Fort, AAS 2022]. More importantly, taking into account the error in all other parameter values will inflate the estimated error in the learning rate (as I wrote in my previous report, this was observed for the cross-mating rate in Ref. ⁵⁶). So I suggest to change this sentence into: "However, additional analyses are needed to assess the robustness of our findings to other model parameters. Indeed, a sensitivity analysis over the ranges of all parameters would be needed to see if our southern and northern estimations for the learning rate overlap or not."

12. Line 796 says that the learning rate f obtained using qpAdm is 0.000823 per year. Why not 0.00798 per year, as in line 456? (the leftmost value in Fig. 6, from Supplementary Fig. 5A). Where does the value 0.000823 come from and which of the models in the horizontal axis of Fig. 6 has been used to obtain this estimate and that from the 53 markers (0.00103 per year)?

13. The reproduction part of the agent-based simulations could be clearer:

The authors write (line 805): "we computed an equilibrium fertility rate of 0.1, for each mature individual over the age of 11. I.e., this fertility rate compensates for yearly deaths, which allows the population size to stay constant over time". But as the wave-of-advance propagates, the population has to grow, so maybe add "in regions where it has reached the carrying capacity".

They also write (line 849): "When a suitable mate is found, then the number of offspring is sampled from a Poisson distribution with a fertility rate of 0.1." Not clear enough for any reader to reproduce the results easily. Please include a Supplementary Note in the Supp. Info. with the equation of this Poisson distribution (rather than just "offspring number \sim Poisson($\lambda = 0.1$)" as in Supp. Note 4), a plot of it (I suppose that the number of offspring or children will correspond to the horizontal axis), a plot of the cumulative probability, and an explanation similar to this one (if it is correct): "We generate a number at random between 0 and 1. If it is e.g. 0.56, we see from the plot of the cumulative probability that it corresponds to an offspring of ... children."

14. Line 934: "one neutral marker mutation is mutation are initialized every megabase along the 247Mb..." → "one neutral marker mutation is initialized at every megabase along the 247Mb..."

15. Line 954: "We generated an age-specific 'equilibrium' mortality rate that corresponds to the age-at-death distribution observed in the age-at-death data (Supplementary Table 1; Supplementary Table 2; Supplementary Fig 7)". First, please mention if you apply it both to farmers and HGs. Second, it is not clear [to me] how these tables and figure were computed. Please include the observed age-at-death data and

explain carefully the method in a Supplementary Note, so that any reader can easily repeat the calculations and reproduce these tables and figure.

16. Line 957: " Using these 'equilibrium' mortality rates for each age, we computed an equilibrium fertility rate of 0.1, for each mature individual above the age of 11". Please include the detailed calculation in a Suppl. Note. How do you apply this fertility of 0.1 in the Poisson distribution of offspring? (point 13 above).

17. Line 963: morality → mortality.

18. Line 999: "density-dependent competition takes into account all individuals, independent of cultural identity. As a result, HG populations gradually decline over time as the population pressure from close by farming populations becomes too large for their smaller *KHG*". I do not understand. From lines 971-981 I thought that density-dependent competition was just an increase of farming populations with densities below *KF*, a decrease of farming populations above *KF*, and similarly an increase of HG populations below *KHG* and a decrease of HG populations above *KHG*. If this is wrong and the numbers of farmers and HGs within a radius of 30 km surrounding each individual are added up, what carrying capacity is used to calculate the population pressure? *KF*, *KHG*, *KF+KHG*, or *KF* if the individual is a farmer and *KHG* if he/she is a HG? Moreover, my understanding of the process was that HG populations decline only because of cultural transmission. If for the agent model there is an additional cause of HG decline (density-dependent competition), for clarity please mention that this is in contrast with the reaction-diffusion model, where the only cause of HG decline is cultural transmission.

19. Line 1083: Please mention that in future work it would be useful to apply dispersal distances and probabilities from ethnographic pre-industrial populations, which are never normal distributions but are known to have an important effect on the spread rate and the cultural effect (e.g., Bancell and Fort, AAS 2024, tables 1-2).

10. Supplementary Note 3: This explanation on the use MLE is very detailed and the authors should be commended for writing it. However: (1) As they explain, it is based on several assumptions. (2) As they also note, MLE gives more weight to more precise estimates but this effect is likely negligible because the error bars in Fig. 5B are rather similar. (3) In any case, it would be intuitively clearer for most readers (who are not versed on MLE) to find the best value of f simply by minimizing the sum of the errors (differences between the datapoints and the blue line in Fig. 5B). This does not mean that the MLE results are not interesting, but in any case both approaches should be compared. I suggest to include a new Supplementary Note (right after Supp. Note 3) with the two last paragraphs in p. 17 of their response to my suggestions and the two figures in the upper p. 18. I suggest to include also Fig. 6 in the main paper and the 11 values and error bars of the learning rate f obtained by minimizing the sum of the errors. An error range for each of these 11 values of f can be obtained by bootstrap resampling (i.e., choosing a value from the error bar of each data point in Fig. 5B). Perhaps the values of f will be very similar but the error bars (with 95% confidence level) will be larger than those from MLE. It is important to know. Some conclusions (such as different values of f in southern and northern Europe) depend on how large the error bars are.

21. Also in Supp. Note 3, the error from MLE is based on a variance, so is its confidence level (CL) only 68%? Perhaps this is why the error bars in Fig. 6 are so narrow? Should this error be multiplied by 1.96 to obtain the usual confidence level of 95%? If so, the error bars in Fig. 6 and the text should ideally be given with 95% CL. In any case, the % of CL should be given.

22. Please report the CL of the error bars from *qpAdm* in the datapoints in Fig. 5b (also in the main paper, line 1234). I believe that it should be the same CL as that used in Fig. 6.

23. Supplementary Note 4: For non-biologist readers, please explain the meaning of "parental homologous chromosomes". Also, there is "a 1% chance of a crossover event occurring per one million base pairs." My interpretation is that on average, there is a crossover event (i.e., a "change of the parent to be copied") per

100 million base pairs. But since the genomes simulated by the authors have only 494 "1"s or "0"s, how is this implemented in practice? I would have simply expected about half of these 494 values (chosen at random) to be copied from one parent and the rest from the other. If they are not chosen at random, please explain why. Perhaps because it is known experimentally that long strings are copied from each parent? Perhaps every marker belongs to a different megabase, so the first 100 values are copied from one parent, the next 100 from the other parent, the next 100 from the first parent, the next 100 from the second parent, and the final 94 from the first parent? It would help a lot if you could explain this in detail.

This important paper should be definitely published in *Nature Comm.*, even if the authors refuse to introduce these changes. But in my view, they would make their nice work clearer and sounder.